# Comparison of aqueous SOA product distributions from guaiacol oxidation by non-phenolic and phenolic methoxybenzaldehydes as photosensitizers in the absence and presence of ammonium nitrate

Beatrix Rosette Go Mabato[1,2], Yong Jie Li[3], Dan Dan Huang[4], Yalin Wang[3], and Chak K. Chan[1,2]*

[1]School of Energy and Environment, City University of Hong Kong, Hong Kong, China
[2]City University of Hong Kong Shenzhen Research Institute, Shenzhen, China
[3]Department of Civil and Environmental Engineering, and Centre for Regional Ocean, Faculty of Science and Technology, University of Macau, Macau, China
[4]Shanghai Academy of Environmental Sciences, Shanghai 200233, China

*Correspondence to*: Chak K. Chan (Chak.K.Chan@cityu.edu.hk)

**Abstract.** Aromatic carbonyls (e.g., methoxybenzaldehydes), an important class of photosensitizers, are abundant in the atmosphere. Photosensitization and nitrate-mediated photo-oxidation can occur simultaneously, yet studies about their interactions, particularly for aqueous secondary organic aerosol (aqSOA) formation, remain limited. This study compared non-phenolic (3,4-dimethoxybenzaldehyde, DMB) and phenolic (vanillin, VL) methoxybenzaldehydes as photosensitizers for aqSOA formation via guaiacol (GUA) oxidation in the absence and presence of ammonium nitrate (AN) under atmospherically relevant cloud and fog conditions. GUA oxidation by triplet excited states of DMB ($^3$DMB*) (GUA+DMB) was ~4 times faster and exhibited greater light absorption than oxidation by $^3$VL* (GUA+VL). Both GUA+DMB and GUA+VL formed aqSOA composed of oligomers, functionalized monomers, oxygenated ring-opening species, and N-containing products in the presence of AN. The observation of N-heterocycles such as imidazoles indicates the participation of ammonium in the reactions. The majority of generated aqSOA are potential brown carbon (BrC) chromophores. Oligomerization and functionalization dominated in GUA+DMB and GUA+VL, but functionalization appeared to be more important in GUA+VL due to contributions from VL itself. AN did not significantly affect the oxidation kinetics, but it had distinct effects on the product distributions, likely due to differences in the photosensitizing abilities and structural features of DMB and VL. In particular, the more extensive fragmentation in GUA+DMB than in GUA+VL likely generated more N-containing products in GUA+DMB+AN. In GUA+VL+AN, the increased oligomers may be due to VL-derived phenoxy radicals induced by ˙OH or ˙NO$_2$ from nitrate photolysis. Furthermore, increased nitrated products observed in the presence of both DMB or VL and AN than in AN alone implies that photosensitized reactions may promote nitration. This work demonstrates how the structural features of photosensitizers affect aqSOA formation via non-carbonyl phenol oxidation. Potential interactions between photosensitization and AN photolysis were also elucidated. These findings facilitate a better understanding of photosensitized aqSOA formation and highlight the importance of AN photolysis in these reactions.

## 1 Introduction

Photosensitized reactions involving triplet excited states of organic compounds ($^3$C*) are efficient pathways for the formation of secondary organic aerosol in the aqueous phase (aqSOA; Smith et al., 2014, 2015, 2016; Yu et al., 2014, 2016; Chen et al., 2018; Lu et al., 2019; Ye et al., 2019; Chen et al., 2020; Liu et al., 2020; Jiang et al., 2021; Ma et al., 2021; Misovich et al., 2021; Ou et al., 2021; F. Li et al., 2022; X. Li et al., 2022; Aregahegn et al., 2022; Mabato et al., 2022; Wang et al., 2022). Upon irradiation by solar radiation, photosensitizers form an excited triplet state that directly reacts with substrates (e.g., phenols), and can generate singlet oxygen ($^1O_2$), superoxide ($O_2^{\bullet-}$) or hydroperoxyl ($^{\bullet}HO_2$) radicals, and hydroxyl radicals ($^{\bullet}OH$) upon reactions with $O_2$ and substrates (George et al., 2018; Chen et al., 2020), thereby facilitating the oxidation of rather volatile species and contributing to aqSOA formation. An important class of photosensitizers is aromatic carbonyls (e.g., methoxybenzaldehydes) which are abundant in aerosol particles, cloud waters, and fog waters (Anastasio et al., 1997; Felber et al., 2021). Aromatic carbonyls can be emitted from anthropogenic sources and biomass burning (BB; Lipari et al.,1984; Edye and Richards, 1991; Hawthorne et al., 1992; Simoneit et al., 1993, 1999; Anastasio et al., 1997; Felber et al., 2021), or formed via atmospheric oxidation of aromatic hydrocarbons (Hoshino et al., 1978; Calvert and Madronich, 1987; Anastasio et al., 1997; Felber et al., 2021). BB is also a significant source of phenols through lignin pyrolysis (Simpson et al., 2005). Phenolic carbonyls have a hydroxyl (–OH) group on the aromatic ring, whereas non-phenolic carbonyls do not. BB smoke has been reported to have comparable concentrations of phenolic and non-phenolic carbonyls (Simoneit et al., 1993; Anastasio et al., 1997).

Most previous studies on aqSOA formation via photosensitized non-carbonyl phenol oxidation have examined 3,4-dimethoxybenzaldehyde (DMB), a non-phenolic methoxybenzaldehyde, as the photosensitizer (Smith et al., 2014, 2015; Yu et al., 2014, 2016; Ye et al., 2019; Chen et al., 2020; Jiang et al., 2021; Ma et al., 2021; Misovich et al., 2021; Ou et al., 2021; X. Li et al., 2022). By contrast, phenolic carbonyls have been mainly studied as aqSOA precursors via $^{\bullet}OH$-, nitrate-, nitrite-, and $^3$DMB*-mediated oxidation (Li et al., 2014; Huang et al., 2018; Pang et al., 2019; Jiang et al., 2021; Misovich et al., 2021). However, strongly light-absorbing phenolic carbonyls (e.g., molar absorptivity above 300 nm $\geq 7 \times 10^3$ M$^{-1}$ cm$^{-1}$) can also serve as photosensitizers to promote aqSOA formation (Smith et al., 2016; Mabato et al., 2022). For instance, the direct photosensitized oxidation of phenolic carbonyls (i.e., oxidation of phenolic carbonyls by their $^3$C* or $^3$C*-derived oxidants) such as vanillin (VL; another methoxybenzaldehyde) efficiently form low-volatility products, with aqSOA mass yields of up to 140% (Smith et al., 2016). Moreover, the aqSOA mass yields from the oxidation of syringol by $^3$DMB* and $^3$VL* are similar (111% and 114%, respectively; Smith et al., 2014, 2016). In addition, we recently reported that the direct photosensitized oxidation of VL and guaiacol oxidation by $^3$VL* yield similar products (oligomers, functionalized monomers, and oxygenated ring-opening products) as observed with $^3$DMB* (Yu et al., 2014; Mabato et al., 2022). Guaiacol is a non-carbonyl BB methoxyphenol with an emission rate from fireplace wood combustion in the range of 172 to 279 mg/kg (Schauer et al., 2001; Simoneit, 2002). The atmospheric reactivity of methoxyphenols has recently been reviewed (Liu et al., 2022). However, our previous experiments (Mabato et al., 2022) were performed at a concentration (0.1 mM VL)

higher than what was typically used for DMB (0.005 to 0.01 mM; Smith et al., 2014, 2015; Yu et al., 2014, 2016). Therefore,
direct comparisons between photosensitization by $^3$DMB* and $^3$VL* cannot be made. Despite the above findings, much is
still unknown about how aqSOA formation proceeds in systems using phenolic carbonyls as photosensitizers.
BB aerosols are typically internally mixed with other aerosol components, such as ammonium nitrate (AN;
Zielinski et al., 2020). Hence, aromatic carbonyls and phenols may coexist with AN in BB aerosols. Nitrate and ammonium
facilitate the formation of aqSOA and brown carbon (BrC) via a number of pathways. Nitrate photolysis can produce $^•$OH
and nitrating agents (e.g., $^•$NO$_2$; Minero et al., 2007; Huang et al., 2018; Mabato et al., 2022; Wang et al., 2022; Yang et al.,
2022), and ammonium reacts with carbonyls to yield N-containing heterocycles (e.g., imidazoles) and oligomers capable of
UV-Vis light absorption (De Haan et al., 2009, 2011; Nozière et al., 2009, 2010, 2018; Shapiro et al., 2009; Yu et al., 2011;
Lee et al., 2013; Powelson et al., 2014; Gen et al., 2018; Grace et al., 2019; Mabato et al., 2019). Furthermore, nitrate
photolysis may be an important process for SO$_2$ oxidation and SOA formation in the particle phase (Gen et al., 2019a,
2019b, 2022; Zhang et al., 2020, 2021, 2022), and it can potentially modify the morphology of atmospheric viscous particles
(Liang et al., 2021). Yet, understanding of the effects of inorganic nitrate on aqSOA formation remains limited. In addition,
aqSOA formation studies involving aromatic carbonyls and phenols have probed either photosensitization or nitrate-
mediated photo-oxidation, but these reactions can occur simultaneously. For instance, we previously reported nitrated
compounds, including a potential imidazole derivative from the direct photosensitized oxidation of VL in the presence of AN
(Mabato et al., 2022). Accordingly, investigations on reaction systems including both photosensitizers and AN may provide
further insights into the aqueous-phase processing of BB aerosols.
In this work, we compared aqSOA formation from photosensitized guaiacol (GUA) oxidation by $^3$C* of non-
phenolic and phenolic methoxybenzaldehydes under identical conditions (simulated sunlight and reactants concentration)
relevant to cloud and fog waters. The effects of AN on photosensitized aqSOA formation were also examined. In this study,
the dominant aqSOA precursor is GUA (Henry's law constant of $9.2 \times 10^2$ M atm$^{-1}$; Sagebiel et al., 1992), and DMB and VL
were used as photosensitizers to oxidize GUA. DMB and VL (Henry's law constants of $7.3 \times 10^3$ M atm$^{-1}$ and $4.7 \times 10^5$ M
atm$^{-1}$, respectively; Yaws, 1994; EPI Suite version 4.1, 2012; Felber et al., 2021), which are also abundant in BB emissions
(Schauer et al., 2001; Li et al., 2014; Chen et al., 2017; Pang et al., 2019; Mabato et al., 2022) and whose structures differ
only by one functional group (–OCH$_3$ for the former and –OH for the latter, Fig. 1), represented non-phenolic and phenolic
methoxybenzaldehydes, respectively. The structures of GUA, DMB, and VL are provided in Figure 1. Based on their
quantum yield of $^3$C* formation, DMB and VL have been classified as moderate and poor photosensitizers, respectively
(Felber et al., 2021). The photosensitized oxidation of GUA by $^3$DMB* or $^3$VL* in the absence (and presence) of AN are
referred to as GUA+DMB(+AN) and GUA+VL(+AN), respectively. GUA photo-oxidation by AN alone (GUA+AN) was
also explored for comparison with GUA+DMB+AN and GUA+VL+AN. The molar absorptivities of GUA, DMB, VL, and
nitrate are shown in Figure 1. The precursor and photosensitizer decay kinetics, detected products, and absorbance
enhancement were used to characterize the reactions. However, it should be noted that we mainly focused on the analyses of
the reaction products and product distribution.

While several studies on photo-oxidation of BB emissions are available, this work focuses on the comparison

between non-phenolic and phenolic methoxybenzaldehydes as photosensitizers in the absence and presence of AN for
aqSOA formation. We found that GUA oxidation by $^3$DMB* was faster and exhibited greater light absorption relative to
GUA+VL. These are likely attributed to the stronger photosensitizing ability of DMB and the –OH group of VL, making it
more prone to oxidation and more reactive towards electrophilic aromatic substitution. Oligomerization and functionalization
dominated in GUA+DMB and GUA+VL, but functionalization appeared to be more significant in GUA+VL due to VL
transformation products. Although AN did not significantly influence the oxidation kinetics due to the predominant role of
photosensitizer chemistry compared to nitrate, AN promoted the formation of N-containing products. These include N-
heterocycles (e.g., imidazoles), suggesting the participation of ammonium in the reactions. Moreover, the product
distributions indicate distinct interactions between photosensitization by $^3$DMB* and $^3$VL* and AN photolysis. In particular,
AN generated more N-containing products in GUA+DMB+AN than in GUA+VL+AN, and increased the oligomers in
GUA+VL+AN. Furthermore, increased nitrated compounds in GUA+DMB+AN and GUA+VL+AN compared to GUA+AN
suggest that photosensitized reactions may promote reactions by nitrate photolysis.
**2 Methods**
**2.1 Aqueous phase photo-oxidation experiments**
Procedures for the photo-oxidation experiments are presented in detail in our previous study (Mabato et al., 2022).
Experimental solutions were prepared using 0.1 mM guaiacol (GUA, Sigma Aldrich, ≥98.0%) and 0.01 mM 3,4-
dimethoxybenzaldehyde (DMB, Acros Organics, 99+%) or 0.01 mM vanillin (VL, Acros Organics, 99%, pure), in the
absence and presence of ammonium nitrate (1 mM; AN, Acros Organics, 99+%, for analysis). These GUA and
methoxybenzaldehydes concentrations are within the values expected in cloud or fog drops in areas with significant wood
combustion (Anastasio et al., 1997; Rogge et al., 1998; Nolte et al., 2001). The AN concentration represents values usually
observed in cloud and fog waters (Munger et al., 1983; Collett et al., 1998; Zhang and Anastasio, 2003; Li et al., 2011;
Giulianelli et al., 2014; Bianco et al., 2020). It must be noted that this study did not intend to identify the AN concentrations
that would affect the kinetics but attempted to analyze the effects of AN on photosensitized aqSOA formation. A solution
composed of 0.1 mM GUA and 1 mM AN (GUA+AN) was also examined for comparison with GUA+DMB+AN and
GUA+VL+AN. Sulfuric acid ($H_2SO_4$; Acros Organics, ACS reagent, 95% solution in water) was used to adjust the pH of the
solutions to 4, which is within typical cloud pH values (2–7; Pye et al., 2020) and pH values observed in wood burning-
impacted cloud and fog waters (Collett et al., 1998; Raja et al., 2008). The solutions (initial volume of 500 mL) were
bubbled with synthetic air (0.5 $dm^3$/min) for 30 min before irradiation and throughout the reactions to achieve air-saturated
conditions (Du et al., 2011; Chen et al., 2020) and were continuously magnetically stirred. In this study, the reactions can
generate $^3$DMB*/$^3$VL* and secondary oxidants ($^1O_2$, $O_2^{\cdot-}$/$^{\cdot}HO_2$, $^{\cdot}OH$) but not ozone. Solutions contained in a quartz
photoreactor were irradiated using a xenon lamp (model 6258, Ozone free xenon lamp, 300 W, Newport) equipped with a

longpass filter (20CGA-305 nm cut-on filter, Newport) to eliminate light below 300 nm. The reaction temperatures were maintained at $27 \pm 2$ °C using cooling fans positioned around the photoreactor and lamp housing. The averaged initial photon flux in the reactor measured from 300 to 380 nm was $\sim 3 \times 10^{15}$ photons cm$^{-2}$ s$^{-1}$ nm$^{-1}$ (Fig. 1), similar to our previous work (Mabato et al., 2022). Samples were collected every 30 min for 180 min for offline analyses of (1) GUA, DMB, and VL concentrations using ultra-high-performance liquid chromatography with photodiode array detector (UHPLC-PDA) and (2) absorbance measurements using UV-Vis spectrophotometry. Moreover, the samples collected before and after irradiation (180 min) were analyzed for (3) reaction products using UHPLC coupled with heated electrospray ionization Orbitrap mass spectrometry (UHPLC-HESI-Orbitrap-MS) operated in positive and negative ion modes and (4) concentrations of small organic acids using ion chromatography (IC). Each experiment was repeated independently at least three times. The reported decay rate constants, small organic acids concentration, and absorbance enhancement were averaged from triplicate experiments, and the corresponding errors represent one standard deviation. The pseudo-first-order rate constant ($k'$) for GUA decay was determined using the following equation (Huang et al., 2018):

$$ln\left([GUA]_t/[GUA]_0\right) = -k't \qquad \text{(Eq. 1)}$$

where $[GUA]_t$ and $[GUA]_0$ are GUA concentrations at time $t$ and 0, respectively. DMB or VL decay rate constants were calculated by replacing GUA with DMB or VL in Eq. 1. The decay rate constants were normalized to the photon flux measured for each experiment through dividing $k'$ by the measured 2-nitrobenzaldehyde (2NB; a chemical actinometer) decay rate constant, $j$(2NB) (Mabato et al., 2022). In addition, the decay rate constants were corrected for the internal light screening due to DMB, VL, and AN (Leifer, 1988; Zhang and Anastasio, 2003; Smith et al., 2014, 2015, 2016). The values of the internal light screening factor ($S_\lambda$) determined around the peak in the light absorption action spectrum (DMB: 310-335 nm, VL: 304-364 nm, nitrate: 300-331 nm) (Smith et al., 2014, 2015, 2016) for an 8.5 cm cell were 0.95 for GUA+AN, 0.51 for GUA+DMB, 0.54 for GUA+DMB+AN, 0.57 for GUA+VL, and 0.59 for GUA+VL+AN. Moreover, two independently prepared samples for each reaction condition were analyzed using UHPLC-HESI-Orbitrap-MS. Only peaks that were reproducibly detected in both sets of samples were considered. For clarity, the formulas discussed in this work correspond to neutral analytes (e.g., with $H^+$ or $NH_4^+$ removed from the ion formula). The details of the analytical procedures are provided in the Supplement (Sects. S1 to S4).

## 2.2 Calculation of normalized abundance of products

Several recent studies have used comparisons of relative abundance of products based on peak areas from mass spectrometry (MS) results (e.g., Lee et al., 2014; Romonosky et al., 2017; Wang et al., 2017; Fleming et al., 2018; Song et al., 2018; Klodt et al., 2019; Ning et al., 2019) to show the relative importance of different types of compounds (K. Wang et al., 2021). However, comparisons of relative abundance among different compounds can be subject to uncertainties as ionization efficiencies in soft ionization, such as ESI, may significantly vary between different compounds (Kebarle, 2000; Schmidt et al., 2006; Leito et al., 2008; Perry et al., 2008; Kruve et al., 2014). In our previous work (Mabato et al., 2022), we introduced the normalized abundance of products ([P], unitless) (Eq. 2) as a semi-quantitative analysis that gives an overview of how

the signal intensities changed under different experimental conditions but not the quantification of the absolute product
concentration. The calculation assumes equal ionization efficiencies of different compounds, which is commonly used to
estimate O:C ratios of SOA (Bateman et al., 2012; Lin et al., 2012; Laskin et al., 2014; De Haan et al., 2019):

$$[P] = \frac{A_{P,t}}{A_{GUA,t}} \cdot \frac{[GUA]_t}{[GUA]_0}$$
(Eq. 2)

where $A_{P,t}$ and $A_{GUA,t}$ are the extracted ion chromatogram (EIC) peak areas of the product P and GUA from UHPLC-HESI-
Orbitrap-MS analyses at time $t$, respectively; $[GUA]_t$ and $[GUA]_0$ are the GUA concentrations (μM) determined using
UHPLC-PDA at time $t$ and 0, respectively. Note that the normalized abundance of products has intrinsic uncertainties due to
the variability in ionization efficiencies for various compounds. Moreover, it should be noted that the normalized abundance
of products was calculated using only the positive ion mode data as the GUA signal from the negative ion mode was weak
and thus may present large uncertainties during normalization. Therefore, products that may not give signals or may have
weak signals in the positive ion mode were possibly underestimated in the normalized product abundance. Nevertheless, it
enables the comparison of MS results among different experiments. As demonstrated in our previous work (Mabato et al.,
2022) and the current study, a higher normalized abundance of products generally correlates with higher efficiency of
oxidation. The reported uncertainties were propagated from the changes in [GUA] measured using UHPLC-PDA and the MS
signal intensities.

**3 Results and Discussion**
Using kinetics data, MS analyses, and absorbance enhancement data, we first examined the differences between GUA+DMB
and GUA+VL (Sect. 3.1). Then, we analyzed GUA+DMB+AN, GUA+VL+AN, and GUA+AN (Sect. 3.2) to explore the
effects of nitrate photolysis and ammonium on photosensitized aqSOA formation.
**3.1 Comparison of photosensitized GUA oxidation by non-phenolic ($^3$DMB*) and phenolic ($^3$VL*)**
**methoxybenzaldehydes**
Prior studies have reported that photosensitized non-carbonyl phenol oxidation in the presence of 3,4-
dimethoxybenzaldehyde (DMB) and vanillin (VL) (separately) was mainly driven by $^3$DMB* and $^3$VL*, respectively (Smith
et al., 2014; Mabato et al., 2022), while contributions from secondary oxidants such as $^1O_2$ and ·OH were likely minor.
However, both $^3$DMB* and $^3$VL* are efficiently quenched by $O_2$, suggesting that energy transfer should be considered in
evaluating photosensitized processes involving these methoxybenzaldehydes (Felber et al., 2021). Moreover, it was found
that $^3$DMB*, $^1O_2$, and $O_2^{·-}$ were the major contributors to the photosensitized oxidation of 4-ethylguaiacol (Chen et al., 2020).
Recently, the oxidation of guaiacyl acetone (a non-conjugated phenolic carbonyl) in the presence of DMB has been reported
to be initiated by $^3$DMB*, $^1$O$_2$, $^\bullet$OH, or methoxy radical ($^\bullet$OCH$_3$) (Misovich et al., 2021). Further studies are thus required to
identify the specific oxidants in these reaction systems. In this study, reactions initiated in the presence of DMB or VL are
collectively referred to as photosensitized reactions. The reaction conditions, initial guaiacol (GUA) and DMB or VL decay
rate constants, normalized product abundance, and the chemical characteristics of aqSOA formed in this work are
summarized in Table 1.

### 3.1.1 Kinetic analysis of photosensitization by $^3$DMB* and $^3$VL*

No significant loss of GUA or photosensitizers was observed for dark experiments ($p > 0.05$). Figure S1 shows the decay of
GUA, DMB, and VL under different experimental conditions. Upon irradiation, the GUA decay rate constant in GUA+DMB
was ~4 times higher than in GUA+VL. In GUA+DMB, the decay rate constant of GUA was ~8 times higher than that of
DMB, consistent with a previous study (Smith et al., 2014). Contrastingly, the decay rate constant of VL was 2.4 times
higher than that of GUA in GUA+VL. This VL consumption was also observed in our earlier work using 0.1 mM GUA +
0.1 mM VL (Mabato et al., 2022). These trends could be explained by the following reasons. First, DMB has a stronger
photosensitizing ability than VL based on its higher quantum yield of $^3$C* formation and longer lifetime of $^3$DMB*
compared to $^3$VL* (Felber et al., 2021). Second, VL is also a phenolic compound similar to GUA, and is therefore highly
reactive towards oxidation. For instance, its –OH group can be oxidized by $^3$VL* via H-atom abstraction to form phenoxy
radicals which can undergo coupling to form oligomers (Kobayashi and Higashimura, 2003; Sun et al., 2010; Mabato et al.,
2022). The faster consumption of VL than GUA suggests a competition between ground-state VL and GUA for reaction with
$^3$VL*. Moreover, compared to a –OCH$_3$ group (in DMB), an –OH group (in VL) has a stronger electron-donating ability and
is thus more activating towards electrophilic aromatic substitution. It should be noted that the differences in the GUA decay
rate constants among different reaction systems are not quantitatively equivalent to photosensitizing efficiencies, and a
detailed quantitative analysis of which is beyond the scope of this study. Nonetheless, these results suggested that GUA
oxidation in GUA+DMB was overall more efficient than in GUA+VL. Our kinetic analysis focused on the decay rate
constants of the aqSOA precursor (GUA) and the photosensitizers (DMB and VL) during photosensitization under the same
experimental conditions (same aqSOA precursor and concentration, same photosensitizer concentration, and same lamp
photon flux). The effects of other factors (e.g., intersystem crossing efficiency) on the rate constants were not examined.
Explicit kinetic studies (e.g., Smith et al., 2014, 2015) that measure second-order rate constants should be conducted in the
future to extend the applicability of the kinetic parameters to other conditions.

### 3.1.2 Product distributions and chemical characteristics of aqSOA from photosensitization by $^3$DMB* and $^3$VL*

The products detected using UHPLC-HESI-Orbitrap-MS were used to characterize the aqSOA formed in this work. The
signal-weighted distributions of aqSOA calculated from combined positive (POS) and negative (NEG) ion modes MS results
are summarized in Figure 2. The signal-weighted distributions calculated separately from POS and NEG ion modes MS
results are available in Figures S2 and S3. It should be noted that in this work, the product distributions for all experiments

were based on the same irradiation time of 180 min. An irradiation time of 180 min was chosen as it was sufficient to show the differences in the extent of reaction of GUA among the reaction systems studied. For reaction systems with precursors of different reactivities, chemical analysis at a fixed reaction time may be looking at different generations of products of each precursor, as Yu et al. (2014) reported. Measuring the product distribution at a fixed time might have missed the information on what/how many products are formed at the similar amounts of precursors reacted. The situation could be even more complicated if different precursors had major differences in pathways and dominant intermediates. However, comparing the product distributions after a certain time of light exposure, as is the case for this study, is useful to evaluate what products would form after a certain time of photosensitization. Oligomers and derivatives of GUA dominated both GUA+DMB and GUA+VL, in agreement with pronounced oligomerization from triplet-mediated oxidation of relatively high phenol concentration (e.g., 0.1 to 3 mM; Li et al., 2014; Yu et al., 2014, 2016; Slikboer et al., 2015; Ye et al., 2019; Mabato et al., 2022). Figure 3 schematically depicts the main differences between photosensitized GUA oxidation by $^3$DMB* and $^3$VL* in the absence and presence of AN. As shown in Fig. 3, $^3$DMB* and $^3$VL* can oxidize GUA via H-atom abstraction to form phenoxy radicals which undergo coupling to form oligomers (Kobayashi and Higashimura, 2003; Sun et al., 2010; Mabato et al., 2022). The higher oligomer contribution in GUA+DMB is likely due to the better photosensitizing ability of DMB than VL and partly the lower abundance of $^3$VL* due to fast VL consumption. VL was consumed faster than DMB during GUA oxidation ascribable to the –OH group of VL, making it more susceptible to oxidation and more reactive towards electrophilic aromatic substitution. In addition, the normalized product abundance for GUA+DMB was ~4 times higher than that for GUA+VL (Table 1), further suggesting more efficient photosensitized GUA oxidation by $^3$DMB* than by $^3$VL*. The oxidation of GUA or transient organic intermediates by secondary oxidants (e.g., $^1O_2$ and $^•$OH from $^3$DMB* or $^3$VL* and the fragmentation of larger compounds generate highly oxidized ring-opening products (Yu et al., 2014; Huang et al., 2018; Chen et al., 2020). GUA+DMB had a higher contribution of ring-opening products than GUA+VL, likely due to the greater availability of secondary oxidants in the former and fast VL consumption lowering the production of these species in GUA+VL. The IC analyses also indicate the formation of small organic acids (e.g., formic acid), which appeared to have higher concentrations in the presence of DMB than in VL (Fig. S4). Although no data is available for the concentration changes (every 30 min) of small organic acids during the reaction, it is likely that an increasing trend would be observed as fragmentation, which leads to the decomposition of initially formed oligomers and the generation of smaller oxygenated products, becomes important at longer irradiation times (Huang et al., 2018). This trend has also been observed in our previous work on the direct photosensitized oxidation of VL (Mabato et al., 2022), as well as other studies on photosensitized oxidation of non-carbonyl phenols and phenolic carbonyls (e.g., Yu et al., 2016; Jiang et al., 2021). The reactions of secondary oxidants or ring-opening products with GUA can form functionalized products. Notably, the contribution of monomers in GUA+VL was almost twice as high as in GUA+DMB, ascribable to VL transformation products. We previously showed that for the direct photosensitized oxidation of VL, functionalization prevails over oligomerization at 0.01 mM VL, the [VL] used in this work, while oligomerization dominates at higher [VL] (0.1 mM; Mabato et al., 2022).

It has been reported that oligomerization could occur during the electrospray ionization process (Yasmeen et al., 2010). In this work, it was confirmed that the oligomers observed were generated in the solutions via aqueous reactions instead of being artefacts of HESI-MS. This is based on the absence of dimers and higher oligomers in the HESI mass spectra of dark control solutions acquired by direct infusion (Yu et al., 2016).

The major GUA+DMB and GUA+VL products (Tables S1-S2) are mostly oligomers which can be formed through the coupling of phenoxy radicals (Kobayashi and Higashimura, 2003; Sun et al., 2010; Mabato et al., 2022). GUA+DMB products matched those reported in previous works on $^3$DMB*- and/or $^{\bullet}$OH-mediated phenol oxidation (Yu et al., 2014, 2016). These include GUA dimers and trimers (e.g., $C_{14}H_{14}O_4$ and $C_{21}H_{18}O_8$, #1 and 19; Table S1), aldehydes ($C_7H_6O_4$, #13; Table S1), and esters ($C_{16}H_{18}O_6$, #14; Table S1). Functionalized products include $C_{11}H_{12}O_5$ and $C_{10}H_{12}O_3$ (#8 and 12; Table S1). More than half of the major GUA+VL products are the same oligomers detected from GUA+DMB (e.g., $C_{13}H_{10}O_4$ and $C_{20}H_{18}O_6$, #4 and 21; Table S1). The rest are mainly functionalized species such as $C_7H_8O_4$ and $C_8H_8O_5$ (#28 and 35; Table S2), corresponding to a hydroxylated GUA and hydroxylated VL, respectively.

The average elemental ratios and elemental distribution of the products (Fig. S5a–d) were consistent with those in previous studies on similar reaction systems (Yu et al., 2014, 2016; Mabato et al., 2022). The majority of the GUA+DMB and GUA+VL products had H:C ≤1.0 and O:C ≤0.5, typical for aromatic species (Mazzoleni et al., 2012; Kourtchev et al., 2014; Jiang et al., 2021). GUA+DMB had more compounds with higher O:C (≥0.6), in agreement with higher contributions of ring-opening products than in GUA+VL (Fig. 2). The higher ⟨$OS_C$⟩ for GUA+VL than in GUA+DMB (Table 1) was probably due to the significant functionalization in the former. Moreover, the distributions of $OS_C$ and carbon number (Fig. S6a–d) show that these aqSOA products have similar elemental composition to those of low-volatility oxygenated organic aerosols (LV-OOA), semi-volatile oxygenated organic aerosols (SV-OOA), and slightly with biomass burning organic aerosols (BBOA) (Kroll et al., 2011). Further discussions on van Krevelen diagrams (Fig. S5a–d) and $OS_C$ vs. carbon number plots (Fig. S6a–d) for GUA+DMB and GUA+VL aqSOA are available in the Supplement (Sect. S5). In brief, $^3$DMB*-initiated GUA oxidation was faster and yielded higher normalized product abundance than oxidation by $^3$VL*. This is likely due to the stronger photosensitizing ability of DMB than VL and the –OH group of VL facilitating its rapid consumption. In addition, oligomerization and functionalization dominated in both GUA+DMB and GUA+VL, as reported in similar studies (Yu et al., 2014, 2016; Chen et al., 2020; Jiang et al., 2021; Misovich et al., 2021; Mabato et al., 2022). However, functionalization was more prominent in the latter, attributable to the transformation of VL. Nonetheless, it must be noted that for phenolic aqSOA, fragmentation will ultimately be more predominant at longer irradiation times (Huang et al., 2018; Yu et al., 2016; Mabato et al., 2022).

### 3.1.3 Light absorption of aqSOA from photosensitization by $^3$DMB* and $^3$VL*

The absorbance enhancement of phenolic aqSOA generated via reactions with $^3$C* has been linked to the formation of conjugated structures due to oligomerization and functionalization (e.g., additions of hydroxyl and carbonyl groups; Yu et al., 2014, 2016; Smith et al., 2016; Ye et al., 2019; Chen et al., 2020; Jiang et al., 2021; Misovich et al., 2021; Ou et al.,

2021; F. Li et al., 2022; X. Li et al., 2022; Mabato et al., 2022; Wang et al., 2022). Moreover, the aqueous-phase photo-
oxidation of BB emissions can enhance BrC absorbance via the formation of aromatic dimers and functionalized products
(Hems et al., 2020). The increase in light absorption throughout 180 min of irradiation and the change in the rate of sunlight
absorption ($\Delta R_{abs}$) (Jiang et al., 2021) from 350 to 550 nm at 180 min during typical clear and haze days in Beijing, China for
all the reaction systems studied are provided in Figure 4. Figure S7 shows the absorption spectra after 180 min of irradiation
for each reaction system studied. In this work, the absorbance enhancement of GUA+DMB and GUA+VL (Fig. 4a) could be
due to oligomers and functionalized monomers, which are the highest contributors to the product signals. Identifying the
chromophores responsible for the absorbance enhancement may be beneficial in understanding the impact of aqSOA on the
Earth's radiative balance and determining the reactions that affect light absorption by aqSOA (Mabato et al., 2022).
However, the detected products did not exhibit distinct peaks in the UHPLC-PDA chromatograms, likely due to the
concentration of the chromophores being below the detection limit of PDA. Nevertheless, the higher absorbance
enhancement and $\Delta R_{abs}$ for GUA+DMB than GUA+VL was probably due to the higher contribution and normalized
abundance (by ~6 times) of oligomers in the former.

Additional information about aqSOA light absorption can be deduced from the plots of the double bond equivalent
(DBE) values vs. carbon number ($n_C$) (Lin et al., 2018). Figure S8 shows these plots along with the DBE reference values of
fullerene-like hydrocarbons (Lobodin et al., 2012), cata-condensed polycyclic aromatic hydrocarbons (PAHs; Siegmann and
Sattler, 2000), and linear conjugated polyenes with a general formula $C_xH_{x+2}$. The shaded area indicates a sufficient level of
conjugation for visible light absorption, and species within this region are potential BrC chromophores. GUA+DMB and
GUA+VL aqSOA exhibited a significant overlap in the DBE vs. $n_C$ space; nearly all products from both systems, including
the high-relative-abundance species, are potential BrC chromophores. GUA+DMB had more oligomeric products with high
relative abundance ($n_C \geq 12$ and DBE $\geq 8$). For GUA+VL, high-relative-abundance products also include monomeric species
($n_C = 7-8$ and 4-5 DBE) corresponding to hydroxylated products (e.g., $C_7H_8O_4$ and $C_8H_8O_5$; 28 and 35; Table S2). These
observations further indicate the importance of oligomerization and functionalization for the absorbance enhancement of
aqSOA generated via photosensitization by $^3DMB^*$ and $^3VL^*$. In summary, $^3DMB^*$ and $^3VL^*$ can oxidize GUA resulting in
aqSOA and BrC formation, but GUA+DMB products exhibited stronger light absorption. In GUA+VL, the extent of GUA
oxidation was limited by significant VL consumption.
**3.2 Comparison of photosensitized GUA oxidation by non-phenolic ($^3DMB^*$) and phenolic ($^3VL^*$)**
**methoxybenzaldehydes in the presence of AN**
**3.2.1 Kinetic analysis of photosensitization by $^3DMB^*$ and $^3VL^*$ in the presence of AN**
Ammonium nitrate (AN) did not significantly affect ($p > 0.05$) the decay rate constants of GUA, DMB, and VL for both
GUA+DMB+AN and GUA+VL+AN (Table 1), likely due to the higher molar absorptivities of the photosensitizers
compared to that of nitrate. This implies that the chemistry of $^3DMB^*$ and $^3VL^*$ dominated that of nitrate. In this work, the
GUA decay rate constants decreased in the order of GUA+DMB/GUA+DMB+AN > GUA+VL/GUA+VL+AN > GUA+AN

(Table 1). Note that as the molar absorptivities of the photosensitizers are higher than that of nitrate, the kinetics data were also analyzed on a per-photon-absorbed basis for a more appropriate comparison of reaction efficiency (Sect. S6). The apparent quantum efficiency of GUA photodegradation ($\varphi_{GUA}$) in the presence of nitrate (GUA+AN: $0.17 \pm 3.8 \times 10^{-2}$) was ~2 and ~7 times higher than that in the presence of DMB ($0.10 \pm 2.9 \times 10^{-3}$) or VL ($0.026 \pm 7.2 \times 10^{-3}$), respectively. This suggests that nitrate-mediated GUA photo-oxidation is more efficient than photosensitization by $^3$DMB* or $^3$VL* on a per-photon-absorbed basis.

**3.2.2 Product distributions and chemical characteristics of aqSOA from photosensitization by $^3$DMB* and $^3$VL* in the presence of AN**

For both GUA+DMB+AN and GUA+VL+AN, AN had no significant effect on the normalized product abundance (Table 1), but it induced the formation of N-containing products composed of N-heterocycles (e.g., imidazoles and pyridines) and oligomers, as well as nitrated species. Similarly, we previously reported a potential imidazole derivative from the direct photosensitized oxidation of VL in the presence of AN, which was attributed to the reaction of ring-opening products with dissolved ammonia (Mabato et al., 2022). Oligomers remained the highest signal contributors in the presence of AN (Fig. 2), but interactions between photosensitization by $^3$DMB* and $^3$VL* and AN photolysis were distinct. First, nitrated species had similar contributions in both cases, but the contribution and normalized abundance of all N-containing products in GUA+DMB+AN were 2 and ~14 times higher, respectively, than in GUA+VL+AN. This difference can be attributed to the higher contribution of N-heterocycles and N-containing oligomers in GUA+DMB+AN. Compared to GUA+VL, GUA+DMB had a higher contribution of ring-opening products which can react with ammonia, as discussed earlier (Figs. 2 and 3). Second, the decrease in oligomers in GUA+DMB+AN may be due to their fragmentation induced by $^\bullet$OH from nitrate photolysis, then conversion to N-containing products. Correspondingly, the contribution of possibly ring-retaining N-containing products in GUA+DMB+AN (18.6%) was ~3 times higher than that in GUA+VL+AN (6.5%). While fragmentation of oligomers likely occurred in GUA+VL+AN as well, the increase in oligomers suggests that other reactions have taken place. For GUA+VL+AN, $^\bullet$OH or $^\bullet$NO$_2$ from nitrate photolysis may have initiated H-atom abstraction from the –OH group of VL, generating phenoxy radicals which can undergo coupling to form more oligomers (Kobayashi and Higashimura, 2003; Sun et al., 2010; Mabato et al., 2022). This may also explain the more significant decrease of monomers in GUA+VL+AN (~3 times) compared to GUA+DMB+AN (~2 times). Similarly, we previously observed an increase in oligomers during the direct photosensitized oxidation of 0.01 mM VL (Mabato et al., 2022), the [VL] used in this work, upon adding 1 mM AN. These findings indicate that photosensitization by non-phenolic and phenolic methoxybenzaldehydes may interact differently with AN photolysis.

GUA+AN mainly formed oligomers analogous to $^\bullet$OH-mediated phenol oxidation (Yu et al., 2014, 2016), followed by N-containing products. The normalized product abundance of GUA+AN was the lowest among all experiments, likely due to the lower GUA decay constant relative to photosensitized oxidation. Moreover, the normalized abundance of N-containing products in GUA+AN was ~12 times lower than that in GUA+DMB+AN but comparable to that in

GUA+VL+AN. This discrepancy for GUA+VL+AN might be due to the weaker signals of its N-containing products in the positive compared to the negative ion mode. As previously mentioned, the normalized product abundance was calculated using only the positive ion mode data as the GUA signal from the negative ion mode was weak and thus may present large uncertainties during normalization. Interestingly, the contributions from nitrated species in GUA+DMB+AN and GUA+VL+AN were higher than in GUA+AN, suggesting possible enhancement of nitration reactions. This is likely due to the increased formation of $\cdot NO_2$, for instance, via the reactions of $\cdot OH$ and $O_2^{\cdot-}$ (from $^3DMB^*$ or $^3VL^*$) with $NO_2^-$ (Pang et al., 2019; Mabato et al., 2022). Similarly, we previously reported enhanced nitration via the direct photosensitized oxidation of VL in the presence of AN under air-saturated conditions ($O_2$ is present) relative to nitrogen-saturated conditions (Mabato et al., 2022). These imply that photosensitization may promote reactions induced by nitrate photolysis.

The major products from GUA+DMB+AN, GUA+VL+AN, and GUA+AN (Tables S3–S5) include oligomers and functionalized monomers detected in GUA+DMB and GUA+VL (Tables S1–S2). The N-heterocycles from GUA+DMB+AN include $C_6H_6N_4$ (#41; Table S3), which may be 2,2'-biimidazole (BI), a reaction product from glyoxal + reduced nitrogenous compounds (e.g., ammonium salts) (De Haan et al., 2009; Galloway et al., 2009; Nozière et al., 2009; Shapiro et al., 2009; Yu et al., 2011; Kampf et al., 2012; Gen et al., 2018; Mabato et al., 2019). The nitrated products include $C_{12}H_{11}N_3O_3$ and $C_{15}H_{10}N_4O_3$ (#42 and 49; Table S3), which possibly have a nitrated imidazole moiety and a nitrophenol moiety, respectively. For GUA+VL+AN, oligomers ($C_{14}H_{12}O_6$ and $C_{20}H_{16}O_7$; #55 and 59, Table S4) which were not among the major products in GUA+VL were noted. $C_{10}H_8O_2$ likely has a furanone group (#50; Table S4); furanones are the primary products of the reaction of $\cdot OH$ with toluene and other aromatic hydrocarbons (Smith et al., 1999). Moreover, $C_{11}H_9N_3O_3$ (#57; Table S4) has a nitrated imidazole moiety. Among the N-containing compounds in GUA+AN is $C_4H_3N_3O_3$ (#69; Table S5), which may be a nitrated imidazole-2-carboxaldehyde. Imidazole-2-carboxaldehyde is also a reaction product from glyoxal + reduced nitrogenous compounds (e.g., ammonium salts) (De Haan et al., 2009; Galloway et al., 2009; Nozière et al., 2009; Shapiro et al., 2009; Yu et al., 2011; Kampf et al., 2012; Gen et al., 2018; Mabato et al., 2019).

The ⟨O:C⟩ for GUA+DMB+AN and GUA+VL+AN were lower than those in the absence of AN (Table 1), possibly due to the formation of N-heterocycles, altering the elemental ratios. The ⟨O:C⟩ and ⟨H:C⟩ were comparable in GUA+DMB+AN and GUA+VL+AN, but the ⟨N:C⟩ for the former was higher, implying a greater extent of reactions involving AN. Relative to GUA+DMB+AN and GUA+VL+AN, GUA+AN had a higher ⟨N:C⟩, as can be expected given that AN was the only oxidant source. The lower ⟨$OS_C$⟩ of GUA+DMB+AN and GUA+VL+AN compared to GUA+AN may be attributed to triplet-initiated oxidation generating higher-molecular-weight products with less fragmentation compared to $\cdot OH$-mediated oxidation (Yu et al., 2014; Chen et al., 2020). Nonetheless, AN generally increased the ⟨$OS_C$⟩ for both GUA+DMB and GUA+VL, with a more noticeable increase for the former, suggesting more oxidized products. Similarly, in a previous work, the more oxygenated and oxidized aqSOA from the photo-oxidation of phenolic carbonyls in AN solutions than in ammonium sulfate solutions has been ascribed to nitrate photolytic products promoting the reactions (Huang et al., 2018). Furthermore, GUA+DMB+AN and GUA+VL+AN aqSOA had mainly similar features in the $OS_C$ vs. $n_C$ plots as those observed in the absence of AN (Fig. S6). More information on van Krevelen diagrams (Figs. S5e–h and S9) and $OS_C$

vs. $n_C$ plots (Figs. S6e–h and S10) for GUA+DMB+AN, GUA+VL+AN, and GUA+AN aqSOA are provided in the
Supplement (Sect. S7). In essence, AN had no significant effect on the decay kinetics ascribable to photosensitizer chemistry
prevailing over nitrate, but it induced the formation of N-containing products. Moreover, AN modified the product
distributions, albeit in different ways (Figs. 2 and 3). In particular, N-containing products were more abundant in
GUA+DMB+AN, probably due to more extensive fragmentation in GUA+DMB than in GUA+VL. In GUA+VL+AN, AN
promoted oligomer formation likely via the -OH group of VL. Furthermore, GUA+DMB+AN and GUA+VL+AN had more
nitrated products than GUA+AN, suggesting that photosensitized reactions may promote nitrate photolysis-initiated
reactions.
**3.2.3 Light absorption of aqSOA from photosensitization by [3]DMB\* and [3]VL\* in the presence of AN**
The presence of AN also did not appreciably affect the absorbance enhancement and $\Delta R_{abs}$ for both GUA+DMB+AN and
GUA+VL+AN (Fig. 4). For GUA+DMB+AN, the N-containing products may have offset the decrease in oligomers to
maintain the absorbance enhancement observed from GUA+DMB. Wang et al. (2022) reported that nitration might
contribute significantly to absorbance enhancement for methoxyphenols in sodium nitrate. In GUA+VL+AN, the decrease in
monomers may have counteracted the increased oligomers and the generated N-containing products. Compared to
GUA+DMB+AN, the N-containing products from GUA+VL+AN probably had less impact on the absorbance enhancement
based on their smaller signal contribution.

Similar to experiments without AN, CHO species from GUA+DMB+AN and GUA+VL+AN were mainly

overlapped in the DBE vs. $n_C$ space (Fig. S8c,d) and were mostly potential BrC chromophores. In both systems, GUA dimers
were the products with the highest relative abundance. For GUA+DMB+AN, products with high relative abundance also
include a CHN species, while two CHON species had high $n_C$ (18,20) and DBE (16,14) values. In GUA+VL+AN, products
with high relative abundance include a CHON species ($n_C$ = 11 and 9 DBE). Approximately 30% and 43% of the N-
containing products for GUA+DMB+AN and GUA+VL+AN, respectively, were among the potential BrC chromophores.
This suggests the possible significance of N-containing products for light absorption of aqSOA from photosensitization by
methoxybenzaldehydes and AN photolysis. Correspondingly, nitroaromatic compounds and N-heterocycles are frequently
noted in BBOA (Iinuma et al., 2010; Kitanovski et al., 2012; Kourtchev et al., 2016) and have been proposed to be potential
contributors to BrC light absorption (Laskin et al., 2015). Relative to GUA+DMB+AN and GUA+VL+AN, only 19% of the
N-containing products in GUA+AN were potential BrC chromophores (Fig. S8e,f), and these did not include CHN species.
These indicate that the N-containing products formed in the presence of both photosensitizers and AN may be more
significant contributors to the light absorption of phenolic aqSOA than those formed in AN only.

## 4 Conclusions and atmospheric implications

The photosensitized oxidation of guaiacol (GUA) by triplet excited states of 3,4-dimethoxybenzaldehyde ($^3$DMB*) and vanillin ($^3$VL*) (separately) in the absence and presence of ammonium nitrate (AN) were compared under identical conditions (simulated sunlight and reactants concentration) relevant to atmospheric cloud and fog waters. Compared to GUA+VL, faster GUA oxidation and stronger light absorption by the products were observed in GUA+DMB. Moreover, VL was consumed faster relative to DMB, limiting the extent of GUA oxidation in GUA+VL. These differences are rooted in DMB having a better photosensitizing ability than VL and the –OH group of VL, making it more susceptible to oxidation and more reactive towards electrophilic aromatic substitution. Both GUA+DMB and GUA+VL generated aqSOA (including potential BrC chromophores) composed of oligomers, functionalized monomers, oxygenated ring-opening products, and N-containing products in the presence of AN. The major aqSOA formation processes for GUA+DMB and GUA+VL were oligomerization and functionalization, but functionalization appeared to be more significant in GUA+VL due to VL transformation products. The photochemical evolution of aqSOA from GUA+DMB has been reported by Yu et al. (2016). Similar experiments for aqSOA from GUA+VL should be conducted in the future to better understand photosensitized reactions involving phenolic carbonyl photosensitizers.

AN did not significantly affect the decay kinetics due to the predominant effect of $^3$DMB* and $^3$VL* chemistry compared to nitrate, but it promoted the formation of N-containing products; these are composed of N-heterocycles (e.g., imidazoles) and oligomers and nitrated species. The observation of N-heterocycles agrees with our previous findings that ammonium participates in photosensitized oxidation of phenolic compounds in the presence of AN (Mabato et al., 2022). These results also suggest that photosensitized oxidation of phenolic compounds in the presence of AN might be an important source of N-heterocycles and nitrated products. Identifying the sources of N-heterocycles and nitrated compounds is important due to their environmental and health impacts (Laskin et al., 2009). Moreover, photosensitized reactions by non-phenolic and phenolic methoxybenzaldehydes may be differently influenced by AN photolysis. For instance, the more extensive fragmentation in GUA+DMB than in GUA+VL possibly resulted in more N-containing products in GUA+DMB+AN. Furthermore, the increased oligomers in GUA+VL+AN may be due to VL-derived phenoxy radicals induced by $^{\bullet}$OH or $^{\bullet}$NO$_2$ from nitrate photolysis. In addition, more nitrated compounds observed in GUA+DMB+AN and GUA+VL+AN than in GUA+AN imply that photosensitized reactions may promote nitrate-mediated photolytic reactions. On a related note, the significance of photosensitization by BrC (via formation of solvated electrons; Y. Wang et al., 2021) and marine dissolved organic matter (via O$_2^{\bullet-}$ formation; Garcia et al., 2021) in enhanced nitrite production from nitrate photolysis have been reported. A recent study from our group has shown that glyoxal photo-oxidation mediated by both nitrate photolysis and photosensitization can significantly enhance the atmospheric sink of glyoxal (Zhang et al., 2022). Further studies are needed to improve our understanding of the interplay between photosensitized reactions and nitrate photolysis.

This study demonstrates that the structural features of photosensitizers affect aqSOA formation via non-carbonyl phenol oxidation. The VL results are broadly relevant to other phenolic carbonyls, but the effects of different functional groups should still be considered. For instance, the aldehyde/ketone pair of syringaldehyde and acetosyringone, both phenolic carbonyls, have been reported to have equal reactivity towards direct photosensitized oxidation. This is due to the greater light absorption by the aldehyde form but higher quantum efficiency for loss for the ketone form (Smith et al. 2016). However, more aqSOA was observed from syringaldehyde than acetosyringone (in either AN or ammonium sulfate; Huang et al., 2018). Our findings also imply that while the contributions of photosensitization by $^3$VL* (and other phenolic carbonyls with similar photosensitizing abilities) to aqSOA formation would be relatively less compared to that of $^3$DMB* (and other non-phenolic carbonyls with similar photosensitizing abilities), these are not negligible. As both non-phenolic and phenolic carbonyls such as the methoxybenzaldehydes examined in this work are emitted in large amounts from biomass burning, future experiments should probe the aqSOA contribution of a wider variety of photosensitizers. Moreover, further experiments on photosensitized reactions in authentic particulate matter (PM) samples should be conducted in the future. Multicomponent reactions such as GUA+DMB+AN and GUA+VL+AN should also be explored for a more accurate simulation of ambient conditions. These would be useful in assessing the overall impact of photosensitized reactions and AN photolysis on aqSOA formation in areas impacted by biomass burning and high AN concentrations, and for their better representation in aqSOA models.

*Data availability.*

The data used in this publication are available to the community and can be accessed by request to the corresponding author.

*Author contributions.*

BRGM designed and conducted the experiments; BRGM and CKC wrote the paper. All co-authors contributed to the discussion of the manuscript.

*Competing interests.*

The authors declare that they have no conflict of interest.

*Acknowledgments.*

C.K.C. gratefully acknowledges support from the National Natural Science Foundation of China (42075100, 41875142, and 42275104) and Hong Kong Research Grants Council (11304121). Y.J.L. acknowledges funding support from the Science and Technology Development Fund, Macau SAR (File No. 0019/2020/A1), and a multiyear research grant (No. MYRG2022-00027-FST) from the University of Macau. The authors also thank the University Research Facility in Chemical and Environmental Analysis (UCEA) at The Hong Kong Polytechnic University for the use of its UHPLC-HESI-Orbitrap Mass Spectrometer and Dr Sirius Tse and Dr Chi Hang Chow for assistance with sample analyses.

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

**Table 1.** Reaction conditions, initial GUA (and DMB or VL) decay rate constants, normalized abundance of products, average elemental ratios, and average carbon oxidation state ($\langle OS_C \rangle$) in each experiment. The reaction systems consisted of GUA (0.1 mM), DMB (0.01 mM), VL (0.01 mM), and AN (1 mM) under air-saturated conditions after 180 min of simulated sunlight irradiation. The UHPLC-HESI-Orbitrap-MS data were obtained in both positive (POS) and negative (NEG) ion modes.

| Exp no. | Reaction conditions | Initial GUA (and DMB or VL) decay rate constants $(\text{min}^{-1}/\text{s}^{-1})$[a] | Normalized abundance of products[b] | Normalized abundance of N-containing compounds[b] | $\langle O{:}C \rangle$[c] | $\langle H{:}C \rangle$[c] | $\langle N{:}C \rangle$[c] | $\langle OS_C \rangle$[c] |
|---|---|---|---|---|---|---|---|---|
| 1 | GUA+DMB | GUA: 6.3 ± 0.25 DMB: 0.78 ± 0.10 | 376 ± 22 | NA | POS: 0.34 | 0.91 | NA | -0.22 |
| | | | | | NEG: 0.40 | 0.94 | NA | -0.15 |
| 2 | GUA+ DMB+AN | GUA: 5.3 ± 0.50 DMB: 0.69 ± 0.052 | 310 ± 4 | 114 | POS: 0.28 | 0.94 | 0.12 | -0.03 |
| | | | | | NEG: 0.37 | 0.91 | 0.04 | -0.05 |
| 3 | GUA+VL | GUA: 1.5 ± 0.14 VL: 3.6 ± 0.55 | 94 ± 5 | NA | POS: 0.41 | 0.91 | NA | -0.10 |
| | | | | | NEG: 0.40 | 0.94 | NA | -0.14 |
| 4 | GUA+ VL+AN | GUA: 1.6 ± 0.12 VL: 2.9 ± 0.032 | 100 ± 2 | 8 | POS: 0.31 | 1.02 | 0.02 | -0.34 |
| | | | | | NEG: 0.39 | 0.91 | 0.03 | -0.02 |
| 5 | GUA+AN | 0.57 ± 0.036 | 23 ± 1 | 9 | POS: 0.35 | 0.99 | 0.16 | 0.19 |
| | | | | | NEG: 0.38 | 1.01 | 0.05 | -0.08 |

[a]The data fitting was performed in the initial linear region. Each value is the average of results from triplicate experiments, corrected for internal light screening due to DMB, VL, and AN, and normalized to the experimental photon flux. Errors represent one standard deviation. [b]The normalized product abundance was calculated using the data from UHPLC-HESI-Orbitrap-MS in the positive (POS) ion mode as the GUA signal from the negative (NEG) ion mode was weak, which may introduce significant uncertainties during normalization. The uncertainties were propagated from the changes in [GUA] measured using UHPLC-PDA and the MS signal intensities. The samples for experiments without AN (marked with NA) were not analyzed for N-containing compounds. [c]The average elemental ratios ($\langle O{:}C \rangle$, $\langle H{:}C \rangle$, and $\langle N{:}C \rangle$) and $\langle OS_C \rangle$ were based on the UHPLC-HESI-Orbitrap-MS results and estimated using the signal-weighted method (Bateman et al., 2012). The $OS_C$ of GUA, DMB, and VL are -0.57, -0.44, and -0.25, respectively.

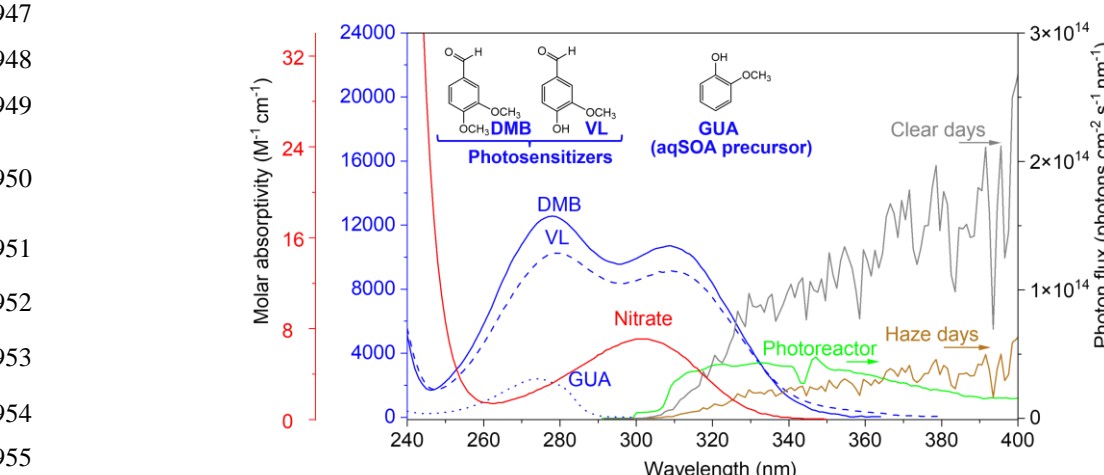

**Figure 1.** The base-10 molar absorptivities (M⁻¹ cm⁻¹) of 3,4-dimethoxybenzaldehyde (DMB, blue solid line), vanillin (VL, blue dashed line), guaiacol (GUA, blue dotted line), and nitrate (red solid line). The green line is the photon flux in the aqueous photoreactor. The gray and brown lines are the photon fluxes on typical clear and haze days, respectively, in Beijing, China (Mabato et al., 2022). The top of the figure also shows the structures of DMB, VL, and GUA.

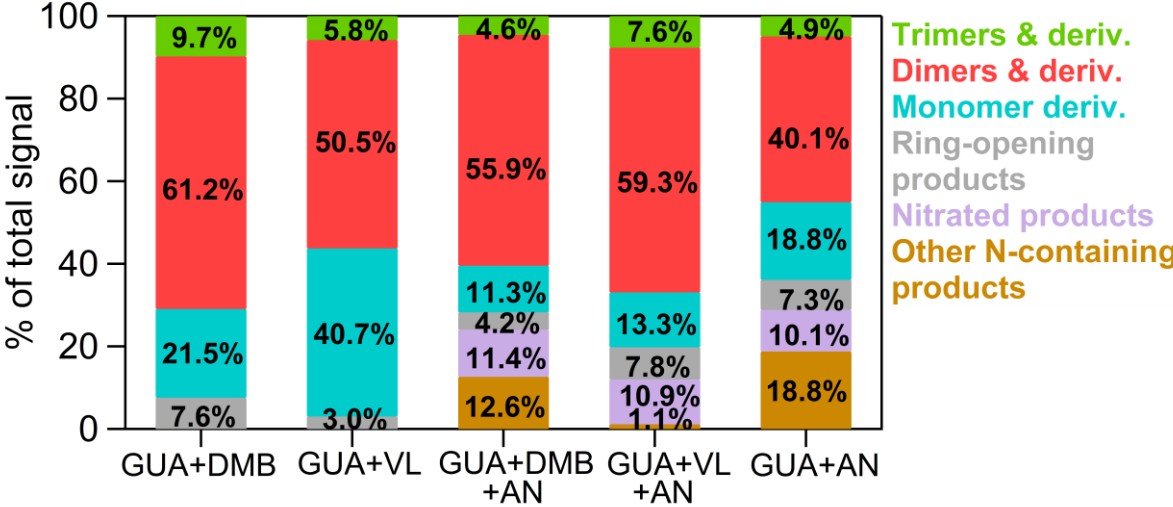

**Figure 2.** Signal-weighted distributions of aqSOA from GUA+DMB, GUA+VL, GUA+DMB+AN, GUA+VL+AN, and GUA+AN. These product distributions were calculated from combined UHPLC-HESI-Orbitrap-MS data obtained in positive (POS) and negative (NEG) ion modes. The values indicate the contribution of different product classifications to the total signals for each reaction condition.

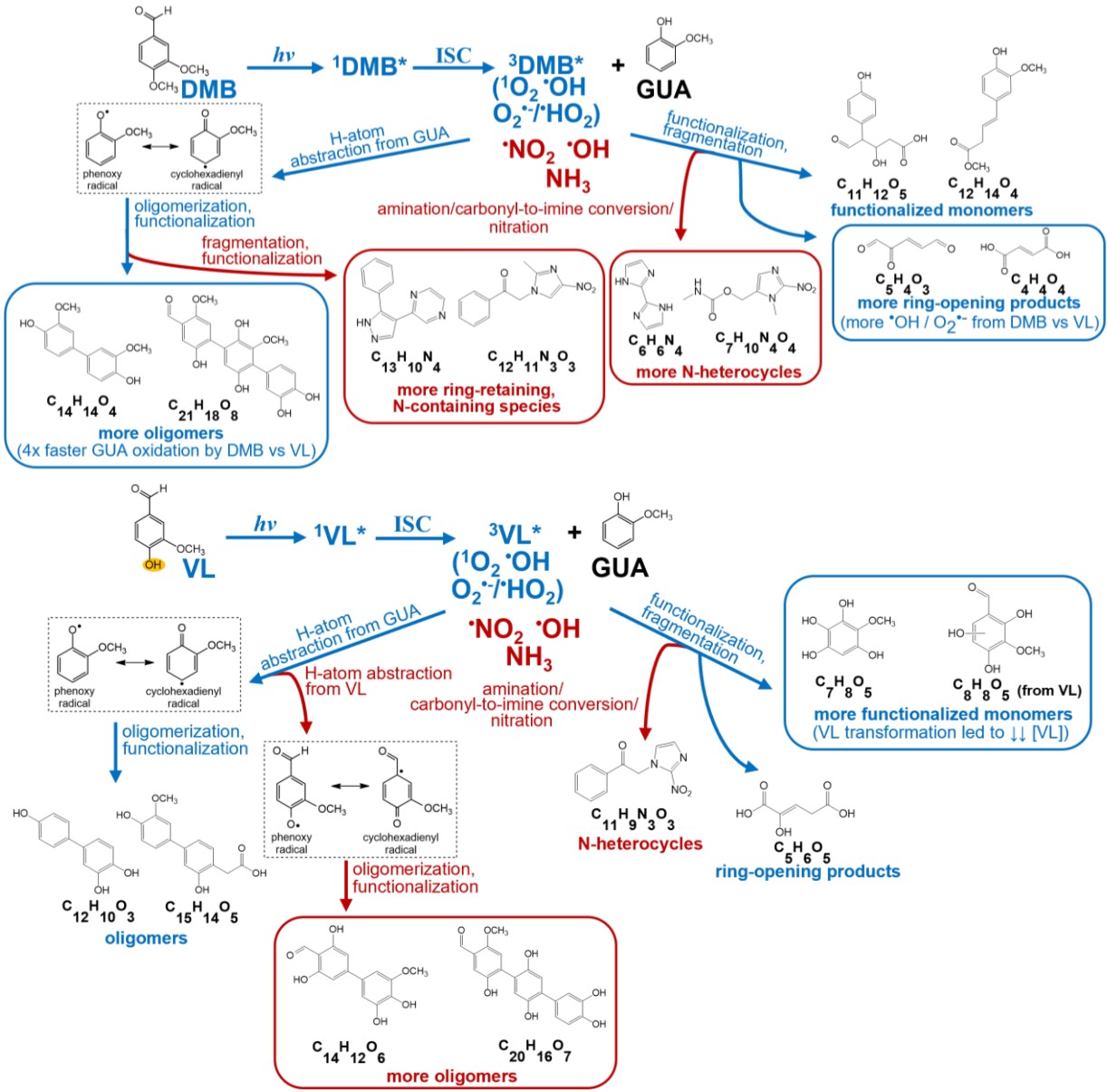

**Figure 3.** Summary of the main differences between photosensitized GUA oxidation by $^3$DMB* (top) and $^3$VL* (bottom) in the absence (blue labels and boxes) and presence (red labels and boxes) of ammonium nitrate at pH 4 under air-saturated conditions. Boxed structures indicate product classifications with notable differences. DMB and VL absorb light and are promoted to their singlet excited states ($^1$DMB* and $^1$VL*), which then undergo intersystem crossing (ISC) to form $^3$DMB* and $^3$VL*. Secondary oxidants ($^1$O$_2$, O$_2^{•-}$/$^•$HO$_2$, $^•$OH) can be formed from $^3$DMB* and $^3$VL* upon reactions with O$_2$ and GUA (George et al., 2018; Chen et al., 2020; Misovich et al., 2021; Mabato et al., 2022). The structures shown are examples of the major products (Tables S1 to S4) for different product classifications.

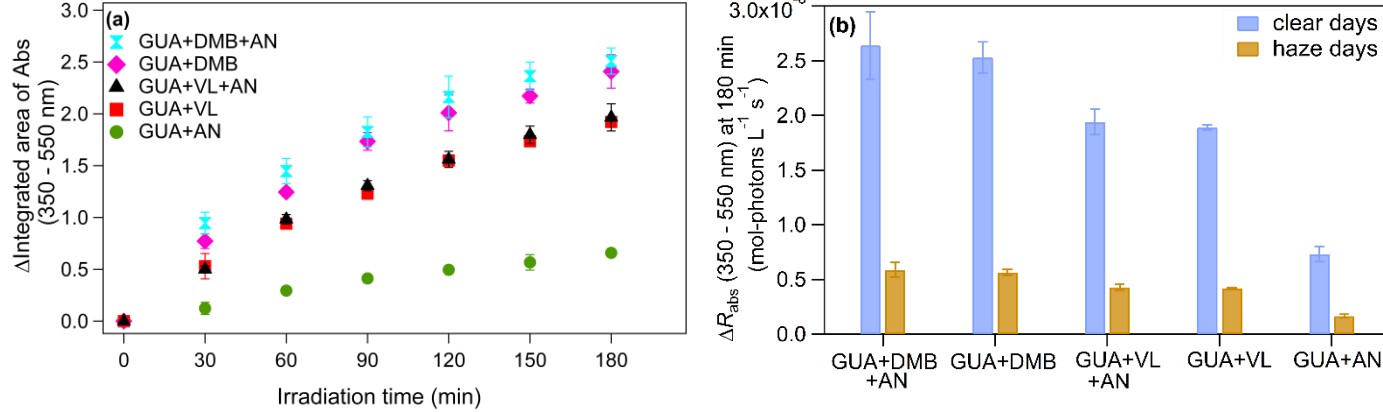

**Figure 4.** (a) Increase in light absorption throughout 180 min of irradiation for all reaction systems studied and (b) Change
in the rate of sunlight absorption ($\Delta R_{abs}$) from 350-550 nm at 180 min during typical clear and haze days in Beijing, China
for aqSOA from GUA+DMB+AN, GUA+DMB, GUA+VL+AN, GUA+VL, and GUA+AN. Error bars represent one
standard deviation of triplicate experiments.