# Peer review of "Comparison of aqueous SOA product distributions from guaiacol oxidation by non-phenolic and phenolic methoxybenzaldehydes as photosensitizers in the absence and presence of ammonium nitrate"

_Atmospheric Chemistry and Physics, 2022_

## Author Response (AR1)

Author Response for "Aqueous SOA formation from photosensitized guaiacol oxidation: Comparison between non-phenolic and phenolic methoxybenzaldehydes as photosensitizers in the absence and presence of ammonium nitrate" by Mabato et al.

We thank the Reviewer for their thorough comments. We have revised the manuscript accordingly, and below are our point-by-point responses (in blue) to the comments (in black) and changes to the manuscript (in red). In those changes that begin with line numbers, the original text is also in blue. In addition, please note that the line numbers in the responses correspond to those in the original manuscript.

**Reviewer 1**

This study analyzes the characteristic of the DMB and VL as photosensitizers reacting with GUA for aqSOA formation, including kinetic analysis, product distributions and chemical characteristics, as well as optical properties. Meanwhile, the effect of AN for aqSOA formation was analyzed. In general, the paper is well written and conclusions are convincing in terms of rational and rigorous experimental design and analyses. I just have several minor comments on it.

1. In terms of abundance of products, for GUA+DMB, the abundance of products in the presence of AN is less than that in the absence of AN, however, for GUA+VL, the results are the opposite. Please elaborate.

Response: It is very common to compare the relative abundance of products based on peak areas from mass spectrometry (MS) results (e.g., Lee et al., 2014; Romonosky et al., 2017; Wang et al., 2017; Fleming et al., 2018; Song et al., 2018; Klodt et al., 2019; Ning et al., 2019) to show the relative importance of different types of compounds (K. Wang et al., 2021). However, comparisons of relative abundance among different compounds can be subject to uncertainties as ionization efficiencies in soft ionization, such as ESI, may significantly vary between different compounds (Kebarle, 2000; Schmidt et al., 2006; Leito et al., 2008; Perry et al., 2008; Kruve et al., 2014). **In our previous work (Mabato et al., 2022), we introduced the normalized abundance of products ([P], unitless).** The calculation assumes equal ionization efficiencies of different compounds, which is commonly used for estimation of O:C ratios of SOA (e.g., Bateman et al., 2012; Lin et al., 2012; Laskin et al., 2014; De Haan et al., 2019). **This allows us to have a better comparison with the literature results.** It is therefore emphasized that the normalized abundance of products is a semi-quantitative analysis of the detected products under different experimental conditions, instead of absolute concentrations of them.

Moreover, as stated in the footnote of Table 1, the normalized abundance of products was calculated using only the positive ion mode data as the GUA signal from the negative ion mode was weak and thus may present large uncertainties during normalization. Therefore, products that may not give signals or may have weak signals in the positive ion mode were possibly underestimated in the normalized product abundance. The contrasting trends of the normalized abundance of products for GUA+DMB and GUA+VL in the absence and presence of AN may therefore be related to the differences in the ionization efficiencies of the products formed. Nevertheless, the much higher normalized abundance of products for

GUA+DMB vs. GUA+VL is consistent with the other more quantitative measurements in this work (i.e., faster GUA decay and stronger light absorption by reaction products in GUA+DMB vs. in GUA+VL).

Sect. 2.2 has been revised to give more information about the normalized abundance of products as follows:

**Section 2.2 Calculation of  normalized abundance of products**

Several recent studies have used comparisons of relative abundance of products based on peak areas from mass spectrometry (MS) results (e.g., Lee et al., 2014; Romonosky et al., 2017; Wang et al., 2017; Fleming et al., 2018; Song et al., 2018; Klodt et al., 2019; Ning et al., 2019) to show the relative importance of different types of compounds (K. Wang et al., 2021). However, comparisons of relative abundance among different compounds can be subject to uncertainties as ionization efficiencies in soft ionization, such as ESI, may significantly vary between different compounds (Kebarle, 2000; Schmidt et al., 2006; Leito et al., 2008; Perry et al., 2008; Kruve et al., 2014).  In our previous work (Mabato et al., 2022), we introduced the normalized abundance of products ([P], unitless) (Eq. 2) as  a semi-quantitative analysis that gives an overview of how the signal intensities changed under different experimental conditions but not the quantification of the absolute product concentration.  The calculation assumes equal ionization efficiencies of different compounds, which is commonly used to estimate O:C ratios of SOA (Bateman et al., 2012; Lin et al., 2012; Laskin et al., 2014; De Haan et al., 2019) :

$$[\mathrm{P}] = \frac{A_{P,t}}{A_{GUA,t}} \cdot \frac{[\mathrm{GUA}]_t}{[\mathrm{GUA}]_0} \qquad\qquad (\text{Eq. 2})$$

where $A_{P,t}$ and $A_{GUA,t}$ are the extracted ion chromatogram (EIC) peak areas of the product P and GUA from UHPLC-HESI-Orbitrap-MS analyses at time $t$, respectively; $[\mathrm{GUA}]_t$ and $[\mathrm{GUA}]_0$ are the GUA concentrations (µM) determined using UHPLC-PDA at time $t$ and 0, respectively. Note that the normalized abundance of products has intrinsic uncertainties due to the variability in ionization efficiencies for various compounds. Moreover, it should be noted that the normalized abundance of products was calculated using only the positive ion mode data as the GUA signal from the negative ion mode was weak and thus may present large uncertainties during normalization. Therefore, products that may not give signals or may have weak signals in the positive ion mode were possibly underestimated in the normalized product abundance. Nevertheless, it enables the comparison of MS results among different experiments. As demonstrated in our previous work (Mabato et al., 2022) and the current study, a higher normalized abundance of products generally correlates with higher efficiency of oxidation. The reported uncertainties were propagated from the changes in [GUA] measured using UHPLC-PDA and the MS signal intensities.

2. For Fig.4, why choose an absorbance wavelength of 180 min for the study? What is the change of absorbance during the whole reaction process, and is the effect of AN obvious on the change of absorbance?

Response: We apologize for the confusion. In Figure 4, the absorbance enhancement was based on the integrated area of absorbance from 350 to 550 nm, and 180 min refers to the total irradiation time. An irradiation time of 180 min was chosen for this study as it was sufficient to show the differences in the extent of the reaction of GUA among the reaction systems studied. Moreover, the same irradiation time was applied to all experiments as we were trying to evaluate the product distributions after a certain time of photosensitization. For reaction systems with precursors of different reactivities, chemical analysis at a fixed reaction time may be looking at different generations of products of each precursor, as Yu et al. (2014) reported. Measuring the product distribution at a fixed time might have missed the information on what/how many products are formed at the similar amounts of precursors reacted. The situation could be even more complicated if different precursors had major differences in pathways and dominant intermediates. However, comparing the product distributions after a certain time of light exposure, as is the case for this study, is useful to evaluate what products would form after a certain time of photosensitization. This information has been added to the discussion of product distributions as follows:

Lines 194-197: The products detected using UHPLC-HESI-Orbitrap-MS were used to  characterize the aqSOA formed in this work. The signal-weighted distributions of aqSOA calculated from combined positive (POS) and negative (NEG) ion modes MS results are summarized in Figure 2. The signal-weighted distributions calculated separately from POS and NEG ion modes MS results are available in Figures S2 and S3. It should be noted that in this work, the product distributions for all experiments were based on the same irradiation time of 180 min. An irradiation time of 180 min was chosen as it was sufficient to show the differences in the extent of reaction of GUA among the reaction systems studied. For reaction systems with precursors of different reactivities, chemical analysis at a fixed reaction time may be looking at different generations of products of each precursor, as Yu et al. (2014) reported. Measuring the product distribution at a fixed time might have missed the information on what/how many products are formed at the similar amounts of precursors reacted. The situation could be even more complicated if different precursors had major differences in pathways and dominant intermediates. However, comparing the product distributions after a certain time of light exposure, as is the case for this study, is useful to evaluate what products would form after a certain time of photosensitization.

For all experiments in this work, we measured the absorbance every 30 min from 0 to 180 min and observed an increase in visible light absorption from 350 to 550 nm. As mentioned in lines 353-358, the effect of AN was not evident on the absorbance enhancement. For GUA+DMB+AN, the N-containing products may have offset the decrease in oligomers to maintain the absorbance enhancement noted from GUA+DMB. For GUA+VL+AN, the decrease in monomers may have counteracted the increased oligomers and the generated N-containing products.

The absorbance enhancement from 0 to 180 min for all reaction systems studied have been added to Figure 4(a). Based on suggestions from Reviewer 4, the original Figure 4 (now Figure 4b) has also been replaced with the change in the rate of sunlight absorption ($\Delta R_{abs}$) from 350-550 nm at 180 min during typical clear and haze days in Beijing, China. Moreover, the absorption spectra after 180 min of irradiation for each solution have been added to the supplement (Figure S7) based on suggestions from Reviewer 4.

The updated Figure 4 and added Figure S7 are shown below:

[Figure]

**Figure 4.** (a) Increase in light absorption throughout 180 min of irradiation for all reaction systems studied and (b) Change in the rate of sunlight absorption ($\Delta R_{abs}$) from 350-550 nm at 180 min during typical clear and haze days in Beijing, China for aqSOA from GUA+DMB+AN, GUA+DMB, GUA+VL+AN, GUA+VL, and GUA+AN.  Error bars represent one standard deviation of triplicate experiments.

[Figure]

**Figure S7**. UV-Vis absorption spectra of GUA+DMB+AN, GUA+DMB, GUA+VL+AN, GUA+VL, and GUA+AN after 180 min of irradiation. The inset is the expanded view from 350 to 550 nm.

The corresponding revisions in the text are as follows:

Lines 253-260: The increase in light absorption throughout 180 min of irradiation and the change in the rate of sunlight absorption ($\Delta R_{abs}$) (Jiang et al., 2021) from 350 to 550 nm at 180 min during typical clear and haze days in Beijing, China for all the reaction systems studied are provided in Figure 4. Figure S7 shows the absorption spectra after 180 min of irradiation for each reaction system studied. In this work, the absorbance enhancement of GUA+DMB and GUA+VL (Fig. 4a) could be due to  oligomers and functionalized monomers, which are the highest contributors to the product signals. Identifying the chromophores responsible for the absorbance enhancement may be beneficial in understanding the impact of aqSOA on the Earth's radiative balance and determining the reactions that affect light absorption by aqSOA (Mabato et al., 2022). However, the detected products did not exhibit distinct peaks in the UHPLC-PDA chromatograms, likely due to the concentration of the chromophores being below the detection limit of PDA. Nevertheless, the higher absorbance enhancement and $\Delta R_{abs}$ for GUA+DMB than GUA+VL was  probably due to the higher contribution and normalized abundance (by ~6 times) of oligomers in the former.

Line 353: The presence of AN also did not appreciably affect the absorbance enhancement and $\Delta R_{abs}$ for both GUA+DMB+AN and GUA+VL+AN (Fig. 4).

3. (a) Why the entire reaction time of this study was 180 minutes, and (b) did the precursors get reacted completely?

Response: (a) As mentioned in our response to question #2, an irradiation time of 180 min was chosen for this work as it was sufficient to show the differences in the extent of reaction of GUA among the reaction systems studied. Moreover, the same irradiation time was applied to all experiments as we were trying to evaluate the product distributions after a certain time of photosensitization. To clarify this in the text, the discussion of product distributions has been amended, as shown in the response to question #2.

(b) No, GUA and the photosensitizers were not fully consumed within the 180 min irradiation. The estimated fraction of GUA and the photosensitizers that remained after irradiation were as follows:

| Reaction conditions | Estimated fraction of GUA and the photosensitizers that remained after 180 min of irradiation |
|---|---|
| GUA+AN | 84% GUA |
| GUA+DMB | 39% GUA and 89% DMB |
| GUA+DMB+AN | 40% GUA and 89% DMB |
| GUA+VL | 79% GUA and 46% VL |
| GUA+VL+AN | 78% GUA and 47% VL |

**References for responses to Reviewer 1:**

[revised manuscript text omitted]

Author Response for "Aqueous SOA formation from photosensitized guaiacol oxidation: Comparison between non-phenolic and phenolic methoxybenzaldehydes as photosensitizers in the absence and presence of ammonium nitrate" by Mabato et al.

We thank the Reviewer for their thorough comments. We have revised the manuscript accordingly, and below are our point-by-point responses (in blue) to the comments (in black) and changes to the manuscript (in red). In those changes that begin with line numbers, the original text is also in blue. In addition, please note that the line numbers in the responses correspond to those in the original manuscript.

**Reviewer 2**

This paper systematically investigated the physicochemical properties of aqueous SOA formed from the photosensitized guaiacol oxidation by using DMB and VL as photosensitizers in the presence and absence of AN. In general, this paper is well written, readable and logical. Before accepted for publication, some revisions should be made. The specific comments are listed as follows:

1. Why did not show aqSOA yields in this work? In my opinion, these data are very useful for readers to understand the importance of these oxidation processes. When these data were described in the paper, the comparison between these date and other similarly reported results should be made.

Response: Since this work mainly focused on the analyses of the reaction products and product distribution, we did not perform measurements of aqSOA mass yields. As similar aqSOA mass yields have been reported from the photosensitized oxidation of syringol (another non-carbonyl phenol) by $^3$DMB* (111%) and $^3$VL* (114%) (Smith et al., 2014, 2016) (mentioned in lines 57-59), we assumed that aqSOA mass yields from guaiacol photo-oxidation by $^3$DMB* and $^3$VL* would also be comparable.

2. As mentioned in section 2.1, the samples were collected every 30 min for 180 min for offline analyses. Therefore, authors can provide more information about the changes of signal-weighted distributions and visible light absorption of aqSOA formed under different conditions during the whole reaction processes. In addition, the concentration changes of small organic acids during the whole reaction processes should be also supplemented.

Response: Thank you for pointing this out. We apologize for the confusion regarding the samples subjected to offline analyses. In this work, the samples collected every 30 min were analyzed using only UHPLC-PDA and UV-Vis spectrophotometry for GUA and photosensitizers decay kinetics and absorbance changes, respectively. The absorbance enhancement from 0 to 180 min for all reaction systems studied have been added to Figure 4(a). Based on suggestions from Reviewer 4, the original Figure 4 (now Figure 4b) has also been replaced with the change in the rate of sunlight absorption ($\Delta R_{abs}$) from 350-550 nm at 180 min during typical clear and haze days in Beijing, China. Moreover, the absorption spectra after 180 min of irradiation for each solution have been added to the supplement (Figure S7) based on suggestions from Reviewer 4.

The updated Figure 4 and added Figure S7 are shown below:

[Figure]

**Figure 4.** (a) Increase in light absorption throughout 180 min of irradiation for all reaction systems studied and (b) Change in the rate of sunlight absorption ($\Delta R_{abs}$) from 350-550 nm at 180 min during typical clear and haze days in Beijing, China for aqSOA from GUA+DMB+AN, GUA+DMB, GUA+VL+AN, GUA+VL, and GUA+AN.  Error bars represent one standard deviation of triplicate experiments.

[Figure]

**Figure S7**. UV-Vis absorption spectra of GUA+DMB+AN, GUA+DMB, GUA+VL+AN, GUA+VL, and GUA+AN after 180 min of irradiation. The inset is the expanded view from 350 to 550 nm.

The corresponding revisions in the text are as follows:

Lines 253-260: The increase in light absorption throughout 180 min of irradiation and the change in the rate of sunlight absorption ($\Delta R_{abs}$) (Jiang et al., 2021) from 350 to 550 nm at 180 min during typical clear and haze days in Beijing, China for all the reaction systems studied are provided in Figure 4. Figure S7 shows the absorption spectra after 180 min of irradiation for each reaction system studied. In this work, the absorbance enhancement of GUA+DMB and GUA+VL (Fig. 4a) could be due to  oligomers and functionalized monomers, which are the highest contributors to the product signals. Identifying the chromophores responsible for the absorbance enhancement may be beneficial in understanding the impact of aqSOA on the Earth's radiative balance and determining the reactions that affect light absorption by aqSOA (Mabato et al., 2022). However, the detected products did not exhibit distinct peaks in the UHPLC-PDA chromatograms, likely due to the concentration of the chromophores being below the detection limit of PDA. Nevertheless, the higher absorbance enhancement and $\Delta R_{abs}$ for GUA+DMB than GUA+VL was  probably due to the higher contribution and normalized abundance (by ~6 times) of oligomers in the former.

Line 353: The presence of AN also did not appreciably affect the absorbance enhancement and $\Delta R_{abs}$ for both GUA+DMB+AN and GUA+VL+AN (Fig. 4).

Moreover, the detection of products and small organic acids was performed only for the samples collected before and after irradiation (180 min). To clarify these differences, section 2.1 has been revised as follows:

Lines 132-137: Samples were collected every 30 min for 180 min for offline analyses of (1) GUA, DMB, and VL concentrations using ultra-high-performance liquid chromatography with photodiode array detector (UHPLC-PDA) and (2) absorbance measurements using UV-Vis spectrophotometry. Moreover, the samples collected before and after irradiation (180 min) were analyzed for (3) reaction products using UHPLC coupled with heated electrospray ionization Orbitrap mass spectrometry (UHPLC-HESI-Orbitrap-MS) operated in positive and negative ion modes and (3) concentrations of small organic acids using ion chromatography (IC).

Other revisions in the text were as follows:

Line 213: The IC analyses also indicate the formation of small organic acids (e.g., formic acid), which appeared to have higher concentrations in the presence of DMB than in VL (Fig. S4). Although no data is available for the concentration changes (every 30 min) of small organic acids during the reaction, it is likely that an increasing trend would be observed as fragmentation, which leads to the decomposition of initially formed oligomers and the generation of smaller oxygenated products, becomes important at longer irradiation times (Huang et al., 2018). This trend has also been observed in our previous work on the direct photosensitized oxidation of VL (Mabato et al., 2022), as well as other studies on photosensitized oxidation of non-carbonyl phenols and phenolic carbonyls (e.g., Yu et al., 2016; Jiang et al., 2021).

Line 382: The major aqSOA formation processes for GUA+DMB and GUA+VL were oligomerization and functionalization, but functionalization appeared to be more significant in GUA+VL due to VL transformation products. The photochemical evolution of aqSOA from GUA+DMB has been reported by Yu et al. (2016). Similar experiments for aqSOA from GUA+VL should be conducted in the future to better understand photosensitized reactions involving phenolic carbonyl photosensitizers.

3. Please provide the reason why selected the whole reaction times as 180 min.

Response: An irradiation time of 180 min was chosen for this work as it was sufficient to show the differences in the extent of reaction of GUA among the reaction systems studied. Moreover, the same irradiation time was applied to all experiments as we were trying to evaluate the product distributions after a certain time of photosensitization.

This information has been added to the discussion of the product distributions as follows:

Lines 194-197: The products detected using UHPLC-HESI-Orbitrap-MS were used to  characterize the aqSOA formed in this work. The signal-weighted distributions of aqSOA calculated from combined positive (POS) and negative (NEG) ion modes MS results are summarized in Figure 2. The signal-weighted distributions calculated separately from POS and NEG ion modes MS results are available in Figures S2 and S3. It should be noted that in this work, the product distributions for all experiments were based on the same irradiation time of 180 min. An irradiation time of 180 min was chosen as it was sufficient to show the differences in the extent of reaction of GUA among the reaction systems studied. For reaction systems with precursors of different reactivities, chemical analysis at a fixed reaction time may be looking at different generations of products of each precursor, as Yu et al. (2014) reported. Measuring the product distribution at a fixed time might have missed the information on what/how many products are formed at the similar amounts of precursors reacted. The situation could be even more complicated if different precursors had major differences in pathways and dominant intermediates. However, comparing the product distributions after a certain time of light exposure, as is the case for this study, is useful to evaluate what products would form after a certain time of photosensitization.

4. There are still some language mistakes, please carefully check.

Response: Thank you for pointing this out. We have carefully checked the text for language mistakes.

**References for responses to Reviewer 2:**

Huang, D. D., Zhang, Q., Cheung, H. H. Y., Yu, L., Zhou, S., Anastasio, C., Smith, J. D., and Chan, C. K.: Formation and evolution of aqSOA from aqueous-phase reactions of phenolic carbonyls: comparison between ammonium sulfate and ammonium nitrate solutions, Environ. Sci. Technol., 52, 9215–9224, https://doi.org/10.1021/acs.est.8b03441, 2018.

Jiang, W., Misovich, M. V., Hettiyadura, A. P. S., Laskin, A., McFall, A. S., Anastasio, C., and Zhang, Q.: Photosensitized reactions of a phenolic carbonyl from wood combustion in the aqueous phase—chemical evolution and light absorption properties of aqSOA, Environ. Sci. Technol., 55, 5199–5211, https://doi.org/10.1021/acs.est.0c07581, 2021.

Mabato, B. R. G., Lyu, Y., Ji, Y., Li, Y. J., Huang, D. D., Li, X., Nah, T., Lam, C. H., and Chan, C. K.: Aqueous secondary organic aerosol formation from the direct photosensitized oxidation of vanillin in the absence and presence of ammonium nitrate, Atmos. Chem. Phys., 22, 273–293, https://doi.org/10.5194/acp-22-273-2022, 2022.

Smith, J. D., Sio, V., Yu, L., Zhang, Q., and Anastasio, C.: Secondary organic aerosol production from aqueous reactions of atmospheric phenols with an organic triplet excited state, Environ. Sci. Technol., 48, 1049–1057, https://doi.org/10.1021/es4045715, 2014.

Smith, J. D., Kinney, H., and Anastasio, C.: Phenolic carbonyls undergo rapid aqueous photodegradation to form low-volatility, light-absorbing products, Atmos. Environ., 126, 36–44, https://doi.org/10.1016/j.atmosenv.2015.11.035, 2016.

Yu, L., Smith, J., Laskin, A., Anastasio, C., Laskin, J., and Zhang, Q.: Chemical characterization of SOA formed from aqueous-phase reactions of phenols with the triplet excited state of carbonyl and hydroxyl radical, Atmos. Chem. Phys., 14, 13801–13816, https://doi.org/10.5194/acp-14-13801-2014, 2014.

Yu, L., Smith, J., Laskin, A., George, K. M., Anastasio, C., Laskin, J., Dillner, A. M., and Zhang, Q.: Molecular transformations of phenolic SOA during photochemical aging in the aqueous phase: competition among oligomerization, functionalization, and fragmentation, Atmos. Chem. Phys., 16, 4511–4527, https://doi.org/10.5194/acp-16-4511-2016.

Author Response for "Aqueous SOA formation from photosensitized guaiacol oxidation: Comparison between non-phenolic and phenolic methoxybenzaldehydes as photosensitizers in the absence and presence of ammonium nitrate" by Mabato et al.

We thank the Reviewer for their thorough comments. We have revised the manuscript accordingly, and below are our point-by-point responses (in blue) to the comments (in black) and changes to the manuscript (in red). In those changes that begin with line numbers, the original text is also in blue. In addition, please note that the line numbers in the responses correspond to those in the original manuscript.

**Reviewer 3**

This manuscript describes a comparative study of the photosensitization by phenolic and non-phenolic methoxybenzaldehydes in reactions of guaiacol (another phenolic compound, but without an aldehyde functional group), with and without the presence of ammonium nitrate salts. The experiments were conducted in bulk aqueous phase samples in a solar simulator.

The combination of photosensitizing reactions of methoxybenzaldehyde species with ammonium nitrate photochemistry in a series of experiments is especially interesting. The primary conclusion is that the non-phenolic species DNB is approximately 4 times more effective as a photosensitizer than the phenolic species vanillin, and produces slightly more brown carbon. The manuscript includes a great number of qualitative comparisons, but the authors highlight the most important ones in the abstract and conclusion. It will be of interest to atmospheric scientists studying mechanisms of formation of brown carbon and aqueous secondary organic aerosol.

(a) My first concern is that the authors may have oversimplified the complex task of comparing the photosensitizing abilities of VL and DNB, when VL is reacting away at ~20x the rate of DNB (a factor of 8 x 2.4). The reactivity of VL is so great that it successfully competes with GUA in the reaction with the VL triplet (3VL*), reacting with it 24% of the time over the course of the reaction even though the VL concentration is 10x less than GUA. (I estimated this reaction fraction from the stated 2.4x faster decay rate of VL times the VL / GUA concentration ratio of 0.01mM/0.1mM, resulting in a relative loss rate for VL of 0.24 if GUA loss rate = 1.) If one takes into account 3VL* reactions with both VL and GUA, DNB would be at most only 3 times faster than VL at promoting photosensitization reactions in general. A more nuanced kinetics analysis would thus be helpful for GUA + VL and GUA + VL + AN reactions. Furthermore, it could allow some qualitative statements in the paper, such as those in line 204 and 207, to become quantitative: (b) when integrated over the full course of the reaction, what is the impact of the loss of the reactant VL on the total amount of products generated?

Response: (a) Thank you for pointing this out. We would like to emphasize that the kinetics analysis and apparent quantum efficiency of GUA photodegradation suggested faster GUA oxidation in GUA+DMB vs. GUA+VL, which we attributed to two reasons: (1) DMB having a stronger photosensitizing ability than VL based on its higher quantum yield of $^3C^*$ formation and longer lifetime of $^3DMB^*$ compared to $^3VL^*$ (Felber et al., 2021) and (2) VL being highly reactive towards oxidation as it is also a phenolic compound, similar to GUA. However, these trends **do not indicate that DMB is 4 times more effective as a photosensitizer** compared to VL. Also, during GUA oxidation, the calculated decay rate constant of VL was only 4.6 times higher (not ~20 times) than that of DMB (VL decay rate constant in GUA+VL: 3.6 min$^{-1}$/s$^{-1}$ vs. DMB decay rate constant in GUA+DMB: 0.78 min$^{-1}$/s$^{-1}$; note that the decay rate constants were corrected for internal light screening due to DMB and VL light absorption, and normalized to the experimental photon flux).

The Reviewer is correct that comparing photosensitizing abilities is a complex task. However, a detailed quantitative analysis of the photosensitizing abilities which necessitates more experiments, e.g., determining the intersystem crossing quantum yield for VL (Smith et al., 2014) or using time-resolved absorption spectroscopy (Felber et al., 2021) is beyond the scope of the paper. Nonetheless, the estimated decay rate constants and apparent quantum efficiency of GUA photodegradation indicate that GUA oxidation in GUA+DMB was overall more efficient than in GUA+VL. These measurements can be useful for comparison with GUA oxidation by other oxidants or photosensitizers. Our kinetic analysis focused on the decay rate constants of the aqSOA precursor (GUA) and the photosensitizers (DMB and VL) during photosensitization under the same experimental conditions (same aqSOA precursor and concentration, same photosensitizer concentration, and same lamp photon flux). The effects of other factors (e.g., intersystem crossing efficiency) on the rate constants were not examined. Hence, lacking rate constants that are either universal or specific for a number of experimental conditions in the literature, we hope that these parameters obtained in our study can provide first-order estimates for modeling since the experimental conditions (GUA+DMB, GUA+VL, GUA+DMB+AN, GUA+VL+AN, and GUA+AN based on concentrations relevant to cloud and fog conditions at pH 4 in air) are atmospherically relevant. Explicit kinetic studies (e.g., Smith et al., 2014, 2015) that measure second-order rate constants should be conducted in the future to extend the applicability of the kinetic parameters to other conditions.

The following sentences have been added at the end of Sect. 3.1.1 to clarify these:

Line 192: It should be noted that the differences in the GUA decay rate constants among different reaction systems are not quantitatively equivalent to photosensitizing efficiencies, and a detailed quantitative analysis of which is beyond the scope of this study. Nonetheless, these results suggested that GUA oxidation in GUA+DMB was overall more efficient than in GUA+VL. Our kinetic analysis focused on the decay rate constants of the aqSOA precursor (GUA) and the photosensitizers (DMB and VL) during photosensitization under the same experimental conditions (same aqSOA precursor and concentration, same photosensitizer concentration, and same lamp photon flux). The effects of other factors (e.g., intersystem crossing efficiency) on the rate constants were not examined. Explicit kinetic studies (e.g., Smith et al., 2014, 2015) that measure second-order rate constants should be conducted in the future to extend the applicability of the kinetic parameters to other conditions.

Moreover, it should be noted that in this work, we mainly focused on the analyses of the reaction products and product distributions.

The title of the paper has been revised to 'Comparison of aqueous SOA product distributions from guaiacol oxidation by non-phenolic and phenolic methoxybenzaldehydes as photosensitizers in the absence and presence of ammonium nitrate' to better reflect the focus of the work.

Furthermore, the following sentences have been added to the text to clarify the focus of this work:

Line 96: The precursor and photosensitizer decay kinetics, detected products, and absorbance enhancement were used to characterize the reactions. However, it should be noted that we mainly focused on the analyses of the reaction products and product distribution.

(b) We have given a lot of thought to this comment. We assumed that the question pertains to the normalized abundance of products. The calculation of the normalized abundance of products involves the absolute GUA concentration changes measured before and after irradiation using UHPLC-PDA, which is related to the loss of the photosensitizers. Therefore, the estimated normalized abundance of products already covers the loss of the photosensitizers. The following are mentioned in lines 207-208: "In addition, the normalized product abundance for GUA+DMB was ~4 times higher than that for GUA+VL (Table 1), further suggesting more efficient photosensitized GUA oxidation by $^3$DMB* than by $^3$VL*." However, we would like to emphasize that the normalized abundance of products in this work represents a semi-quantitative analysis of products in different experiments rather than absolute concentration of products.

Other comments:

1. Line 95: How are products counted if they appear in both positive and negative modes of ionization?

Response: For the signal-weighted distributions in Figure 2, the peak area of each product (whether it appeared in either positive or negative ion mode only, or it appeared in both ion modes) was normalized to the total signal areas summed from positive and negative ion modes. For reference, the signal-weighted distributions calculated separately from positive and negative ion modes are provided in Figures S2 and S3 (formerly Figures S1 and S2).

2. Line 335: What could highly oxidized species decompose into, that would not be detected and therefore not contribute to the measured O/C ratio? Is this statement alluding to CO2 production?

Response: We apologize for the confusion. Lines 334-335 are based on an earlier work on photo-oxidation of phenolic carbonyls in ammonium nitrate (AN) and ammonium sulfate solutions using a time-of-flight aerosol mass spectrometer (AMS) and therefore involved the aerosolization of samples before analysis (Huang et al., 2018). In that study, highly oxidized species such as small carboxylic acids with O:C =1 (e.g., acetic acid) possibly formed but may have evaporated during aerosolization as they are too volatile. As a result, these species cannot be detected by the AMS. Solutions with AN had a higher concentration of organic acids (without atomization) in the aqueous samples compared to ammonium sulfate solutions, ascribable to nitrate photolytic products promoting the reactions. In brief, the study by Huang et al. (2018) suggested that AN promoted the formation of oxygenated and oxidized products.

As mentioned in lines 340-342, AN generally increased the average $OS_c$ values for both GUA+DMB and GUA+VL, indicating the formation of more oxidized products, similar to the findings by Huang et al. (2018). In this work, AN also possibly promoted the formation of oxygenated products. The lower average O:C for GUA+DMB+AN and GUA+VL+AN than those in the absence of AN could be due to the formation of N-heterocycles, altering the elemental ratios.

The corresponding revisions in the text are as follows:

Line 334: The ⟨O:C⟩ for GUA+DMB+AN and GUA+VL+AN were lower than those in the absence of AN (Table 1),  possibly due to the formation of N-heterocycles, altering the elemental ratios.

Line 340: Nonetheless, AN generally increased the ⟨OS_C⟩ for both GUA+DMB and GUA+VL, with a more noticeable increase for the former, suggesting more oxidized products. Similarly, in a previous work, the more oxygenated and oxidized aqSOA from the photo-oxidation of phenolic carbonyls in AN solutions than in ammonium sulfate solutions has been ascribed to nitrate photolytic products promoting the reactions (Huang et al., 2018).

3. Figure 3: at the top right, C12 and C11 products are referred to as functionalized monomers. How is this different from a ring-opened dimer? How exactly do the authors distinguish functionalization from dimerization?

Response: The dimers in this work have minimum carbon atoms of 12 for 2 benzene rings, and considering demethylation of both -OCH$_3$ groups of typical guaiacol dimer (C$_{14}$H$_{14}$O$_4$; #1 to C$_{12}$H$_{10}$O$_4$; #17; Table S1). The C12 and C11 products in Figure 3 were referred to as functionalized monomers and not ring-opened dimers as the substituents (other than -OH and -OCH$_3$) in these products have carbon atoms <6 which were not enough to form another aromatic ring. These substituents were more likely to be highly oxygenated small species (carbon atoms <6) from oxidation and fragmentation reactions which have also been reported in previous studies on similar reaction systems (e.g., Yu et al., 2014, 2016).

Functionalization involves the addition of polar oxygenated functional groups (e.g., hydroxyl, carbonyl, carboxyl etc.) as well as highly oxygenated small species (carbon atoms < 6) from oxidation and fragmentation reactions to the aromatic ring. Dimerization is characterized by two covalently bound units of the aromatic compounds studied.

4. Figure 4: In this graph, does 1 or zero = no change in integrated absorbance? In other words, is it normalized somehow?

Response: In the original Figure 4, a value of zero means that at 180 min, there was no change in the integrated area of absorbance from 350 to 550 nm compared to 0 min (before irradiation). The integrated area of absorbance from 350 to 550 nm at 180 min was normalized by subtracting the corresponding values at 0 min (before irradiation). For all experiments in this work, we measured the absorbance every 30 min from 0 to 180 min and observed an increase in visible light absorption from 350 to 550 nm.

The absorbance enhancement from 0 to 180 min for all reaction systems studied have been added to Figure 4(a). Based on suggestions from Reviewer 4, the original Figure 4 (now Figure 4b) has also been replaced with the change in the rate of sunlight absorption ($\Delta R_{abs}$) from 350-550 nm at 180 min during typical clear and haze days in Beijing, China. Moreover, the absorption spectra after 180 min of irradiation for each reaction system have been added to the supplement (Figure S7) based on suggestions from Reviewer 4.

The updated Figure 4 and added Figure S7 are shown below:

[Figure]

**Figure 4.** (a) Increase in light absorption throughout 180 min of irradiation for all reaction systems studied and (b) Change in the rate of sunlight absorption ($\Delta R_{abs}$) from 350-550 nm at 180 min during typical clear and haze days in Beijing, China for aqSOA from GUA+DMB+AN, GUA+DMB, GUA+VL+AN, GUA+VL, and GUA+AN.  Error bars represent one standard deviation of triplicate experiments.

[Figure]

**Figure S7**. UV-Vis absorption spectra of GUA+DMB+AN, GUA+DMB, GUA+VL+AN, GUA+VL, and GUA+AN after 180 min of irradiation. The inset is the expanded view from 350 to 550 nm. The corresponding revisions in the text are as follows:

Lines 253-260: The increase in light absorption throughout 180 min of irradiation and the change in the rate of sunlight absorption ($\Delta R_{abs}$) (Jiang et al., 2021) from 350 to 550 nm at 180 min during typical clear and haze days in Beijing, China for all the reaction systems studied are provided in Figure 4. Figure S7 shows the absorption spectra after 180 min of irradiation for each reaction system studied. In this work, the absorbance enhancement of GUA+DMB and GUA+VL (Fig. 4a) could be due to  oligomers and functionalized monomers, which are the highest contributors to the product signals. Identifying the chromophores responsible for the absorbance enhancement may be beneficial in understanding the impact of aqSOA on the Earth's radiative balance and determining the reactions that affect light absorption by aqSOA (Mabato et al., 2022). However, the detected products did not exhibit distinct peaks in the UHPLC-PDA chromatograms, likely due to the concentration of the chromophores being below the detection limit of PDA. Nevertheless, the higher absorbance enhancement and $\Delta R_{abs}$ for GUA+DMB than GUA+VL was  probably due to the higher contribution and normalized abundance (by ~6 times) of oligomers in the former.

Line 353: The presence of AN also did not appreciably affect the absorbance enhancement and $\Delta R_{abs}$ for both GUA+DMB+AN and GUA+VL+AN (Fig. 4).

Technical corrections

1. Line (1)94: "represent" should be "characterize"

Response: Thank you for the correction. "represent" has been replaced by "characterize"

2. Line 328: should this say "likely has a furanone group"? Otherwise, how do the authors know this is the correct structure from the many possibilities?

Response: The Reviewer is correct. The text has been revised accordingly.

**References for responses to Reviewer 3:**

Felber, T., Schaefer, T., He, L., and Herrmann, H.: Aromatic carbonyl and nitro compounds as photosensitizers and their photophysical properties in the tropospheric aqueous phase, J. Phys. Chem. A, 125, 5078–5095, https://doi.org/10.1021/acs.jpca.1c03503, 2021.

Huang, D. D., Zhang, Q., Cheung, H. H. Y., Yu, L., Zhou, S., Anastasio, C., Smith, J. D., and Chan, C. K.: Formation and evolution of aqSOA from aqueous-phase reactions of phenolic carbonyls: comparison between ammonium sulfate and ammonium nitrate solutions, Environ. Sci. Technol., 52, 9215–9224, https://doi.org/10.1021/acs.est.8b03441, 2018.

Jiang, W., Misovich, M. V., Hettiyadura, A. P. S., Laskin, A., McFall, A. S., Anastasio, C., and Zhang, Q.: Photosensitized reactions of a phenolic carbonyl from wood combustion in the aqueous phase—chemical evolution and light absorption properties of aqSOA, Environ. Sci. Technol., 55, 5199–5211, https://doi.org/10.1021/acs.est.0c07581, 2021.

Mabato, B. R. G., Lyu, Y., Ji, Y., Li, Y. J., Huang, D. D., Li, X., Nah, T., Lam, C. H., and Chan, C. K.: Aqueous secondary organic aerosol formation from the direct photosensitized oxidation of vanillin in the absence and presence of ammonium nitrate, Atmos. Chem. Phys., 22, 273–293, https://doi.org/10.5194/acp-22-273-2022, 2022.

Smith, J. D., Sio, V., Yu, L., Zhang, Q., and Anastasio, C.: Secondary organic aerosol production from aqueous reactions of atmospheric phenols with an organic triplet excited state, Environ. Sci. Technol., 48, 1049–1057, https://doi.org/10.1021/es4045715, 2014.

Smith, J. D., Kinney, H., and Anastasio, C.: Aqueous benzene-diols react with an organic triplet excited state and hydroxyl radical to form secondary organic aerosol, Phys. Chem. Chem. Phys., 17, 10227–10237, https://doi.org/10.1039/C4CP06095D, 2015.

Yu, L., Smith, J., Laskin, A., Anastasio, C., Laskin, J., and Zhang, Q.: Chemical characterization of SOA formed from aqueous-phase reactions of phenols with the triplet excited state of carbonyl and hydroxyl radical, Atmos. Chem. Phys., 14, 13801–13816, https://doi.org/10.5194/acp-14-13801-2014, 2014.

Yu, L., Smith, J., Laskin, A., George, K. M., Anastasio, C., Laskin, J., Dillner, A. M., and Zhang, Q.: Molecular transformations of phenolic SOA during photochemical aging in the aqueous phase: competition among oligomerization, functionalization, and fragmentation, Atmos. Chem. Phys., 16, 4511–4527, https://doi.org/10.5194/acp-16-4511-2016.

Author Response for "Aqueous SOA formation from photosensitized guaiacol oxidation: Comparison between non-phenolic and phenolic methoxybenzaldehydes as photosensitizers in the absence and presence of ammonium nitrate" by Mabato et al.

We thank the Reviewer for their thorough comments. We have revised the manuscript accordingly, and below are our point-by-point responses (in blue) to the comments (in black) and changes to the manuscript (in red). In those changes that begin with line numbers, the original text is also in blue. In addition, please note that the line numbers in the responses correspond to those in the original manuscript.

**Reviewer 4**

Mabato and co-authors studied the aqueous photochemical reactions of guaiacol (GUA), a methoxyphenol from wood burning, in the presence of vanillin (VL; a phenolic carbonyl), dimethoxybenzaldehyde (DMB; a non-phenolic carbonyl), and/or ammonium nitrate (AN). They examined photochemistry in five different reaction solutions: (1) GUA + AN, (2) GUA + DMB, (3) GUA + DMB + AN, (4) GUA + VL, and (5) GUA + VL + AN. For each system, they give the kinetics of loss, information about the products formed, and some very cursory information about light absorption from the resulting reaction mixture.

There are a few interesting pieces in the manuscript, most notably the suggested interaction of AN photoproducts with triplet photoproducts, which I wish the authors had explored more. But, otherwise, the research seems to largely repeat ideas and experiments that have been reported previously. I don't see new, interesting questions that are driving the current work. In addition, I see several other important weaknesses, as described below.

Response: Thank you for the comprehensive review. In this work, our focuses were 1) to **compare** aqSOA formation from **photosensitization by non-phenolic and phenolic methoxybenzaldehydes** as photosensitizers using GUA as the dominant aqSOA precursor and 2) to examine the **effects of AN** on these reactions. Biomass burning smoke has been reported to have comparable concentrations of phenolic and non-phenolic carbonyls (Simoneit et al., 1993; Anastasio et al., 1997), and certain phenolic carbonyls including VL, have been shown to be capable of photosensitization, contributing to aqSOA formation (Smith et al., 2016; Mabato et al., 2022). However, compared to DMB (a non-phenolic carbonyl), the most commonly used photosensitizer in related studies, aqSOA formation using phenolic carbonyls as photosensitizers is less understood. DMB and VL were chosen to represent non-phenolic and phenolic methoxybenzaldehydes photosensitizers because (1) their structures differ only by one functional group (–OCH$_3$ for the former and –OH for the latter), (2) they are both abundant in biomass burning emissions, and (3) there is available information on their photophysical properties (e.g., quantum yield of $^3$C* formation and $^3$C* lifetime) (Felber et al., 2021). Moreover, aromatic carbonyls and phenols may coexist with AN in biomass burning aerosols. However, aqSOA formation **studies have investigated either photosensitization or nitrate-mediated photo-oxidation, even though these reactions can occur simultaneously**. In general, **studies** on the **effects of inorganic nitrate on aqSOA formation remain very limited**. We believe that **this work is one of the very few in the literature that examines this complex system. The Reviewer has given us a lot of valuable comments highlighting the complexity of the photosensitization reactions** when even just a photosensitizer and a nonphenolic carbonyl are involved in the reactions, let alone the addition of AN in the system. **It is** understood that **much more work needs to be done to fully unravel the interactions of photosensitization and nitrate photolysis.** We hope this work can **stimulate researchers to look into these important but rarely explored interactions in the photochemistry of aerosols.** Further information on the novelty of this work is given in the response to major comment #1.

**Major Comments**

1. It is not clear what is novel enough about this work that it deserves to be published in ACP. (a) A number of the systems or parts of the manuscript have been reported previously, both by this group and other groups. For example, Mabato et al. (2022) reported results for GUA + VL, while Smith et al. and Yu et al. reported results for GUA + DMB and for another phenol (SYR) with VL. (b) The addition of ammonium nitrate in the manuscript has no significant impact on kinetics or normalized product amounts, but does lead to incorporation of N into the aqSOA, a point made for VL + AN by Mabato et al. (2022) and for general carbonyl + ammonium systems by several past authors. (c) One result is that there is significant repetition of past work.  Some examples: (1) the first two paragraphs on page 8 largely repeat what has been shown in previous work, (2) there's nothing new in Figure 1, as all of these molar absorptivities have been shown by previous groups, and (3) most of the pieces of Figure 3 have been shown in Mabato et al. (2022) or in past work by the Zhang group.

Response: (a) The Reviewer is correct that GUA+DMB and GUA+VL have been explored in earlier studies. Our previous study (Mabato et al., ACP, 2022) focused on aqSOA formation from the direct photosensitized oxidation of VL. GUA+VL in that study was compared with direct GUA photodegradation. We discussed the decay kinetics of both VL and GUA, absorbance changes, and a few detected products **but not product distribution**. Smith et al. (2014) and Yu et al. (2014, 2016) provided valuable information on GUA+DMB, focusing on the decay kinetics of GUA, aqSOA mass yields, product characterization, photochemical evolution of aqSOA, and aqSOA formation pathways. For VL+syringol, SYR (another non-carbonyl phenol), Smith et al. (2016) reported VL and SYR decay kinetics and absorbance changes. **However, as stated in the original manuscript, our previous experiments were performed at a higher concentration of VL (0.1 mM) compared to DMB concentration (0.005 to 0.01 mM) used by Smith et al. (2014) and Yu et al. (2014, 2016).** Moreover, different instruments were used to characterize the products in these earlier works. Thus, direct comparisons between GUA+DMB and GUA+VL based on atmospherically relevant conditions cannot be made using the above studies. **This paper is the first, to the best of our knowledge, to make such comparisons at the same concentration range with the same analytical techniques systematically.**

Moreover, previous aqSOA formation studies involving aromatic carbonyls and non-carbonyl phenols have examined either photosensitization or nitrate-mediated photo-oxidation **separately, even though these reactions can occur simultaneously**. In general, aqSOA formation **studies on multicomponent reaction systems w**here inorganic and organic species are present **remain limited. In the current work,** we attempted to **address this gap by introducing AN, a common aerosol component** that is also abundant in cloud and fog waters, **into the reaction systems studied.**

We would like to emphasize that the novelty of the current work includes: 1) a thorough comparison of aqSOA formation from photosensitization by non-phenolic and phenolic methoxybenzaldehydes as photosensitizers using GUA as the dominant aqSOA precursor based on GUA and photosensitizers decay kinetics, detected products, and absorbance enhancement and 2) the examination of the effects of AN on these reactions. Despite some similar findings with earlier studies, our results offer new insights into the reaction systems examined. For instance, our results demonstrate that **the structural features of photosensitizers affect aqSOA formation** via GUA (non-carbonyl phenol) oxidation. Moreover, we found that **photosensitization by non-phenolic and phenolic methoxybenzaldehydes may be differently influenced by AN photolysis.** For example, compared to GUA+VL, the more extensive fragmentation in GUA+DMB likely resulted in a higher contribution of ring-opening products. These ring-opening species reacted with ammonia resulting in higher contributions of N-heterocycles and N-oligomers and yielding an overall higher contribution of N-containing products in GUA+DMB+AN than in GUA+VL+AN. The increase in oligomers in GUA+VL+AN could be due to the generation of phenoxy radicals from VL via H-atom abstraction by $^{\bullet}OH$ or $^{\bullet}NO_2$ from nitrate photolysis. Furthermore, our results suggest that **photosensitization may promote reactions by nitrate photolysis.** In particular, $^{\bullet}OH$ and $O_2^{\bullet-}$ (secondary oxidants from $^3DMB^*$ or $^3VL^*$) can react with $NO_2^-$ from nitrate photolysis to increase the formation of $^{\bullet}NO_2$ (Pang et al., 2019; Mabato et al., 2022).

(b) The Reviewer is right that the formation of N-heterocycles which indicates the participation of ammonium in the reactions has also been reported in our previous experiments on VL+AN (Mabato et al., 2022) and for smaller carbonyls + ammonium systems in several past works (e.g., De Haan et al., 2009; Galloway et al., 2009; Nozière et al., 2009; Shapiro et al., 2009; Yu et al., 2011; Kampf et al., 2012; Gen et al., 2018; Mabato et al., 2019). **However, our present study also demonstrated that photosensitized aqSOA formation involving non-phenolic and phenolic methoxybenzaldehydes as photosensitizers may be differently influenced by AN photolysis.** In addition, we also found that **photosensitized reactions may promote nitrate-mediated photolytic reactions.** Overall, our conclusions provide a clear path forward for further work on other types of photosensitizers for aqSOA formation, which to date is **limited mainly to DMB, and indicate a potential interplay between photosensitized reactions and AN photolysis, beyond the formation of nitrated compounds.** We also revised the abstract to further highlight the novelty of the work, as shown below:

**Abstract:**

Aromatic carbonyls (e.g., methoxybenzaldehydes), an important class of photosensitizers, are abundant in the atmosphere. Photosensitization and nitrate-mediated photo-oxidation can occur simultaneously, yet studies about their interactions, particularly for aqueous secondary organic aerosol (aqSOA) formation, remain limited. This study compared non-phenolic (3,4-dimethoxybenzaldehyde, DMB) and phenolic (vanillin, VL) methoxybenzaldehydes as photosensitizers for aqSOA formation via guaiacol (GUA) oxidation in the absence and presence of ammonium nitrate (AN) under atmospherically relevant cloud and fog conditions…

(c) We thank the Reviewer for highlighting these three areas that contain a lot of information based on the literature. However, they mainly **provide background for the readers to follow the results and discussions.** We are unsure if they are the best examples for evaluating the novelty of this paper.

(1) The first 2 paragraphs on page 8 compare **the major products, average elemental ratios, and elemental distributions of products** from GUA+DMB and GUA+VL. In addition, we **discussed how this information relates to the differences in the product distributions, which are not available in previous works.**

(2) Aside from showing molar absorptivities, Figure 1 provides the structures of the compounds studied in this work, the photon flux in the photoreactor, and photon fluxes on clear or haze days. **This figure is meant to provide sufficient background information for readers. It is not a "result" figure.**

(3) The related figure by Yu et al. (2014) shows the **main pathways of aqSOA formation via photosensitized oxidation of non-carbonyl phenols by $^3$DMB\* only**. Also, the related figure in our previous work (Mabato et al., 2022) shows the potential aqSOA formation pathways for the direct photosensitized oxidation of VL and VL+AN. **The comparisons about N-containing products in that work pertained to pH effects, not photosensitizers**. In addition, the aqSOA precursor (and photosensitizer) in that study is VL, whereas here, it is GUA. Figure 3 illustrates the **differences between GUA oxidation by $^3$DMB\* and $^3$VL\* in the absence and presence of AN, highlighting the major products detected in this work**. **The comparisons are based on the product distributions which have not been reported for GUA+VL. Moreover, the effects of AN on the product distributions of aqSOA from photosensitized reactions have not been examined previously.**

2. The results aren't quantitative in a way that they could be used to model aqSOA formation from 3VL\*. For example, the rate constants for decay of GUA given in Table 1 are probably specific for the experimental conditions used here. The same is true for the quantum yields given in the text - these are almost certainly a function of GUA concentration. It would be much more useful to measure fundamental quantities (e.g., second-order rate constants for 3VL\* + GUA) that can be used across a wide range of conditions. Can the current set of data be used to determine some fundamental quantities that are widely applicable? If not, how will people use these data to quantitatively understand these reaction systems?

Response: Agreed. Having said that, we believe this is a common limitation of laboratory kinetic studies in complex chemical systems. While there are some studies (e.g., Smith et al., 2015) citing various kinetic parameters on photosensitization based on selected experimental studies, we are unsure if there are kinetic parameters that are **proven universal** in photosensitization studies. Moreover, it should be noted that in this work, we mainly focused on the analyses of the reaction products and product distributions. Our kinetic analysis focused on the decay rate constants of the aqSOA precursor (GUA) and the photosensitizers (DMB and VL) during photosensitization under the same experimental conditions (same aqSOA precursor and concentration, same photosensitizer concentration, and same lamp photon flux). The effects of other factors (e.g., intersystem crossing efficiency) on the rate constants were not examined. Hence, lacking rate constants that are either universal or specific for a number of experimental conditions in the literature, we hope that these parameters obtained in our study can provide first-order estimates for modeling since the experimental conditions (GUA+DMB, GUA+VL, GUA+DMB+AN, GUA+VL+AN, and GUA+AN based on concentrations relevant to cloud and fog conditions at pH 4 in air) are atmospherically relevant. Explicit kinetic studies (e.g., Smith et al., 2014, 2015) that measure second-order rate constants should be conducted in the future to extend the applicability of the kinetic parameters to other conditions.

We agree with the Reviewer that the apparent quantum efficiency of GUA photodegradation in the presence of DMB and likely VL as well is dependent on GUA concentration based on an earlier work (Anastasio et al., 1997). Regardless, this information would still be useful for comparison with GUA oxidation by other oxidants or photosensitizers.

The title of the paper has been revised to 'Comparison of aqueous SOA product distributions from guaiacol oxidation by non-phenolic and phenolic methoxybenzaldehydes as photosensitizers in the absence and presence of ammonium nitrate' to better reflect the focus of the work.

Furthermore, the following sentences have been added to the text to clarify the focus of this work:

Line 96: The precursor and photosensitizer decay kinetics, detected products, and absorbance enhancement were used to characterize the reactions. However, it should be noted that we mainly focused on the analyses of the reaction products and product distribution.

Line 192: Our kinetic analysis focused on the decay rate constants of the aqSOA precursor (GUA) and the photosensitizers (DMB and VL) during photosensitization under the same experimental conditions (same aqSOA precursor and concentration, same photosensitizer concentration, and same lamp photon flux). The effects of other factors (e.g., intersystem crossing efficiency) on the rate constants were not examined. Explicit kinetic studies (e.g., Smith et al., 2014, 2015) that measure second-order rate constants should be conducted in the future to extend the applicability of the kinetic parameters to other conditions.

3. (a) I don't see the utility of [P], the normalized product abundance. In the big picture, what does [P] indicate about a certain reaction system and what do differences in [P] between reaction systems indicate? (b) The authors present it as an equation without any in-depth discussion of its strengths and weaknesses.  Since [GUA]t/[GUA]0 is the inverse of the fraction of initial GUA that is present at time t, Equation 2 for [P] could be simplified as A(P,t)/A(GUA,0).  Thus [P] depends on at least three variables: (1) the extent of reaction, since A(P,t) probably rises initially and later falls, (2) the fraction of products that give a signal in the HPLC-Orbitrap (e.g., small organic acids probably do not), and (3) the ionization efficiency of each product in the Orbitrap. These issues need to be described in the manuscript; as part of this, the authors should say something about the IE values for the different classes of products that they see. (c) [P] also depends on what is used to normalize peak areas, e.g., VL in Mabato et al. (2022) and GUA here (making it very difficult to compare across the papers), and probably the initial concentration of the normalizing species. Given all of these variables, what do we learn from the Table 1 data of [P] after 180 min of illumination?  If they authors want to use [P] to describe products, they need to state what they think this parameter indicates, give us some experimental evidence that it's useful, and say something about its strengths and weaknesses. As it currently stands, the reported values of [P] seem to have no real utility.

Response: (a) The normalized abundance of products in this work is a semi-quantitative analysis intended to provide an overview of how the mass spectrometry (MS) signal intensities varied under different experimental conditions, but it does not quantify the absolute concentration of products from reaction systems. As demonstrated in our previous work (Mabato et al., 2022) and the current study, a higher normalized abundance of products correlates with higher efficiency of oxidation.

(b) Thank you for the suggestion. In this work and in our previous study (Mabato et al., 2022), we measured the absolute concentration changes of the precursor and photosensitizers using UHPLC-PDA. As we already have this quantitative data and given that it is a direct quantification of GUA concentration, we included the GUA concentration changes measured using UHPLC-PDA in the calculation of normalized product abundance to lessen its overall uncertainties. The simplification of the equation suggested by the Reviewer would yield the same trends in the original manuscript and would not change the conclusions. Therefore, we would like to retain the equation and the values presented in the original manuscript.

(1) We agree with the Reviewer that the [P] would depend on the extent of the reaction. However, in this work, the same irradiation time (180 min) was applied to all experiments as we were trying to compare the product distributions after a certain time of light photosensitization. An irradiation time of 180 min was chosen as it was sufficient to show the differences in the extent of reaction of GUA among the reaction systems studied. Moreover, our analysis of product distribution was focused only on aqSOA generated after 180 min of irradiation.

This has been clarified in the text are follows:

Lines 194-197: The products detected using UHPLC-HESI-Orbitrap-MS were used to  characterize the aqSOA formed in this work. The signal-weighted distributions of aqSOA calculated from combined positive (POS) and negative (NEG) ion modes MS results are summarized in Figure 2. The signal-weighted distributions calculated separately from POS and NEG ion modes MS results are available in Figures S2 and S3. It should be noted that in this work, the product distributions for all experiments were based on the same irradiation time of 180 min. An irradiation time of 180 min was chosen as it was sufficient to show the differences in the extent of reaction of GUA among the reaction systems studied. For reaction systems with precursors of different reactivities, chemical analysis at a fixed reaction time may be looking at different generations of products of each precursor, as Yu et al. (2014) reported. Measuring the product distribution at a fixed time might have missed the information on what/how many products are formed at the similar amounts of precursors reacted. The situation could be even more complicated if different precursors had major differences in pathways and dominant intermediates. However, comparing the product distributions after a certain time of light exposure, as is the case for this study, is useful to evaluate what products would form after a certain time of photosensitization.

(2) We agree with the Reviewer that certain products may not give signals (or may have weak signals) in the UHPLC-Orbitrap. We mentioned in the discussion of product distributions that the products detected using UHPLC-HESI-Orbitrap-MS were used to characterize the aqSOA formed in this work (line 194). Moreover, as stated in the footnote of Table 1, the normalized abundance of products was calculated using only the positive ion mode data as the GUA signal from the negative ion mode was weak and thus may present large uncertainties during normalization. Therefore, products that may not give signals or may have weak signals in the positive ion mode were possibly underestimated in the normalized product abundance. For example, small organic acids are more likely to be detected in the negative ion mode as this mode is more sensitive to deprotonatable compounds (Ho et al., 2003).

To clarify this, the following has been added to the description of normalized abundance of products (Sect 2.2):

It should be noted that the normalized abundance of products was calculated using only the positive ion mode data as the GUA signal from the negative ion mode was weak and thus may present large uncertainties during normalization. Therefore, products that may not give signals or may have weak signals in the positive ion mode were possibly underestimated in the normalized product abundance.

(3) It is very common to compare the relative abundance of products based on peak areas from MS results (e.g., Lee et al., 2014; Romonosky et al., 2017; Wang et al., 2017; Fleming et al., 2018; Song et al., 2018; Klodt et al., 2019; Ning et al., 2019) to show the relative importance of different types of compounds (K. Wang et al., 2021). However, comparisons of relative abundance among different compounds can be subject to uncertainties as ionization efficiencies in soft ionization, such as ESI, may significantly vary between different compounds (Kebarle, 2000; Schmidt et al., 2006; Leito et al., 2008; Perry et al., 2008; Kruve et al., 2014). **In our previous work (Mabato et al., 2022), we introduced the normalized abundance of products ([P], unitless).** The calculation assumes equal ionization efficiencies of different compounds, which is commonly used for the estimation of O:C ratios of SOA (e.g., Bateman et al., 2012; Lin et al., 2012; Laskin et al., 2014; De Haan et al., 2019). **This allows us to have a better comparison with the literature results.** It is therefore emphasized that the normalized abundance of products is a semi-quantitative analysis of the detected products under different experimental conditions, instead of absolute concentrations of them. As stated in the footnote of Table 1, the uncertainties were propagated from the changes in [GUA] measured using UHPLC-PDA and the MS signal intensities.

Unfortunately, there is limited availability of measured relative ionization efficiencies (RIE) for different compounds in the literature. We are in no position to provide the RIE values for the detected products. While ESI ionization is not ideal for quantification analysis of products, Nguyen et al. (2013) (https://doi.org/10.1039/C2AY25682G) demonstrated a positive correlation between ESI signal and "adjusted mass" (= molecular mass × H:C). According to that study, the uncertainty would be a factor of 2 − 4 if only the "adjusted mass" is considered, and further complications of matrix effect and polarity are ignored. **However, what we compared is not the absolute concentrations (or contributions) of the detected products. The comparison was based on how the signal intensities (as normalized in Eq. 2) varied under different experimental conditions so that the responses of the same class of products**

**(e.g., monomers, dimers, etc.) could be compared as the conditions varied.** The ionization efficiency might not be very different within the same class according to the "adjusted mass" concept by Nguyen et al. (2013). We hope that the above discussion has addressed the Reviewer's question.

(c) In this work, GUA was used to normalize the product peak areas as it is found in all samples, enabling comparisons among different experiments. In our previous study (Mabato et al., 2022), VL was used as it was both the dominant aqSOA precursor and the photosensitizer. Although the normalized abundance of products has inherent uncertainties due to the variability in ionization efficiencies for different compounds, **it can be used as a semi-quantitative analysis of the detected products that provides an overview of how the MS signal intensities varied under different experimental conditions.** As demonstrated in our previous work (Mabato et al., 2022) and the current study, a higher normalized abundance of products generally correlates with higher efficiency of oxidation. For instance, in our earlier study on the direct photosensitized oxidation of VL (Mabato et al., 2022), we found higher VL decay rate constants and higher normalized abundance of products under air-saturated conditions vs. nitrogen-saturated conditions. In this study, a higher GUA decay rate constant and higher normalized abundance of products was noted for GUA+DMB vs. GUA+VL.

Based on the responses above, Sect. 2.2 has been amended to give more information about the normalized abundance of products as follows:

**Section 2.2 Calculation of  normalized abundance of products**

Several recent studies have used comparisons of relative abundance of products based on peak areas from mass spectrometry (MS) results (e.g., Lee et al., 2014; Romonosky et al., 2017; Wang et al., 2017; Fleming et al., 2018; Song et al., 2018; Klodt et al., 2019; Ning et al., 2019) to show the relative importance of different types of compounds (K. Wang et al., 2021). However, comparisons of relative abundance among different compounds can be subject to uncertainties as ionization efficiencies in soft ionization, such as ESI, may significantly vary between different compounds (Kebarle, 2000; Schmidt et al., 2006; Leito et al., 2008; Perry et al., 2008; Kruve et al., 2014).  In our previous work (Mabato et al., 2022), we introduced the normalized abundance of products ([P], unitless) (Eq. 2) as  a semi-quantitative analysis that gives an overview of how the signal intensities changed under different experimental conditions but not the quantification of the absolute product concentration.  The calculation assumes equal ionization efficiencies of different compounds, which is commonly used to estimate O:C ratios of SOA (Bateman et al., 2012; Lin et al., 2012; Laskin et al., 2014; De Haan et al., 2019) :

$$[\mathrm{P}] = \frac{A_{P,t}}{A_{GUA,t}} \cdot \frac{[\mathrm{GUA}]_t}{[\mathrm{GUA}]_0} \qquad (\text{Eq. 2})$$

where $A_{P,t}$ and $A_{GUA,t}$ are the extracted ion chromatogram (EIC) peak areas of the product P and GUA from UHPLC-HESI-Orbitrap-MS analyses at time $t$, respectively; $[\mathrm{GUA}]_t$ and $[\mathrm{GUA}]_0$ are the GUA concentrations (μM) determined using UHPLC-PDA at time $t$ and 0, respectively. Note that the normalized abundance of products has intrinsic uncertainties due to the variability in ionization efficiencies for various compounds. Moreover, it should be noted that the normalized abundance of products was calculated using only the positive ion mode data as the GUA signal from the negative ion mode was weak and thus may present large uncertainties during normalization. Therefore, products that may not give signals or may have weak signals in the positive ion mode were possibly underestimated in the normalized product abundance. Nevertheless, it enables the comparison of MS results among different experiments. As demonstrated in our previous work (Mabato et al., 2022) and the current study, a higher normalized abundance of products generally correlates with higher efficiency of oxidation. The reported uncertainties were propagated from the changes in [GUA] measured using UHPLC-PDA and the MS signal intensities.

4. In each of the five systems, products were measured after 180 min of illumination. But this approach ignores the fact that the systems have different reactivities and so a fixed time of analysis is looking at different generations of products in the different systems (as shown by Yu et al.). This is important because the relative amounts of products are a function of oxidation time in any given solution.  So it is difficult to meaningfully compare across different solutions unless the GUA fraction reacted is very similar.  This is further complicated by the much faster decay of VL compared to DMB.

Response: The Reviewer raised a good point. However, in this work, the same irradiation time was applied to all experiments as we were trying to evaluate the product distributions after a certain time of photosensitization. An irradiation time of 180 min was chosen for this work as it was sufficient to show the differences in the extent of reaction of GUA among the reaction systems studied. It is true that for reaction systems with precursors of different reactivities, chemical analysis at a fixed reaction time may be looking at different generations of products of each precursor, as Yu et al. (2014) reported. Measuring the product distribution at a fixed time might have missed the information on what/how many products are formed at the similar amounts of precursors reacted. The situation could be even more complicated if different precursors had major differences in pathways and dominant intermediates. However, comparing the product distributions after a certain time of light exposure, as is the case for this study, is useful to evaluate what products would form after a certain time of photosensitization. The corresponding changes in the text (lines 194-197) are shown in our response to major comment #3.

5. The presentation of light absorption data for the reaction products is insufficient. There is one figure (Fig. 4) that sums absorbance values across 350 – 550 nm. This is interesting in that it shows the presence of ammonium nitrate doesn't affect overall absorbance, but this is too coarse a tool to describe the brown carbon products by itself.  It would be helpful to show spectra for each solution at 180 min in the supplement.  Also, Fig. 4 should be improved by weighting each absorbance spectrum by the spectral actinic flux to properly describe light absorption.  This could also be done by calculating the rate of sunlight absorption by each aqSOA for some standard sunlight condition.  This matters because the number of solar photons increases enormously from 350 to 550 nm.

Response: Thank you for the suggestions. The absorption spectra after 180 min of irradiation for each reaction system have been added to the supplement (Figure S7) as shown below:

[Figure]

**Figure S7**. UV-Vis absorption spectra of GUA+DMB+AN, GUA+DMB, GUA+VL+AN, GUA+VL, and GUA+AN after 180 min of irradiation. The inset is the expanded view from 350 to 550 nm.

Moreover, the absorbance enhancement from 0 to 180 min for all reaction systems studied have been added to Figure 4(a), based on suggestions from Reviewers 1 and 2. The original Figure 4 (now Figure 4b) has also been replaced with the change in the rate of sunlight absorption ($\Delta R_{abs}$) from 350-550 nm at 180 min during typical clear and haze days in Beijing, China. The updated Figure 4 is shown below:

[Figure]

**Figure 4.** (a) Increase in light absorption throughout 180 min of irradiation for all reaction systems studied and (b) Change in the rate of sunlight absorption ($\Delta R_{abs}$) from 350-550 nm at 180 min during typical clear and haze days in Beijing, China for aqSOA from GUA+DMB+AN, GUA+DMB, GUA+VL+AN, GUA+VL, and GUA+AN.  Error bars represent one standard deviation of triplicate experiments.

The corresponding revisions in the text are as follows:

Lines 253-260: The increase in light absorption throughout 180 min of irradiation and the change in the rate of sunlight absorption ($\Delta R_{abs}$) (Jiang et al., 2021) from 350 to 550 nm at 180 min during typical clear and haze days in Beijing, China for all the reaction systems studied are provided in Figure 4. Figure S7 shows the absorption spectra after 180 min of irradiation for each reaction system studied. In this work, the absorbance enhancement of GUA+DMB and GUA+VL (Fig. 4a) could be due to  oligomers and functionalized monomers, which are the highest contributors to the product signals. Identifying the chromophores responsible for the absorbance enhancement may be beneficial in understanding the impact of aqSOA on the Earth's radiative balance and determining the reactions that affect light absorption by aqSOA (Mabato et al., 2022). However, the detected products did not exhibit distinct peaks in the UHPLC-PDA chromatograms, likely due to the concentration of the chromophores being below the detection limit of PDA. Nevertheless, the higher absorbance enhancement and $\Delta R_{abs}$ for GUA+DMB than GUA+VL was  probably due to the higher contribution and normalized abundance (by ~6 times) of oligomers in the former.

Line 353: The presence of AN also did not appreciably affect the absorbance enhancement and $\Delta R_{abs}$ for both GUA+DMB+AN and GUA+VL+AN (Fig. 4).

**Other Comments**

1. Line 125. Just a note: you don't need to bubble air through a solution to make it air saturated. Shaking the solution with air in the headspace, then opening the container, and repeating this several times is sufficient. The downside to bubbling synthetic air is that you can introduce water-soluble contaminants from the air into the solution.

Response: Thank you for the suggestion. We will keep this in mind for our future work. We performed continuous bubbling of air into the reactor to ensure that the reactions would not be limited by oxygen availability.

2. Line 126. What was the flow rate of air through the solutions during illumination. Is it fast enough to be a significant loss mechanism for volatile compounds (e.g,. NOx, small organics, etc.)?

Response: The flow rate used in this work was 0.5 dm³/min. We have added the flow rate used as well as the initial volume of the solutions in the text as follows:

Line 125: The solutions (initial volume of 500 mL) were bubbled with synthetic air (0.5 dm³/min) for 30 min before irradiation and throughout the reactions to achieve air-saturated conditions (Du et al., 2011; Chen et al., 2020) and were continuously magnetically stirred.

Our control experiment of bubbling air through 0.01 mM formic acid for 6 hours did not result in a significant change ($p > 0.05$) in concentration. Based on this, we believe that the flow rate used in this work is not fast enough to be a significant loss mechanism for volatile compounds (e.g., NOx, small organics, etc.).

3. Line 129. (a) What was the pathlength of the photoreactor? (b) What were the corresponding light screening factors for each solution? (c) Are corrections for light screening required to correct the rate constants?

Response: (a) The path length of the photoreactor is 8.5 cm.

(b) Following Smith et al. (2014, 2016), the values of the internal light screening factor ($S_\lambda$) determined around the peak in the light absorption action spectrum (DMB: 310-335 nm, VL: 304-364 nm, and nitrate: 300-331 nm) for an 8.5 cm cell were as follows:

| Reaction conditions | Internal light screening factor ($S_\lambda$) |
|---|---|
| GUA+AN | 0.95 |
| GUA+DMB | 0.51 |
| GUA+DMB+AN | 0.54 |
| GUA+VL | 0.57 |
| GUA+VL+AN | 0.59 |

(c) Thank you for pointing this out. The decay rate constants were initially not corrected for light screening as the same concentration was used for DMB and VL. However, given the significant difference between the $S_\lambda$ of GUA+AN and those of other reaction conditions, we corrected the decay rate constants for $S_\lambda$ in Table 1. The following information has been added to the text to reflect these changes:

Line 143: The decay rate constants were normalized to the photon flux measured for each experiment through dividing $k'$ by the measured 2-nitrobenzaldehyde (2NB; a chemical actinometer) decay rate constant, $j$(2NB) (Mabato et al., 2022). In addition, the decay rate constants were corrected for the internal light screening due to DMB, VL, and AN (Leifer, 1988; Zhang and Anastasio, 2003; Smith et al., 2014, 2016). The values of the internal light screening factor ($S_\lambda$) determined around the peak in the light absorption action spectrum (DMB: 310-335 nm, VL: 304-364 nm, nitrate: 300-331 nm) (Smith et al., 2014, 2016) for an 8.5 cm cell were 0.95 for GUA+AN, 0.51 for GUA+DMB, 0.54 for GUA+DMB+AN, 0.57 for GUA+VL, and 0.59 for GUA+VL+AN.

Footnote of Table 1: [a]The data fitting was performed in the initial linear region. Each value is the average of results from triplicate experiments, corrected for internal light screening due to DMB, VL, and AN, and normalized to the experimental photon flux. Errors represent one standard deviation.

4. Line 141. Were the decays of GUA, VL, and DMB always first-order? It seems unlikely given that the reactions proceeded for many half-lives of some of the compounds (e.g., VL). It would be helpful to show both examples of good (first-order) and not so good kinetics in the supplement.

Response: In this work, the decay of GUA, DMB, and VL were treated as pseudo-first-order, and the reported decay rate constants were obtained by performing the data fitting in the initial linear region, as noted in the original text (line 176) and Table 1. This is based on the consideration that the exact composition of the solutions during the reactions is unknown (e.g., due to the generation of intermediates).

The plots of the GUA, DMB, and VL decay (shown below) have now been added to Figure S1. The corresponding changes in the text are as follows:

Line 180: Figure S1 shows the decay of GUA, DMB, and VL under different experimental conditions.

[Figure]

Figure S1. (a) The decay of GUA during (ammonium) nitrate-mediated photo-oxidation (GUA+AN) and photosensitized oxidation by $^3VL^*$ (GUA+VL) or $^3DMB^*$ (GUA+DMB). (b) The decay of DMB or VL during GUA photo-oxidation in GUA+DMB and GUA+VL, respectively. No statistically significant difference ($p > 0.05$) was noted between GUA+DMB and GUA+DMB+AN and between GUA+VL and GUA+VL+AN. Error bars represent 1 standard deviation; most error bars are smaller than the markers.

5. Line 234. These two sentences seem contradictory: GUA+DMB had more compounds with higher O:C, but GUA+VL had a higher average OS(C). How to reconcile this apparent discrepancy?

Response: We apologize that we may have misled the Reviewer. They are not inconsistent. The statement on GUA+DMB indicates having more compounds in terms of quantity or number with higher O:C than in GUA+VL. However, overall, the average O:C and $OS_C$ calculated using the signal-weighted method (Bateman et al., 2012) were higher for GUA+VL than for GUA+DMB. This is probably due to the more significant functionalization in GUA+VL (as mentioned in line 236).

6. Line 254. Is this statement based only on the higher amounts of oligomers and functionalized monomers in the GUA+DMB case compared to the GUA+VL case? If so, this is weak evidence and really not a "correlation".

Response: Line 254 does not compare the absorbance enhancement for GUA+DMB vs. GUA+VL. Rather, it is about attributing the absorbance enhancement for both GUA+DMB and GUA+VL to oligomers and functionalized monomers which are the highest signal contributors. Regardless, we revised the text as follows:

Line 254: In this work, the absorbance enhancement of GUA+DMB and GUA+VL (Fig. 4a) could be due to  oligomers and functionalized monomers, which are the highest contributors to the product signals.

7. Table 1. (a) Are these rate constants normalized to a specific j(2NB) value? Line 144 indicates that rate constants were normalized by dividing by j(2NB), but this does not appear to have been done to the Table 1 rate constants based on their units. If the authors aren't going to normalize the rate constants, they should discuss the variation in j(2NB) across their samples and give average j(2NB) values for each reaction condition. (b) In the presence of AN, GUA has a rate constant of 8.1E-3 min-1, which is appreciable. But the addition of AN to the DMB or VL solutions has no apparent impact on the rate constant for GUA loss, with a difference much less than the addition increment expected of 8E-3 min-1. How to explain this discrepancy in the kinetics? Is light screening an issue? (c) Experiment #4 is labeled as a second #3.

Response: (a) Thank you for pointing this out. Yes, these rate constants were normalized to the photon flux measured for each experiment, but we inadvertently used the wrong unit for 2-nitrobenzaldehyde (2NB) decay rate constants during normalization. Table 1 now shows the correct photon flux-normalized rate constants, which were also corrected for internal light screening due to DMB, VL, and AN, based on the Reviewer's other comment #3. We apologize for the confusion.

(b) The addition of AN to GUA+DMB and GUA+VL had no significant effect on the decay rate constants, likely due to the higher molar absorptivities of DMB and VL compared to that of nitrate. This indicates that the chemistry of $^3DMB^*$ and $^3VL^*$ dominated that of nitrate photolysis. This information was already provided in lines 277-279. The estimated light-absorbing fraction (molar absorptivities from 300 to ~360 nm × concentration) of DMB, VL, and nitrate in these multicomponent systems were: 95.50% DMB and 4.50% nitrate for GUA+DMB+AN; and 95.12% VL and 4.88% nitrate for GUA+VL+AN. Similarly, in our previous work, we estimated comparable VL decay constants for the direct photosensitized oxidation of VL in the absence and presence of AN (Mabato et al., 2022). We do not think that this is a light screening issue as $S_\lambda$ for GUA+DMB (0.51) and GUA+DMB+AN (0.54) (also for GUA+VL, 0.57, and GUA+VL+AN, 0.59) were comparable. However, although AN did not significantly affect the oxidation kinetics, it had distinct effects on the product distributions.

(c) Thank you for catching this error. The labeling has been corrected in Table 1.

**Minor Comments**

1. Line 87. The Henry's law constant listed for DMB (5.4E1 M/atm) is far too low. This is either a typo or a problem in the source reference.

Response: The initially provided Henry's law constant for DMB ($5.4 \times 10^1$ M atm$^{-1}$) is based on the group method estimate from EPI Suite. We have updated the listed Henry's law constant based on vapor pressure/water solubility estimate, which is more accurate compared to the group and bond methods:

Line 87: DMB and VL (Henry's law constants of  $7.3 \times 10^3$ M atm$^{-1}$ and $4.7 \times 10^5$ M..

2. Line 108. "…AN generated more N-containing products…" More than what condition?

Response: We apologize for the confusion. AN generated more N-containing products in GUA+DMB+AN than in GUA+VL+AN. This has been corrected in the text as follows:

Line 107: In particular, AN generated more N-containing products in GUA+DMB+AN than in GUA+VL+AN, and increased the oligomers in GUA+VL+AN.

3. Line 129. To better simulate sunlight, I recommend the authors add an airmass filter to their illumination system for future work.

Response: Thank you for the suggestion. We will keep this in mind for our future work.

4. Line 191. The authors posit that the OH + VL rate constant is larger than the OH + DMB rate constant, but it seems unlikely that the difference is large given that they are probably both very fast. Have the authors looked for these rate constants?

Response: This statement is based on the -OH group (in VL) being more activating than the -OCH$_3$ group (in DMB). The second-order rate constant for the reaction between $^\bullet$OH and VL has been reported to be $4 \times 10^8$ M$^{-1}$ s$^{-1}$ (Li et al., 2014), while we are unaware of any data for $^\bullet$OH + DMB. To avoid confusion, we omitted this phrase in the text.

5. Line 377. Should clarify that this is referring to stronger light absorption by the products.

Response: Thank you for the suggestion. This has been clarified in the text as follows:

Line 377: Compared to GUA+VL, faster GUA oxidation and stronger light absorption by the products were observed in GUA+DMB.

Line 271: In summary, $^3$DMB* and $^3$VL* can oxidize GUA resulting in aqSOA and BrC formation, but GUA+DMB products exhibited stronger light absorption.

**Recommendation**

This is a difficult manuscript to rate, as it has a few interesting points but some major issues. However, given that there is not a lot that is novel about the work, I am sorry to recommend that it be rejected.

**References for responses to Reviewer 4:**

[revised manuscript text omitted]

**Section S1. UHPLC-PDA analyses**

An ultra-high performance liquid chromatography system (UHPLC, Waters Acquity H-Class, Waters, Milford, USA) coupled to a photodiode array (PDA) detector (Waters, Milford, USA) was used for the quantification of GUA, DMB, and VL concentrations. The samples were first filtered through a 0.2 μm Chromafil® Xtra PTFE filter (Macherey-Nagel GmbH & Co. KG, Germany). The separation of products was conducted using an Acquity HSS T3 column (1.8 μm, 2.1 mm × 100 mm; Waters Corp.). The column oven was held at 30 °C, and the autosampler was cooled at 4 °C. The injection volume was set to 5 μL. The binary mobile phase was composed of water (A) and acetonitrile (B). The gradient elution was performed at a flow rate of 0.2 mL/min: 0–1 min, 10% eluent B; 1–25 min, linear increase to 90% eluent B; 25–29.9 min, hold 90% eluent B; 29.9–30 min, decrease to 10% eluent B; 30–35 min, re-equilibrate at 10% eluent B for 5 min. GUA, DMB, and VL were analyzed using the channels with UV absorption at 274, 274, and 300 nm, respectively.

**Section S2. UHPLC-HESI-Orbitrap-MS analyses**

A Thermo Orbitrap Fusion Lumos Mass Spectrometry (Thermo Fisher Scientific, Waltham, MA, USA) connected to a Thermo Scientific UltiMate 3000 UHPLC system (Thermo Fisher Scientific, Waltham, MA, USA) via heated electrospray ionization (HESI) as the interface (UHPLC-HESI-Orbitrap-MS) was used to characterize the reaction products. The mobile phases used were 0.1% (v/v) formic acid (in milli-Q water) (A) and acetonitrile (B). The same settings (e.g., column, gradient, oven temperature) used in the UHPLC-PDA (Sect. S1) were applied in the UHPLC-HESI-Orbitrap-MS system. The HESI-MS spectra were acquired in both positive and negative ion modes. The HESI parameters were as follows: Spray voltage, 2500 V for both positive and negative HESI; sheath gas, 35 arbitrary units; nebulizer auxiliary gas, 10 arbitrary units; sweep gas, 3 arbitrary units. General instrumental parameters were set as follows: ion transfer tube temperature, 320 °C; vaporizer temperature, 350 °C. The mass range for full scan MS was set at 50-1000 m/z with a mass resolution of 60,000 at 200 m/z.

The automatic gain control (AGC) target was $4.0 \times 10^5$ with a maximum injection time of 50

ms. The UHPLC-HESI-Orbitrap-MS data obtained in positive and negative ion modes were pretreated using Progenesis QI (version 2.4; Nonlinear Dynamics) for peak picking and alignment. Most peaks detected in the blank (~99% for all experiments) were excluded from the samples except for peaks with a minimum of 2.5 times greater intensity in the sample spectrum than in the blank (Laskin et al., 2014). In addition, a peak was considered a product if the difference in the peak area between the samples before and after irradiation is ≥10 times.

In this work, two independently prepared samples for each reaction condition were analyzed using the UHPLC-HESI-Orbitrap-MS. Only peaks that were reproducibly detected in both sets of samples were retained. The formula assignments were carried out using the MIDAS

molecular formula calculator (http://magnet.fsu.edu/~midas/) with the following constraints: C

≤100, H ≤150, O ≤30, and N ≤10, and mass error of 10 ppm. The nitrogen atom was excluded in the constraints for experiments without AN. The ChemSpider database (Royal Society of

Chemistry) was also queried to return valid molecules that may be useful for proposing product structures. Overall, the proposed structures in this work are based on the molecular formulas,

DBE values, and structural and mechanistic information provided in earlier similar works on methoxyphenols (Yee et al., 2013; Li et al., 2014; Yu et al., 2014, 2016; He et al., 2019; Chen et al., 2020; Jiang et al., 2021; Misovich et al., 2021; Mabato et al., 2022). For clarity, the formulas discussed in this work correspond to neutral analytes (e.g., with $H^+$ or $NH_4^+$ removed from the ion formula).

The double bond equivalent (DBE) values (Koch and Dittmar, 2006) and carbon oxidation state ($OS_C$; Kroll et al., 2011, 2015; Lv et al., 2016) of the neutral formulas were calculated using the following equations:

$DBE = C - H/2 + N/2 + 1$  (Eq. S1)

$OS_C = 2 \times O/C + 3 \times N/C - H/C$  (Eq. S2)

where C, H, O, and N correspond to the number of carbon, hydrogen, oxygen, and nitrogen atoms in the neutral formula. Moreover, the average oxygen to carbon (O:C) ratios, $\langle O{:}C \rangle$:

$(\langle O{:}C \rangle = \sum_i (\text{abundance}_i) O_i / \sum_i (\text{abundance}_i) C_i)$, average nitrogen to carbon (N:C) ratios,

$\langle N{:}C \rangle$: $(\langle N{:}C \rangle = \sum_i (\text{abundance}_i) N_i / \sum_i (\text{abundance}_i) C_i)$, and average hydrogen to carbon (H:C) ratios, $\langle H{:}C \rangle$: $(\langle H{:}C \rangle = \sum_i (\text{abundance}_i) H_i / \sum_i (\text{abundance}_i) C_i)$ after the reactions were further estimated using the signal-weighted method (Bateman et al., 2012). The average

$OS_C$, $\langle OS_C \rangle$ was also calculated as follows:

$\langle OS_C \rangle = 2 \times \langle O{:}C \rangle + 3 \times \langle N{:}C \rangle - \langle H{:}C \rangle$  (Eq. S3)

**Section S3. IC analyses of small organic acids**

An ion chromatography system (IC, Dionex ICS-1100, Sunnyvale, CA) equipped with a

Dionex AS-DV autosampler (Sunnyvale, CA) enabled the analyses of small organic acids. The separation was achieved using an IonPac™ AS11 column (4 × 250 mm) with an IonPac™

AG11 guard column (4 × 50 mm). The isocratic elution was applied at a 1.0 mL/min flow rate with 12 mM sodium hydroxide (NaOH) as the eluent. The total run time was set at 10 min. The standard solutions (1–50 μM) of formic, succinic, and oxalic acid were analyzed three times along with the samples and water blank. Formic, succinic, and oxalic acid had retention times of 1.9 min, 3.7 min, and 5.9 min, respectively.

**Section S4. UV-Vis spectrophotometric analyses**

A UV-Vis spectrophotometer (UV-3600, Shimadzu Corp., Japan) was used to measure the absorbance changes for the samples. The absorbance values from 200 to 700 nm were measured instantly after sample collection, and measurements were done in triplicate. The change in the integrated area of absorbance from 350 to 550 nm was used to represent the absorbance enhancements. The increase of light absorption at this wavelength range, where GUA, DMB, and VL did not initially absorb light, suggests the formation of light-absorbing products (Smith et al., 2016).

**Section S5. Further discussions on van Krevelen diagrams and $OS_c$ vs. $n_C$ plots for**

**GUA+DMB and GUA+VL aqSOA**

Consistent with the higher contribution of ring-opening species, GUA+DMB had more products with H:C $\geq$1.5 and O:C $\leq$0.5 (Fig. S54a–b), possibly due to more oxygenated aliphatic species. GUA+VL (Fig. S54c–d) also had high-relative-abundance products with H:C of ~1

and O:C $\geq$0.5. Similar to our previous work (0.1 mM GUA + 0.1 mM VL; Mabato et al., 2022), the two high-relative-abundance species with O:C $\geq$0.5 were associated with hydroxylated products ($C_7H_8O_4$ and $C_8H_8O_5$, #28 and 35; Table S2) that were also observed in earlier works on $^3DMB^*$ and $^{\bullet}OH$-mediated oxidation (Yu et al., 2014, 2016). These hydroxylated products were also present in GUA+DMB but with lower relative abundance. Triplet-mediated phenol oxidation can generate $H_2O_2$ (Anastasio et al., 1997), a photolytic source of $^{\bullet}OH$. Indeed, hydroxylation is significant in aqueous-phase phenol oxidation (Li et al., 2014; Yu et al., 2014,

2016; Chen et al., 2020; Jiang et al., 2021; Misovich et al., 2021; Mabato et al., 2022).

The $OS_C$ vs. $n_C$ plots for both GUA+DMB and GUA+VL display high-relative- abundance species clustered at $n_C$ of 12 to 15 and $OS_C$ >-1, which can be ascribed to dimers and derivatives (Fig. S65a–d). The species with $n_C$ >15 had the highest DBE values and can be attributed to trimers. These compounds were more abundant in GUA+DMB, likely due to the greater extent of photosensitized reactions by $^3DMB^*$ compared to $^3VL^*$. Indeed, oligomerization is an important process in aqSOA formation via triplet-mediated oxidation (Yu et al., 2014, 2016; Chen et al., 2020; Jiang et al., 2021; Misovich et al., 2021; Mabato et al.,

2022). As indicated by the higher quantity of low DBE species, ring-opening and fragmentation pathways were more extensive in GUA+DMB. In GUA+VL, there were also high-relative- abundance products with $n_C$ <10, $OS_C \geq 0$, and DBE <5, corresponding to the hydroxylated products mentioned earlier.

**Section S6. Estimation of the apparent quantum efficiency of guaiacol photodegradation**

The apparent quantum efficiency of GUA photodegradation ($\varphi_{GUA}$) in the presence of DMB,

VL, or nitrate during simulated sunlight illumination can be defined as (Anastasio et al., 1997;

Smith et al., 2014, 2016):

$\Phi_{GUA} = \dfrac{\text{mol GUA destroyed}}{\text{mol photons absorbed}}$                    (Eq. S4)

$\Phi_{GUA}$ was calculated using the measured rate of GUA decay and rate of light absorption by

DMB, VL, or nitrate through the following equation:

$\Phi_{GUA} = \dfrac{\text{rate of GUA decay}}{\text{rate of light absorption by DMB or VL or nitrate}} = \dfrac{k'_{GUA} \times [GUA]}{\sum[(1-10^{-\varepsilon_\lambda[C]l}) \times I'_\lambda]}$          (Eq. S5)

where $k'_{GUA}$ is the pseudo-first-order rate constant for GUA decay, [GUA] is the concentration of GUA (M), $\varepsilon_\lambda$ is the base-10 molar absorptivity ($M^{-1}$ $cm^{-1}$) of DMB, VL, or nitrate at wavelength $\lambda$, [C] is the concentration of DMB, VL, or nitrate (M), $l$ is the pathlength of the illumination cell (cm), and $I'_\lambda$ is the volume-averaged photon flux (mol-photons $L^{-1}$ $s^{-1}$ $nm^{-1}$)

determined from 2NB actinometry:

$j(2NB) = 2.303 \times \Phi_{2NB} \times l \times \sum_{300\,nm}^{350\,nm}(\varepsilon_{2NB,\lambda} \times I'_\lambda \times \Delta\lambda)$          (Eq. S6)

where $j(2NB)$ is the decay rate constant of 2-nitrobenzaldehyde (2NB), the chemical actinometer used to determine the photon flux in the aqueous photoreactor, $\Phi_{2NB,\lambda}$ and $\varepsilon_{2NB,\lambda}$

are the quantum yield (molecule $photon^{-1}$) and base-10 molar absorptivity ($M^{-1}$ $cm^{-1}$) for 2NB, respectively, and $\Delta\lambda$ is the wavelength interval between actinic flux data points (nm).

**Section S7. Further discussions on van Krevelen diagrams and $OS_c$ vs. $n_C$ plots for**

**GUA+DMB+AN, GUA+VL+AN, and GUA+AN aqSOA**

The position of the CHO, CHON, and CHN  species in the van Krevelen diagrams for

GUA+DMB+AN and GUA+VL+AN broadly resembled those of CHO species in the absence of AN (Fig. S5). The CHON species for GUA+DMB+AN and GUA+VL+AN mostly had

O:C ratios <0.7, consistent with previous studies on BBOA e.g., wheat straw burning in K-

Puszta in the Great Hungarian Plain of Hungary, biomass burning at Canadian rural sites such as Saint Anicet, and BBOA from Amazonia (Schmitt-Kopplin et al., 2010; Claeys et al., 2012;

Kourtchev et al., 2017).

The CHN species in GUA+DMB+AN and GUA+VL+AN appeared to have analogous

H:C ratios. GUA+DMB+AN had ~2 times more CHON and CHN species than GUA+VL+AN, and there were more of these species with higher abundance in the former, indicating a greater extent of reactions with AN. The high-relative-abundance products for GUA+DMB+AN and

GUA+VL+AN were similar to those in the absence of AN, except the hydroxylated products (e.g., $C_7H_8O_4$; #28; Table S2) previously mentioned for GUA+VL. Among the high-relative- abundance products for GUA+DMB+AN was a CHN species with H:C of ~0.8. For

GUA+VL+AN, the high-relative-abundance products include two CHON species with O:C

and H:C ratios of 0.3-0.6 and 0.6-0.8. The major difference between GUA+AN and

GUA+DMB+AN/GUA+VL+AN was the presence of more high-relative-abundance CHON

and CHN species (Fig. S9) in GUA+AN which can be expected given that AN was the only source of oxidants in this case. Compared to GUA+AN, more species (CHO, CHON, and

CHN) were observed for GUA+DMB+AN and GUA+VL+AN, attributable to contributions from both photosensitization and (ammonium) nitrate photolysis.

Moreover, GUA+DMB+AN and GUA+VL+AN aqSOA had mainly similar features in the $OS_C$ vs $n_C$ plots as those observed in the absence of AN (Fig. S6). GUA+DMB+AN and

GUA+VL+AN aqSOA also had more CHON and CHN species with higher $OS_C$, $n_C$, and DBE

(Fig. S6e–h) relative to GUA+AN (Fig. S10), indicating more conjugated N-containing compounds. For GUA+DMB+AN and GUA+VL+AN, the CHON and CHN species had a wider range of $OS_C$ compared to CHO species (Fig. S6e–h). The high-relative-abundance species ($n_C$ of 12 to 15 and $OS_C$ >-1) corresponded to dimers and trimers similar to those noted in the absence of AN, along with some N-containing species. These include a CHN species with $n_C$ of 13, $OS_C$ ~0, and 11 DBE for GUA+DMB+AN, and 2 CHON species with $n_C$ of 5

and 11, $OS_C$ of 2.5 and 1, and 6 and 9 DBE for GUA+VL+AN.

**Table S1**. Possible structures of the major products detected from GUA+DMB using UHPLC-
HESI-Orbitrap-MS operated in positive (POS) and negative (NEG) ion modes.

| No. | GUA+DMB POS Molecular formula and exact mass | DBE | Possible structure | No. | GUA+DMB NEG Molecular formula and exact mass | DBE | Possible structure |
|---|---|---|---|---|---|---|---|
| 1 | $C_{14}H_{14}O_4$ (246.0892) | 8 |  | | $C_{14}H_{14}O_4$ (246.0892) (No. 1; GUA+DMB POS) | | |
| 2 | $C_{13}H_{10}O_3$ (214.0630) | 9 |  | 16 | $C_{14}H_{14}O_6$ (278.0790) | 8 |  |
| 3 | $C_{14}H_{12}O_4$ (244.0736) | 9 |  | 17 | $C_{12}H_{10}O_4$ (218.0579) | 8 |  |
| 4 | $C_{13}H_{10}O_4$ (230.0579) | 9 |  | | $C_{13}H_{12}O_4$ (232.0736) (No. 6; GUA+DMB POS) | | |
| 5 | $C_{13}H_{10}O_5$ (246.0528) | 9 |  | 18 | $C_7H_{10}O_5$ (174.0528) | 3 |  |
| 6 | $C_{13}H_{12}O_4$ (232.0736) | 8 |  | 19 | $C_{21}H_{18}O_8$ (398.1002) | 13 |  |
| 7 | $C_{14}H_{12}O_5$ (260.0685) | 9 |  | 20 | $C_{13}H_{12}O_6$ (264.0634) | 8 |  |
| 8 | $C_{11}H_{12}O_5$ (224.0685) | 6 |  | 21 | $C_{20}H_{18}O_6$ (354.1103) | 12 |  |
| 9 | $C_{14}H_{12}O_7$ (292.0583) | 9 |  | 22 | $C_{14}H_{14}O_7$ (294.0740) | 8 |  |
| 10 | $C_{11}H_{14}O_6$ (242.0790) | 5 |  | 23 | $C_{12}H_{14}O_4$ (222.0892) | 6 |  |

| 11 | C$_{18}$H$_{18}$O$_7$ (346.1053) | 10 |  | 24 | C$_{13}$H$_{10}$O$_6$ (262.0477) | 9 |  |
|----|-------------------------------|----|----------------------|----|--------------------------------|---|----------------------|
| 12 | C$_{10}$H$_{12}$O$_3$ (180.0786) | 5 |  | 25 | C$_{13}$H$_{14}$O$_4$ (234.0892) | 7 |  |
| 13 | C$_7$H$_6$O$_4$ (154.0266) | 5 |  | 26 | C$_{14}$H$_{14}$O$_5$ (262.0841) | 8 |  |
| 14 | C$_{16}$H$_{18}$O$_6$ (306.1103) | 8 |  | C$_{13}$H$_{10}$O$_5$ (246.0528) (No. 5; GUA+DMB POS) | | | |
| 15 | C$_7$H$_6$O$_5$ (170.0215) | 5 |  | 27 | C$_{19}$H$_{16}$O$_6$ (340.0947) | 12 |  |

Table S2. Possible structures of the major products detected from GUA+VL using UHPLC-HESI- Orbitrap-MS operated in positive (POS) and negative (NEG) ion modes.

| No. | GUA+VL POS Molecular formula and exact mass | DBE | Possible structure | No. | GUA+VL NEG Molecular formula and exact mass | DBE | Possible structure |
|---|---|---|---|---|---|---|---|
| 28 | $C_7H_8O_4$ (156.0423) | 4 |  | 35 | $C_8H_8O_5$ (184.0372) | 5 |  |
| $C_{13}H_{10}O_4$ (230.0579) (No. 4; GUA+DMB, Table S1) | | | | $C_{13}H_{12}O_4$ (232.0736) (No. 6; GUA+DMB, Table S1) | | | |
| $C_{13}H_{12}O_4$ (232.0736) (No. 6; GUA+DMB, Table S1) | | | | $C_{14}H_{14}O_4$ (246.0892) (No. 1; GUA+DMB, Table S1) | | | |
| $C_{13}H_{10}O_5$ (246.0528) (No. 5; GUA+DMB, Table S1) | | | | $C_{14}H_{14}O_6$ (278.0790) (No. 16; GUA+DMB, Table S1) | | | |
| 29 | $C_7H_8O_5$ (172.0372) | 4 |  | $C_{20}H_{18}O_6$ (354.1103) (No. 21; GUA+DMB, Table S1) | | | |
| 30 | $C_6H_6O_2$ (110.0368) | 4 |  | $C_{12}H_{10}O_4$ (218.0579) (No. 17; GUA+DMB, Table S1) | | | |
| 31 | $C_{10}H_{10}O_4$ (194.0579) | 6 |  | $C_6H_6O_2$ (110.0368) (No. 30; GUA+VL POS) | | | |
| 32 | $C_{11}H_8O_4$ (204.0423) | 8 |  | $C_7H_{10}O_5$ (174.0528) (No. 18; GUA+DMB, Table S1) | | | |
| 33 | $C_{12}H_{10}O_3$ (202.0630) | 8 |  | 36 | $C_{15}H_{14}O_5$ (274.0841) | 9 |  |
| $C_{14}H_{12}O_5$ (260.0685) (No. 7; GUA+DMB, Table S1) | | | | $C_{13}H_{12}O_6$ (264.0634) (No. 20; GUA+DMB, Table S1) | | | |
| $C_{13}H_{14}O_4$ (234.0892) (No. 25; GUA+DMB, Table S1) | | | | 37 | $C_8H_8O_4$ (168.0423) | 5 |  |
| 34 | $C_{11}H_{10}O_6$ (238.0477) | 7 |  | $C_{19}H_{16}O_6$ (340.0947) (No. 27; GUA+DMB, Table S1) | | | |
| $C_{13}H_{10}O_6$ (262.0477) (No. 24; GUA+DMB, Table S1) | | | | $C_{11}H_{10}O_6$ (238.0477) (No. 34; GUA+VL POS) | | | |
| $C_{13}H_{12}O_6$ (264.0634) (No. 20; GUA+DMB, Table S1) | | | | 38 | $C_5H_6O_5$ (146.0215) | 3 |  |
| $C_7H_6O_4$ (154.0266) (No. 13; GUA+DMB, Table S1) | | | | 39 | $C_6H_4O_4$ (140.0110) | 5 |  |

**Table S3**. Possible structures of the major products detected from GUA+DMB+AN using
UHPLC-HESI-Orbitrap-MS operated in positive (POS) and negative (NEG) ion modes.

| No. | GUA+DMB+AN POS Molecular formula and exact mass | DBE | Possible structure | No. | GUA+DMB +AN NEG Molecular formula and exact mass | DBE | Possible structure |
|---|---|---|---|---|---|---|---|
| $C_{14}H_{14}O_4$ (246.0892) (No. 1; GUA+DMB, Table S1) | | | | $C_{13}H_{12}O_4$ (232.0736) (No. 6; GUA+DMB, Table S1) | | | |
| 40 | $C_{13}H_{10}N_4$ (222.0905) | 11 |  | $C_{14}H_{14}O_6$ (278.0790) (No. 16; GUA+DMB, Table S1) | | | |
| $C_{13}H_{10}O_5$ (246.0528) (No. 5; GUA+DMB, Table S1) | | | | $C_{14}H_{14}O_4$ (246.0892) (No. 1; GUA+DMB, Table S1) | | | |
| $C_{13}H_{10}O_4$ (230.0579) (No. 4; GUA+DMB, Table S1) | | | | $C_{12}H_{10}O_4$ (218.0579) (No. 17; GUA+DMB, Table S1) | | | |
| 41 | $C_6H_6N_4$ (134.0592) | 6 |  | $C_{21}H_{18}O_8$ (398.1002) (No. 19; GUA+DMB, Table S1) | | | |
| $C_{13}H_{12}O_4$ (232.0736) (No. 6; GUA+DMB, Table S1) | | | | $C_7H_{10}O_5$ (174.0528) (No. 18; GUA+DMB, Table S1) | | | |
| 42 | $C_{12}H_{11}N_3O_3$ (245.0800) | 9 |  | $C_{13}H_{12}O_6$ (264.0634) (No. 20; GUA+DMB, Table S1) | | | |
| 43 | $C_{10}H_8N_4O$ (200.0698) | 9 |  | 48 | $C_{16}H_{14}N_6O_4$ (354.1076) | 13 |  |
| 44 | $C_6H_6N_4O$ (150.0542) | 6 |  | 49 | $C_{15}H_{10}N_4O_3$ (294.0753) | 13 |  |
| 45 | $C_{10}H_{14}N_4O_4$ (245.1015) | 6 |  | $C_{13}H_{10}O_6$ (262.0477) (No. 24; GUA+DMB, Table S1) | | | |
| 46 | $C_{13}H_{10}N_4O$ (238.0855) | 11 |  | $C_{10}H_{10}O_4$ (194.0579) (No. 31; GUA+VL, Table S2) | | | |
| $C_{13}H_{12}O_6$ (264.0634) (No. 20; GUA+DMB, Table S1) | | | | $C_7H_8O_4$ (156.0423) (No. 28; GUA+VL, Table S2) | | | |
| $C_7H_6O_4$ (154.0266) (No. 13; GUA+DMB, Table S1) | | | | $C_{13}H_{14}O_4$ (234.0892) (No. 25; GUA+DMB, Table S1) | | | |
| $C_{12}H_{10}O_3$ (202.0630) (No. 33; GUA+VL, Table S2) | | | | $C_{13}H_{10}O_5$ (246.0528) (No. 5; GUA+DMB, Table S1) | | | |
| 47 | $C_{13}H_8O_4$ (228.0423) | 10 |  | $C_{14}H_{14}O_5$ (262.0841) (No. 26; GUA+DMB, Table S1) | | | |

**Table S4**. Possible structures of the major products detected from GUA+VL+AN using
UHPLC-HESI-Orbitrap-MS operated in positive (POS) and negative (NEG) ion modes.

| No. | GUA+VL+AN POS Molecular formula and exact mass | DBE | Possible structure | No. | GUA+VL+AN NEG Molecular formula and exact mass | DBE | Possible structure |
|---|---|---|---|---|---|---|---|
| | $C_{14}H_{14}O_4$ (246.0892) (No. 1; GUA+DMB, Table S1) | | | | $C_{13}H_{12}O_4$ (232.0736) (No. 6; GUA+DMB, Table S1) | | |
| 50 | $C_{10}H_8O_2$ (160.0524) | 7 |  | | $C_{14}H_{14}O_6$ (278.0790) (No.16; GUA+DMB, Table S1) | | |
| 51 | $C_{16}H_{18}O_4$ (274.1205) | 8 |  | | $C_{12}H_{10}O_4$ (218.0579) (No.17; GUA+DMB, Table S1) | | |
| | $C_{11}H_{12}O_5$ (224.0685) (No. 8; GUA+DMB, Table S1) | | | 57 | $C_{11}H_9N_3O_3$ (231.0644) | 9 |  |
| | $C_{14}H_{12}O_5$ (260.068) (No. 7; GUA+DMB, Table S1) | | | | $C_7H_{10}O_5$ (174.0528) (No.18; GUA+DMB, Table S1) | | |
| | $C_{12}H_{14}O_4$ (222.0892) (No. 23; GUA+DMB, Table S1) | | | | $C_{15}H_{14}O_5$ (274.0841) (No. 36; GUA+VL, Table S2) | | |
| 52 | $C_{11}H_{12}O_4$ (208.0736) | 6 |  | | $C_{13}H_{12}O_6$ (264.0634) (No. 20; GUA+DMB, Table S1) | | |
| | $C_6H_6N_4O$ (150.0542) (No. 44; GUA+DMB+AN, Table S3) | | | 58 | $C_5H_6O_2$ (98.0368) | 3 |  |
| | $C_{13}H_{12}O_4$ (232.0736) (No. 6; GUA+DMB, Table S1) | | | | $C_{19}H_{16}O_6$ (340.0947) (No. 27; GUA+DMB, Table S1) | | |
| 53 | $C_{12}H_8N_2O_3$ (228.0535) | 10 |  | 59 | $C_{20}H_{16}O_7$ (368.0896) | 13 |  |
| 54 | $C_{11}H_{14}O_4$ (210.0892) | 5 |  | | $C_{21}H_{18}O_8$ (398.1002) (No. 19; GUA+DMB, Table S1) | | |
| | $C_7H_6O_4$ (154.0266) (No. 13; GUA+DMB, Table S1) | | | | $C_7H_6O_4$ (154.0266) (No. 13; GUA+DMB, Table S1) | | |
| 55 | $C_{14}H_{12}O_6$ (276.0634) | 9 |  | | $C_{15}H_{10}N_4O_3$ (294.0753) (No. 49; GUA+DMB+AN, Table S3) | | |
| 56 | $C_{14}H_{10}N_4O_7$ (346.0550) | 12 |  | | $C_{13}H_{10}O_6$ (262.0477) (No. 24; GUA+DMB, Table S1) | | |
| | $C_{13}H_{12}O_6$ (264.0634) (No. 20; GUA+DMB, Table S1) | | | | $C_5H_6O_5$ (146.0215) (No. 38; GUA+VL, Table S2) | | |

 **Table S5**. Possible structures of the major products detected from GUA+AN using UHPLC-
 HESI-Orbitrap-MS operated in positive (POS) and negative (NEG) ion modes.

| No. | GUA+AN POS Molecular formula and exact mass | DBE | Possible structure | No. | GUA+AN NEG Molecular formula and exact mass | DBE | Possible structure |
|---|---|---|---|---|---|---|---|
| | $C_{13}H_{10}O_4$ (230.0579) (No. 4; GUA+DMB, Table S1) | | | | $C_{14}H_{14}O_6$ (278.0790) (No. 16; GUA+DMB, Table S1) | | |
| | $C_6H_6N_4O$ (150.0542) (No. 44; GUA+DMB+AN, Table S3) | | | 68 | $C_{12}H_{19}N_3O$ (221.1528) | 5 |  |
| | $C_{11}H_{12}O_5$ (224.0685) (No. 8; GUA+DMB, Table S1) | | | | $C_{12}H_{10}O_4$ (218.0579) (No. 17; GUA+DMB, Table S1) | | |
| | $C_7H_8O_4$ (156.0423) (No. 28; GUA+VL, Table S2) | | | | $C_{14}H_{14}O_4$ (246.0892) (No. 1; GUA+DMB, Table S1) | | |
| 60 | $C_6H_4N_4$ (132.0436) | 7 |  | | $C_{20}H_{18}O_6$ (354.1103) (No. 21; GUA+DMB, Table S1) | | |
| 61 | $C_{12}H_{14}O_5$ (238.0841) | 6 |  | | $C_7H_{10}O_5$ (174.0528 (No. 18; GUA+DMB, Table S1) | | |
| 62 | $C_{13}H_{12}N_4O_5$ (304.0808) | 10 |  | 69 | $C_4H_3N_3O_3$ (141.0174) | 5 |  |
| | $C_{13}H_{12}O_6$ (264.0634) (No. 20; GUA+DMB, Table S1) | | | | $C_{13}H_{12}O_6$ (264.0634) (No. 20; GUA+DMB, Table S1) | | |
| | $C_{13}H_{12}O_4$ (232.0736) (No. 6; GUA+DMB, Table S1) | | | 70 | $C_{12}H_6N_4O_5$ (286.0338) | 12 |  |
| 63 | $C_8H_{10}N_4O$ (178.0855) | 6 |  | 71 | $C_{13}H_{12}O_5$ (248.0685) | 8 |  |
| 64 | $C_9H_{14}N_4O$ (194.1168) | 5 |  | 72 | $C_6H_6O_4$ (142.0266) | 4 |  |
| 65 | $C_8H_4N_4$ (156.0436) | 9 |  | | $C_{12}H_{10}O_3$ (202.0630) (No. 33; GUA+VL, Table S2) | | |
| 66 | $C_{15}H_{19}N_5O_2$ (301.1539) | 9 |  | 73 | $C_{12}H_{12}O_4$ (220.0736) | 7 |  |
| 67 | $C_7H_{10}N_4O_4$ (214.0702) | 5 |  | | $C_7H_6O_5$ (170.0215) (No. 15; GUA+DMB, Table S1) | | |
| | $C_7H_8O_5$ (172.0372) (No. 29; GUA+VL, Table S2) | | | | $C_7H_8O_4$ (156.0423) (No. 28; GUA+VL, Table S2) | | |

[Figure]

**Figure S1.** (a) The decay of GUA during (ammonium) nitrate-mediated photo-oxidation (GUA+AN) and photosensitized oxidation by $^3$VL* (GUA+VL) or $^3$DMB* (GUA+DMB). (b) The decay of DMB or VL during GUA photo-oxidation in GUA+DMB and GUA+VL, respectively. No statistically significant difference ($p > 0.05$) was noted between GUA+DMB and GUA+DMB+AN and between GUA+VL and GUA+VL+AN. Error bars represent 1 standard deviation; most error bars are smaller than the markers.

[Figure]

**Figure S2.** Signal-weighted distributions of aqSOA from GUA+DMB, GUA+VL,
GUA+DMB+AN, GUA+VL+AN, and GUA+AN. These product distributions were calculated
from UHPLC-HESI-Orbitrap-MS data obtained in the positive (POS) ion mode. The values
indicate the contribution of different product classifications to the total signals for each reaction
condition.

[Figure]

**Figure S3.** Signal-weighted distributions of aqSOA from GUA+DMB, GUA+VL,
GUA+DMB+AN, GUA+VL+AN, and GUA+AN. These product distributions were calculated
from UPLC-HESI-Orbitrap-MS data obtained in the negative (NEG) ion mode. The values
indicate the contribution of different product classifications to the total signals for each reaction
condition.

[Figure]

**Figure S43.** The concentration of formic, oxalic, and succinic acid for GUA+DMB, GUA+VL, GUA+DMB+AN, and GUA+VL+AN aqSOA. Error bars represent one standard deviation of triplicate experiments.

[Figure]

**Figure S54.** Van Krevelen diagrams of aqSOA from (a, b) GUA+DMB, (c, d) GUA+VL, (e, f) GUA+DMB+AN, and (g, h) GUA+VL+AN for positive (POS) and negative (NEG) ion modes. The blue circle markers indicate CHO classes, red triangle indicate CHON classes, and green diamond indicate CHN classes. The marker size reflects the relative abundance in the sample. The location of GUA, DMB, and VL in the plots are indicated only in panels a and c (red markers). The insets are expanded views of the crowded sections of the van Krevelen diagrams. Note the different scales on the axes.

[Figure]

**Figure S6**. Plots of the carbon oxidation state (OS$_C$) vs. the number of carbon atoms (n$_C$) of aqSOA from (a, b) GUA+DMB, (c, d) GUA+VL, (e, f) GUA+DMB+AN, and (g, h) GUA+VL+AN for positive (POS) and negative (NEG) ion modes, colored by the double bond equivalent (DBE) values. The circle, triangle, and diamond markers indicate CHO, CHON and CHN classes, respectively. The marker size reflects the relative abundance in the sample.

[Figure]

**Figure S7**. UV-Vis absorption spectra of GUA+DMB+AN, GUA+DMB, GUA+VL+AN, GUA+VL, and GUA+AN after 180 min of irradiation. The inset is the expanded view from 350 to 550 nm.

[Figure]

**Figure S86.** Plots of the double bond equivalent (DBE) values vs. the number of carbon atoms (nc) (Lin et al., 2018) of aqSOA from (a, b) GUA+DMB and GUA+VL, (c, d) GUA+DMB+AN and GUA+VL+AN, and (e, f) GUA+AN for positive (POS) and negative (NEG) ion modes. For a and b, the blue markers indicate CHO classes for GUA+DMB and red indicate CHO classes for GUA+VL. For c and d, the blue markers indicate CHO classes, red indicate CHON classes, and green indicate CHN classes for GUA+DMB+AN; the pink markers indicate CHO classes, cyan indicate CHON classes, and purple indicate CHN classes for GUA+VL+AN. For e and f, the blue markers indicate CHO classes, red indicate CHON classes, and green indicate

CHN classes for GUA+AN. The marker size reflects the relative abundance in the sample. The
three lines indicate DBE reference values of fullerene-like hydrocarbons (top, black solid line;
Lobodin et al, 2012), cata-condensed polycyclic aromatic hydrocarbons (PAHs; Siegmann and
Sattler, 2000) (middle, orange solid line), and linear conjugated polyenes (general formula
$C_xH_{x+2}$) (bottom, brown solid line). Species within the shaded area are potential BrC
chromophores.

[Figure]

**Figure S9.** Van Krevelen diagrams of aqSOA from GUA+AN for (a) positive (POS) and (b)
negative (NEG) ion modes. The blue markers indicate CHO classes, red indicate CHON
classes, and green indicate CHN classes. The marker size reflects the relative abundance in the
sample. The location of GUA is indicated only in panel a (black marker).

[Figure]

**Figure S10.** Plots of the carbon oxidation state ($OS_C$) vs. the number of carbon atoms ($n_C$) of
aqSOA from GUA+AN for (a) positive (POS) and (b) negative (NEG) ion modes, colored by
the double bond equivalent (DBE) values. The circle, triangle, and diamond markers indicate
CHO, CHON and CHN classes, respectively. The marker size reflects the relative abundance
in the sample.

[revised manuscript text omitted]

Yee, L. D., Kautzman, K. E., Loza, C. L., Schilling, K. A., Coggon, M. M., Chhabra, P. S.,
Chan, M. N., Chan, A. W. H., Hersey, S. P., Crounse, J. D., Wennberg, P. O., Flagan, R. C.,
and Seinfeld, J. H.: Secondary organic aerosol formation from biomass burning intermediates:
phenol    and    methoxyphenols,    Atmos.    Chem.    Phys.,    13,    8019–8043,
https://doi.org/10.5194/acp- 13-8019-2013, 2013.
Yu, L., Smith, J., Laskin, A., Anastasio, C., Laskin, J., and Zhang, Q.: Chemical
characterization of SOA formed from aqueous-phase reactions of phenols with the triplet
excited state of carbonyl and hydroxyl radical, Atmos. Chem. Phys., 14, 13801−13816,
https://doi.org/10.5194/acp-14-13801-2014, 2014.
Yu, L., Smith, J., Laskin, A., George, K. M., Anastasio, C., Laskin, J., Dillner, A. M., and
Zhang, Q.: Molecular transformations of phenolic SOA during photochemical aging in the
aqueous phase: competition among oligomerization, functionalization, and fragmentation,
Atmos. Chem. Phys., 16, 4511−4527, https://doi.org/10.5194/acp-16-4511-2016.

---

## Author Response (AR2)

Author Response for "Comparison of aqueous SOA product distributions from guaiacol oxidation by non-phenolic and phenolic methoxybenzaldehydes as photosensitizers in the absence and presence of ammonium nitrate" by Mabato et al.

We thank the Reviewer for their thorough comments. We have revised the manuscript accordingly, and below are our point-by-point responses (in blue) to the comments (in black) and changes to the manuscript (in red). In those changes that begin with line numbers, the original text is also in blue. In addition, please note that the line numbers in the responses correspond to those in the revised manuscript.

**Reviewer 1**

This work specially investigated the influences from photosentisized organics and nitrate on the aqSOA formation, which is rarely discussed in previous studies, and it further elucidate the product distributions under these two cases. And finally it proposes the implications of these processes to the aerol chemistry. The paper has been reviewed by a few experts and the authors have responded to their comments and revised the paper accordingly. This reviewer, as an addtional one, overall recommends its publication in ACP after some minor corrections suggested below (1) They are a few very recenly published articles regarding the chemistry induced by 3C* and its impact on the light absorption of the products, should be added, a more throughou literature search especailly in 2022 should be conducted and necessary

Response: Thank you for pointing this out. We have added recently published related papers to the introduction and discussion as follows:

Lines 33–35: Photosensitized reactions involving triplet excited states of organic compounds ($^3$C*) are efficient pathways for the formation of secondary organic aerosol in the aqueous phase (aqSOA; Smith et al., 2014, 2015, 2016; Yu et al., 2014, 2016; Chen et al., 2018; Lu et al., 2019; Ye et al., 2019; Chen et al., 2020; Liu et al., 2020; Jiang et al., 2021; Ma et al., 2021; Misovich et al., 2021; Ou et al., 2021; F. Li et al., 2022; X. Li et al., 2022; Aregahegn et al., 2022; Mabato et al., 2022; Wang et al., 2022).

Lines 48–50: Most previous studies on aqSOA formation via photosensitized non-carbonyl phenol oxidation have examined 3,4-dimethoxybenzaldehyde (DMB), a non-phenolic methoxybenzaldehyde, as the photosensitizer (Smith et al., 2014, 2015; Yu et al., 2014, 2016; Ye et al., 2019; Chen et al., 2020; Jiang et al., 2021; Ma et al., 2021; Misovich et al., 2021; Ou et al., 2021; X. Li et al., 2022).

Lines 289–292: The absorbance enhancement of phenolic aqSOA generated via reactions with $^3$C has been linked to the formation of conjugated structures due to oligomerization and functionalization (e.g., additions of hydroxyl and carbonyl groups; Yu et al., 2014, 2016; Smith et al., 2016; Ye et al., 2019; Chen et al., 2020; Jiang et al., 2021; Misovich et al., 2021; Ou et al., 2021; F. Li et al., 2022; X. Li et al., 2022; Mabato et al., 2022; Wang et al., 2022).

(2) Line 91, if DMB and VL are moderate and poor photosensitizers, why they were chosen, enen if they are relatively abudant in the real atmosphere.

Response: Comparable levels of phenolic and non-phenolic carbonyls have been reported in biomass burning smoke (Simoneit et al., 1993; Anastasio et al., 1997). Although certain phenolic carbonyls, including VL, can initiate photosensitized reactions and contribute to aqSOA formation (Smith et al., 2016; Mabato et al., 2022), photosensitization by phenolic carbonyls is less understood compared to photosensitization by DMB (a non-phenolic carbonyl), the most commonly used photosensitizer in related studies.

DMB and VL were chosen to represent non-phenolic and phenolic methoxybenzaldehydes photosensitizers, respectively, as (1) their structures differ only by one functional group ($-OCH_3$ for the former and $-OH$ for the latter), (2) they are both abundant in biomass burning emissions, and (3) there is available information on their photophysical properties (e.g., quantum yield of $^3C*$ formation and $^3C*$ lifetime) (Felber et al., 2021). This information was already provided in lines 86-92.

(3) Are the output light intensity from the Xe lamp stable over the course of the experiments? Did you have a test on that??

Response: The Xe lamp delivered stable output light intensity during our experiments. Based on our control experiments, the output light intensity only fluctuated by, on average, 0.9% within 3 h of operation (the duration of each experiment).

(4) How do you remove the influence of light absorption from the photosensitizers from that from the products? Their light absorption varied during the experiments, so a simple deduction can not do well,can you elaborate?

Response: The increase of light absorption from 350 to 550 nm, where GUA did not initially absorb light and where DMB and VL have little absorption, suggests the formation of light-absorbing products (Smith et al., 2016). This information was already provided in the SI (lines 106-108). In this work, the change in the integrated area of absorbance from 350 to 550 nm was therefore used to represent the absorbance enhancements. For the light absorption measurements in Figure 4a, the integrated area of absorbance from 350 to 550 nm at 180 min was normalized for the absorption of the reactants (GUA + photosensitizer; GUA + photosensitizer + AN; GUA + AN) by subtracting the corresponding values at 0 min (before irradiation) from each data point. Identifying the specific chromophores responsible for the absorbance enhancements can be performed using the methods employed in this study by matching the retention time (RT) of 
[revised manuscript text omitted]

**Section S1. UHPLC-PDA analyses**

An ultra-high performance liquid chromatography system (UHPLC, Waters Acquity H-Class, Waters, Milford, USA) coupled to a photodiode array (PDA) detector (Waters, Milford, USA) was used for the quantification of GUA, DMB, and VL concentrations. The samples were first filtered through a 0.2 μm Chromafil® Xtra PTFE filter (Macherey-Nagel GmbH & Co. KG, Germany). The separation of products was conducted using an Acquity HSS T3 column (1.8 μm, 2.1 mm × 100 mm; Waters Corp.). The column oven was held at 30 °C, and the autosampler was cooled at 4 °C. The injection volume was set to 5 µL. The binary mobile phase was composed of water (A) and acetonitrile (B). The gradient elution was performed at a flow rate of 0.2 mL/min: 0–1 min, 10% eluent B; 1–25 min, linear increase to 90% eluent B; 25–29.9 min, hold 90% eluent B; 29.9–30 min, decrease to 10% eluent B; 30–35 min, re-equilibrate at 10% eluent B for 5 min. GUA, DMB, and VL were analyzed using the channels with UV absorption at 274, 274, and 300 nm, respectively.

**Section S2. UHPLC-HESI-Orbitrap-MS analyses**

A Thermo Orbitrap Fusion Lumos Mass Spectrometry (Thermo Fisher Scientific, Waltham, MA, USA) connected to a Thermo Scientific UltiMate 3000 UHPLC system (Thermo Fisher Scientific, Waltham, MA, USA) via heated electrospray ionization (HESI) as the interface (UHPLC-HESI-Orbitrap-MS) was used to characterize the reaction products. The mobile phases used were 0.1% (v/v) formic acid (in milli-Q water) (A) and acetonitrile (B). The same settings (e.g., column, gradient, oven temperature) used in the UHPLC-PDA (Sect. S1) were applied in the UHPLC-HESI-Orbitrap-MS system. The HESI-MS spectra were acquired in both positive and negative ion modes. The HESI parameters were as follows: Spray voltage, 2500 V for both positive and negative HESI; sheath gas, 35 arbitrary units; nebulizer auxiliary gas, 10 arbitrary units; sweep gas, 3 arbitrary units. General instrumental parameters were set as follows: ion transfer tube temperature, 320 °C; vaporizer temperature, 350 °C. The mass range for full scan MS was set at 50-1000 m/z with a mass resolution of 60,000 at 200 m/z.

The automatic gain control (AGC) target was $4.0 \times 10^5$ with a maximum injection time of 50

ms. The UHPLC-HESI-Orbitrap-MS data obtained in positive and negative ion modes were pretreated using Progenesis QI (version 2.4; Nonlinear Dynamics) for peak picking and alignment. Most peaks detected in the blank (~99% for all experiments) were excluded from the samples except for peaks with a minimum of 2.5 times greater intensity in the sample spectrum than in the blank (Laskin et al., 2014). In addition, a peak was considered a product if the difference in the peak area between the samples before and after irradiation is ≥10 times.

In this work, two independently prepared samples for each reaction condition were analyzed using the UHPLC-HESI-Orbitrap-MS. Only peaks that were reproducibly detected in both sets of samples were retained. The formula assignments were carried out using the MIDAS

molecular formula calculator (http://magnet.fsu.edu/~midas/) with the following constraints: C

≤100, H ≤150, O ≤30, and N ≤10, and mass error of 10 ppm. The nitrogen atom was excluded in the constraints for experiments without AN. The ChemSpider database (Royal Society of

Chemistry) was also queried to return valid molecules that may be useful for proposing product structures. Overall, the proposed structures in this work are based on the molecular formulas,

DBE values, and structural and mechanistic information provided in earlier similar works on methoxyphenols (Yee et al., 2013; Li et al., 2014; Yu et al., 2014, 2016; He et al., 2019; Chen et al., 2020; Jiang et al., 2021; Misovich et al., 2021; Mabato et al., 2022). For clarity, the formulas discussed in this work correspond to neutral analytes (e.g., with $H^+$ or $NH_4^+$ removed from the ion formula).

The double bond equivalent (DBE) values (Koch and Dittmar, 2006) and carbon oxidation state ($OS_C$; Kroll et al., 2011, 2015; Lv et al., 2016) of the neutral formulas were calculated using the following equations:

$DBE = C - H/2 + N/2 + 1$              (Eq. S1)

$OS_C = 2 \times O/C + 3 \times N/C - H/C$        (Eq. S2)

where C, H, O, and N correspond to the number of carbon, hydrogen, oxygen, and nitrogen atoms in the neutral formula. Moreover, the average oxygen to carbon (O:C) ratios, $\langle O:C \rangle$:

($\langle O:C \rangle = \sum_i (\text{abundance}_i)O_i / \sum_i (\text{abundance}_i)C_i$), average nitrogen to carbon (N:C) ratios,

$\langle N:C \rangle$: ($\langle N:C \rangle = \sum_i (\text{abundance}_i)N_i / \sum_i (\text{abundance}_i)C_i$), and average hydrogen to carbon (H:C) ratios, $\langle H:C \rangle$: ($\langle H:C \rangle = \sum_i (\text{abundance}_i)H_i / \sum_i (\text{abundance}_i)C_i$) after the reactions were further estimated using the signal-weighted method (Bateman et al., 2012). The average

$OS_C$, $\langle OS_C \rangle$ was also calculated as follows:

$\langle OS_C \rangle = 2 \times \langle O:C \rangle + 3 \times \langle N:C \rangle - \langle H:C \rangle$       (Eq. S3)

**Section S3. IC analyses of small organic acids**

An ion chromatography system (IC, Dionex ICS-1100, Sunnyvale, CA) equipped with a

Dionex AS-DV autosampler (Sunnyvale, CA) enabled the analyses of small organic acids. The separation was achieved using an IonPac$^{TM}$ AS11 column (4 × 250 mm) with an IonPac$^{TM}$

AG11 guard column (4 × 50 mm). The isocratic elution was applied at a 1.0 mL/min flow rate with 12 mM sodium hydroxide (NaOH) as the eluent. The total run time was set at 10 min. The standard solutions (1–50 μM) of formic, succinic, and oxalic acid were analyzed three times along with the samples and water blank. Formic, succinic, and oxalic acid had retention times of 1.9 min, 3.7 min, and 5.9 min, respectively.

**Section S4. UV-Vis spectrophotometric analyses**

A UV-Vis spectrophotometer (UV-3600, Shimadzu Corp., Japan) was used to measure the absorbance changes for the samples. The absorbance values from 200 to 700 nm were measured instantly after sample collection, and measurements were done in triplicate. The change in the integrated area of absorbance from 350 to 550 nm was used to represent the absorbance enhancements. The increase of light absorption at this wavelength range, where GUA

did not initially absorb light and where DMB and VL have little absorption, suggests the formation of light-absorbing products (Smith et al., 2016).

**Section S5. Further discussions on van Krevelen diagrams and $OS_C$ vs. $n_C$ plots for**

**GUA+DMB and GUA+VL aqSOA**

Consistent with the higher contribution of ring-opening species, GUA+DMB had more products with H:C $\geq$1.5 and O:C $\leq$0.5 (Fig. S5a–b), possibly due to more oxygenated aliphatic species. GUA+VL (Fig. S5c–d) also had high-relative-abundance products with H:C of ~1 and

O:C $\geq$0.5. Similar to our previous work (0.1 mM GUA + 0.1 mM VL; Mabato et al., 2022), the two high-relative-abundance species with O:C $\geq$0.5 were associated with hydroxylated products ($C_7H_8O_4$ and $C_8H_8O_5$, #28 and 35; Table S2) that were also observed in earlier works on $^3DMB^*$ and $^\bullet OH$-mediated oxidation (Yu et al., 2014, 2016). These hydroxylated products were also present in GUA+DMB but with lower relative abundance. Triplet-mediated phenol oxidation can generate $H_2O_2$ (Anastasio et al., 1997), a photolytic source of $^\bullet OH$. Indeed, hydroxylation is significant in aqueous-phase phenol oxidation (Li et al., 2014; Yu et al., 2014,

2016; Chen et al., 2020; Jiang et al., 2021; Misovich et al., 2021; Mabato et al., 2022).

The $OS_C$ vs. $n_C$ plots for both GUA+DMB and GUA+VL display high-relative- abundance species clustered at $n_C$ of 12 to 15 and $OS_C$ >-1, which can be ascribed to dimers and derivatives (Fig. S6a–d). The species with $n_C$ >15 had the highest DBE values and can be attributed to trimers. These compounds were more abundant in GUA+DMB, likely due to the greater extent of photosensitized reactions by $^3DMB^*$ compared to $^3VL^*$. Indeed, oligomerization is an important process in aqSOA formation via triplet-mediated oxidation (Yu et al., 2014, 2016; Chen et al., 2020; Jiang et al., 2021; Misovich et al., 2021; Mabato et al.,

2022). As indicated by the higher quantity of low DBE species, ring-opening and fragmentation pathways were more extensive in GUA+DMB. In GUA+VL, there were also high-relative- abundance products with $n_C$ <10, $OS_C$ ≥0, and DBE <5, corresponding to the hydroxylated products mentioned earlier.

**Section S6. Estimation of the apparent quantum efficiency of guaiacol photodegradation**

The apparent quantum efficiency of GUA photodegradation ($\varphi_{GUA}$) in the presence of DMB,

VL, or nitrate during simulated sunlight illumination can be defined as (Anastasio et al., 1997;

Smith et al., 2014, 2016):

$\Phi_{GUA} = \dfrac{\text{mol GUA destroyed}}{\text{mol photons absorbed}}$            (Eq. S4)

$\Phi_{GUA}$ was calculated using the measured rate of GUA decay and rate of light absorption by

DMB, VL, or nitrate through the following equation:

$\Phi_{GUA} = \dfrac{\text{rate of GUA decay}}{\text{rate of light absorption by DMB or VL or nitrate}} = \dfrac{k'_{GUA} \times [GUA]}{\sum [(1-10^{-\varepsilon_\lambda [C] l}) \times I'_\lambda]}$    (Eq. S5)

where $k'_{GUA}$ is the pseudo-first-order rate constant for GUA decay, [GUA] is the concentration of GUA (M), $\varepsilon_\lambda$ is the base-10 molar absorptivity ($M^{-1}$ $cm^{-1}$) of DMB, VL, or nitrate at wavelength $\lambda$, [C] is the concentration of DMB, VL, or nitrate (M), $l$ is the pathlength of the illumination cell (cm), and $I'_\lambda$ is the volume-averaged photon flux (mol-photons $L^{-1}$ $s^{-1}$ $nm^{-1}$)

determined from 2NB actinometry:

$j(2NB) = 2.303 \times \Phi_{2NB} \times l \times \sum_{300\,nm}^{350\,nm} (\varepsilon_{2NB,\lambda} \times I'_\lambda \times \Delta\lambda)$       (Eq. S6)

where $j(2NB)$ is the decay rate constant of 2-nitrobenzaldehyde (2NB), the chemical actinometer used to determine the photon flux in the aqueous photoreactor, $\Phi_{2NB,\lambda}$ and $\varepsilon_{2NB,\lambda}$

are the quantum yield (molecule $photon^{-1}$) and base-10 molar absorptivity ($M^{-1}$ $cm^{-1}$) for 2NB, respectively, and $\Delta\lambda$ is the wavelength interval between actinic flux data points (nm).

**Section S7. Further discussions on van Krevelen diagrams and $OS_c$ vs. $n_C$ plots for**

**GUA+DMB+AN, GUA+VL+AN, and GUA+AN aqSOA**

The position of the CHO, CHON, and CHN species in the van Krevelen diagrams for

GUA+DMB+AN and GUA+VL+AN broadly resembled those of CHO species in the absence of AN (Fig. S5). The CHON species for GUA+DMB+AN and GUA+VL+AN mostly had O:C

ratios <0.7, consistent with previous studies on BBOA e.g., wheat straw burning in K-Puszta in the Great Hungarian Plain of Hungary, biomass burning at Canadian rural sites such as Saint

Anicet, and BBOA from Amazonia (Schmitt-Kopplin et al., 2010; Claeys et al., 2012;

Kourtchev et al., 2017).

 The CHN species in GUA+DMB+AN and GUA+VL+AN appeared to have analogous

H:C ratios. GUA+DMB+AN had ~2 times more CHON and CHN species than GUA+VL+AN, and there were more of these species with higher abundance in the former, indicating a greater extent of reactions with AN. The high-relative-abundance products for GUA+DMB+AN and

GUA+VL+AN were similar to those in the absence of AN, except the hydroxylated products (e.g., $C_7H_8O_4$; #28; Table S2) previously mentioned for GUA+VL. Among the high-relative- abundance products for GUA+DMB+AN was a CHN species with H:C of ~0.8. For

GUA+VL+AN, the high-relative-abundance products include two CHON species with O:C

and H:C ratios of 0.3-0.6 and 0.6-0.8. The major difference between GUA+AN and

GUA+DMB+AN/GUA+VL+AN was the presence of more high-relative-abundance CHON

and CHN species (Fig. S9) in GUA+AN which can be expected given that AN was the only source of oxidants in this case. Compared to GUA+AN, more species (CHO, CHON, and

CHN) were observed for GUA+DMB+AN and GUA+VL+AN, attributable to contributions from both photosensitization and (ammonium) nitrate photolysis.

 Moreover, GUA+DMB+AN and GUA+VL+AN aqSOA had mainly similar features in the $OS_C$ vs. $n_C$ plots as those observed in the absence of AN (Fig. S6). GUA+DMB+AN and

GUA+VL+AN aqSOA also had more CHON and CHN species with higher $OS_C$, $n_C$, and DBE

(Fig. S6e–h) relative to GUA+AN (Fig. S10), indicating more conjugated N-containing compounds. For GUA+DMB+AN and GUA+VL+AN, the CHON and CHN species had a wider range of $OS_C$ compared to CHO species (Fig. S6e–h). The high-relative-abundance species ($n_C$ of 12 to 15 and $OS_C$ >-1) corresponded to dimers and trimers similar to those noted in the absence of AN, along with some N-containing species. These include a CHN species with $n_C$ of 13, $OS_C$ ~0, and 11 DBE for GUA+DMB+AN, and 2 CHON species with $n_C$ of 5

and 11, $OS_C$ of 2.5 and 1, and 6 and 9 DBE for GUA+VL+AN.

**Table S1**. Possible structures of the major products detected from GUA+DMB using UHPLC-HESI-Orbitrap-MS operated in positive (POS) and negative (NEG) ion modes.

| No. | GUA+DMB POS Molecular formula and exact mass | DBE | Possible structure | No. | GUA+DMB NEG Molecular formula and exact mass | DBE | Possible structure |
|---|---|---|---|---|---|---|---|
| 1 | $C_{14}H_{14}O_4$ (246.0892) | 8 | *(structure)* | | $C_{14}H_{14}O_4$ (246.0892) (No. 1; GUA+DMB POS) | | |
| 2 | $C_{13}H_{10}O_3$ (214.0630) | 9 | *(structure)* | 16 | $C_{14}H_{14}O_6$ (278.0790) | 8 | *(structure)* |
| 3 | $C_{14}H_{12}O_4$ (244.0736) | 9 | *(structure)* | 17 | $C_{12}H_{10}O_4$ (218.0579) | 8 | *(structure)* |
| 4 | $C_{13}H_{10}O_4$ (230.0579) | 9 | *(structure)* | | $C_{13}H_{12}O_4$ (232.0736) (No. 6; GUA+DMB POS) | | |
| 5 | $C_{13}H_{10}O_5$ (246.0528) | 9 | *(structure)* | 18 | $C_7H_{10}O_5$ (174.0528) | 3 | *(structure)* |
| 6 | $C_{13}H_{12}O_4$ (232.0736) | 8 | *(structure)* | 19 | $C_{21}H_{18}O_8$ (398.1002) | 13 | *(structure)* |
| 7 | $C_{14}H_{12}O_5$ (260.0685) | 9 | *(structure)* | 20 | $C_{13}H_{12}O_6$ (264.0634) | 8 | *(structure)* |
| 8 | $C_{11}H_{12}O_5$ (224.0685) | 6 | *(structure)* | 21 | $C_{20}H_{18}O_6$ (354.1103) | 12 | *(structure)* |
| 9 | $C_{14}H_{12}O_7$ (292.0583) | 9 | *(structure)* | 22 | $C_{14}H_{14}O_7$ (294.0740) | 8 | *(structure)* |
| 10 | $C_{11}H_{14}O_6$ (242.0790) | 5 | *(structure)* | 23 | $C_{12}H_{14}O_4$ (222.0892) | 6 | *(structure)* |

| | | | | | | |
|---|---|---|---|---|---|---|
| 11 | $C_{18}H_{18}O_7$ (346.1053) | 10 |  | 24 | $C_{13}H_{10}O_6$ (262.0477) | 9 |  |
| 12 | $C_{10}H_{12}O_3$ (180.0786) | 5 |  | 25 | $C_{13}H_{14}O_4$ (234.0892) | 7 |  |
| 13 | $C_7H_6O_4$ (154.0266) | 5 |  | 26 | $C_{14}H_{14}O_5$ (262.0841) | 8 |  |
| 14 | $C_{16}H_{18}O_6$ (306.1103) | 8 |  | $C_{13}H_{10}O_5$ (246.0528) (No. 5; GUA+DMB POS) | | |
| 15 | $C_7H_6O_5$ (170.0215) | 5 |  | 27 | $C_{19}H_{16}O_6$ (340.0947) | 12 |  |

**Table S2**. Possible structures of the major products detected from GUA+VL using UHPLC-
HESI- Orbitrap-MS operated in positive (POS) and negative (NEG) ion modes.

| No. | GUA+VL POS Molecular formula and exact mass | DBE | Possible structure | No. | GUA+VL NEG Molecular formula and exact mass | DBE | Possible structure |
|---|---|---|---|---|---|---|---|
| 28 | $C_7H_8O_4$ (156.0423) | 4 |  | 35 | $C_8H_8O_5$ (184.0372) | 5 |  |
| $C_{13}H_{10}O_4$ (230.0579) (No. 4; GUA+DMB, Table S1) | | | | $C_{13}H_{12}O_4$ (232.0736) (No. 6; GUA+DMB, Table S1) | | | |
| $C_{13}H_{12}O_4$ (232.0736) (No. 6; GUA+DMB, Table S1) | | | | $C_{14}H_{14}O_4$ (246.0892) (No. 1; GUA+DMB, Table S1) | | | |
| $C_{13}H_{10}O_5$ (246.0528) (No. 5; GUA+DMB, Table S1) | | | | $C_{14}H_{14}O_6$ (278.0790) (No. 16; GUA+DMB, Table S1) | | | |
| 29 | $C_7H_8O_5$ (172.0372) | 4 |  | $C_{20}H_{18}O_6$ (354.1103) (No. 21; GUA+DMB, Table S1) | | | |
| 30 | $C_6H_6O_2$ (110.0368) | 4 |  | $C_{12}H_{10}O_4$ (218.0579) (No. 17; GUA+DMB, Table S1) | | | |
| 31 | $C_{10}H_{10}O_4$ (194.0579) | 6 |  | $C_6H_6O_2$ (110.0368) (No. 30; GUA+VL POS) | | | |
| 32 | $C_{11}H_8O_4$ (204.0423) | 8 |  | $C_7H_{10}O_5$ (174.0528) (No. 18; GUA+DMB, Table S1) | | | |
| 33 | $C_{12}H_{10}O_3$ (202.0630) | 8 |  | 36 | $C_{15}H_{14}O_5$ (274.0841) | 9 |  |
| $C_{14}H_{12}O_5$ (260.0685) (No. 7; GUA+DMB, Table S1) | | | | $C_{13}H_{12}O_6$ (264.0634) (No. 20; GUA+DMB, Table S1) | | | |
| $C_{13}H_{14}O_4$ (234.0892) (No. 25; GUA+DMB, Table S1) | | | | 37 | $C_8H_8O_4$ (168.0423) | 5 |  |
| 34 | $C_{11}H_{10}O_6$ (238.0477) | 7 |  | $C_{19}H_{16}O_6$ (340.0947) (No. 27; GUA+DMB, Table S1) | | | |
| $C_{13}H_{10}O_6$ (262.0477) (No. 24; GUA+DMB, Table S1) | | | | $C_{11}H_{10}O_6$ (238.0477) (No. 34; GUA+VL POS) | | | |
| $C_{13}H_{12}O_6$ (264.0634) (No. 20; GUA+DMB, Table S1) | | | | 38 | $C_5H_6O_5$ (146.0215) | 3 |  |
| $C_7H_6O_4$ (154.0266) (No. 13; GUA+DMB, Table S1) | | | | 39 | $C_6H_4O_4$ (140.0110) | 5 |  |

 **Table S3**. Possible structures of the major products detected from GUA+DMB+AN using
 UHPLC-HESI-Orbitrap-MS operated in positive (POS) and negative (NEG) ion modes.

| No. | GUA+DMB+AN POS Molecular formula and exact mass | DBE | Possible structure | No. | GUA+DMB +AN NEG Molecular formula and exact mass | DBE | Possible structure |
|---|---|---|---|---|---|---|---|
| $C_{14}H_{14}O_4$ (246.0892) (No. 1; GUA+DMB, Table S1) | | | | $C_{13}H_{12}O_4$ (232.0736) (No. 6; GUA+DMB, Table S1) | | | |
| 40 | $C_{13}H_{10}N_4$ (222.0905) | 11 |  | $C_{14}H_{14}O_6$ (278.0790) (No. 16; GUA+DMB, Table S1) | | | |
| $C_{13}H_{10}O_5$ (246.0528) (No. 5; GUA+DMB, Table S1) | | | | $C_{14}H_{14}O_4$ (246.0892) (No. 1; GUA+DMB, Table S1) | | | |
| $C_{13}H_{10}O_4$ (230.0579) (No. 4; GUA+DMB, Table S1) | | | | $C_{12}H_{10}O_4$ (218.0579) (No. 17; GUA+DMB, Table S1) | | | |
| 41 | $C_6H_6N_4$ (134.0592) | 6 |  | $C_{21}H_{18}O_8$ (398.1002) (No. 19; GUA+DMB, Table S1) | | | |
| $C_{13}H_{12}O_4$ (232.0736) (No. 6; GUA+DMB, Table S1) | | | | $C_7H_{10}O_5$ (174.0528) (No. 18; GUA+DMB, Table S1) | | | |
| 42 | $C_{12}H_{11}N_3O_3$ (245.0800) | 9 |  | $C_{13}H_{12}O_6$ (264.0634) (No. 20; GUA+DMB, Table S1) | | | |
| 43 | $C_{10}H_8N_4O$ (200.0698) | 9 |  | 48 | $C_{16}H_{14}N_6O_4$ (354.1076) | 13 |  |
| 44 | $C_6H_6N_4O$ (150.0542) | 6 |  | 49 | $C_{15}H_{10}N_4O_3$ (294.0753) | 13 |  |
| 45 | $C_{10}H_{14}N_4O_4$ (245.1015) | 6 |  | $C_{13}H_{10}O_6$ (262.0477) (No. 24; GUA+DMB, Table S1) | | | |
| 46 | $C_{13}H_{10}N_4O$ (238.0855) | 11 |  | $C_{10}H_{10}O_4$ (194.0579) (No. 31; GUA+VL, Table S2) | | | |
| $C_{13}H_{12}O_6$ (264.0634) (No. 20; GUA+DMB, Table S1) | | | | $C_7H_8O_4$ (156.0423) (No. 28; GUA+VL, Table S2) | | | |
| $C_7H_6O_4$ (154.0266) (No. 13; GUA+DMB, Table S1) | | | | $C_{13}H_{14}O_4$ (234.0892) (No. 25; GUA+DMB, Table S1) | | | |
| $C_{12}H_{10}O_3$ (202.0630) (No. 33; GUA+VL, Table S2) | | | | $C_{13}H_{10}O_5$ (246.0528) (No. 5; GUA+DMB, Table S1) | | | |
| 47 | $C_{13}H_8O_4$ (228.0423) | 10 |  | $C_{14}H_{14}O_5$ (262.0841) (No. 26; GUA+DMB, Table S1) | | | |

**Table S4**. Possible structures of the major products detected from GUA+VL+AN using
UHPLC-HESI-Orbitrap-MS operated in positive (POS) and negative (NEG) ion modes.

| No. | GUA+VL+AN POS Molecular formula and exact mass | DBE | Possible structure | No. | GUA+VL+AN NEG Molecular formula and exact mass | DBE | Possible structure |
|---|---|---|---|---|---|---|---|
| $C_{14}H_{14}O_4$ (246.0892) (No. 1; GUA+DMB, Table S1) | | | | $C_{13}H_{12}O_4$ (232.0736) (No. 6; GUA+DMB, Table S1) | | | |
| 50 | $C_{10}H_8O_2$ (160.0524) | 7 | | $C_{14}H_{14}O_6$ (278.0790) (No.16; GUA+DMB, Table S1) | | | |
| 51 | $C_{16}H_{18}O_4$ (274.1205) | 8 | | $C_{12}H_{10}O_4$ (218.0579) (No.17; GUA+DMB, Table S1) | | | |
| $C_{11}H_{12}O_5$ (224.0685) (No. 8; GUA+DMB, Table S1) | | | | 57 | $C_{11}H_9N_3O_3$ (231.0644) | 9 | |
| $C_{14}H_{12}O_5$ (260.068) (No. 7; GUA+DMB, Table S1) | | | | $C_7H_{10}O_5$ (174.0528) (No.18; GUA+DMB, Table S1) | | | |
| $C_{12}H_{14}O_4$ (222.0892) (No. 23; GUA+DMB, Table S1) | | | | $C_{15}H_{14}O_5$ (274.0841) (No. 36; GUA+VL, Table S2) | | | |
| 52 | $C_{11}H_{12}O_4$ (208.0736) | 6 | | $C_{13}H_{12}O_6$ (264.0634) (No. 20; GUA+DMB, Table S1) | | | |
| $C_6H_6N_4O$ (150.0542) (No. 44; GUA+DMB+AN, Table S3) | | | | 58 | $C_5H_6O_2$ (98.0368) | 3 | |
| $C_{13}H_{12}O_4$ (232.0736) (No. 6; GUA+DMB, Table S1) | | | | $C_{19}H_{16}O_6$ (340.0947) (No. 27; GUA+DMB, Table S1) | | | |
| 53 | $C_{12}H_8N_2O_3$ (228.0535) | 10 | | 59 | $C_{20}H_{16}O_7$ (368.0896) | 13 | |
| 54 | $C_{11}H_{14}O_4$ (210.0892) | 5 | | $C_{21}H_{18}O_8$ (398.1002) (No. 19; GUA+DMB, Table S1) | | | |
| $C_7H_6O_4$ (154.0266) (No. 13; GUA+DMB, Table S1) | | | | $C_7H_6O_4$ (154.0266) (No. 13; GUA+DMB, Table S1) | | | |
| 55 | $C_{14}H_{12}O_6$ (276.0634) | 9 | | $C_{15}H_{10}N_4O_3$ (294.0753) (No. 49; GUA+DMB+AN, Table S3) | | | |
| 56 | $C_{14}H_{10}N_4O_7$ (346.0550) | 12 | | $C_{13}H_{10}O_6$ (262.0477) (No. 24; GUA+DMB, Table S1) | | | |
| $C_{13}H_{12}O_6$ (264.0634) (No. 20; GUA+DMB, Table S1) | | | | $C_5H_6O_5$ (146.0215) (No. 38; GUA+VL, Table S2) | | | |

 **Table S5**. Possible structures of the major products detected from GUA+AN using UHPLC-
 HESI-Orbitrap-MS operated in positive (POS) and negative (NEG) ion modes.

| No. | GUA+AN POS Molecular formula and exact mass | DBE | Possible structure | No. | GUA+AN NEG Molecular formula and exact mass | DBE | Possible structure |
|---|---|---|---|---|---|---|---|
| | $C_{13}H_{10}O_4$ (230.0579) (No. 4; GUA+DMB, Table S1) | | | | $C_{14}H_{14}O_6$ (278.0790) (No. 16; GUA+DMB, Table S1) | | |
| | $C_6H_6N_4O$ (150.0542) (No. 44; GUA+DMB+AN, Table S3) | | | 68 | $C_{12}H_{19}N_3O$ (221.1528) | 5 |  |
| | $C_{11}H_{12}O_5$ (224.0685) (No. 8; GUA+DMB, Table S1) | | | | $C_{12}H_{10}O_4$ (218.0579) (No. 17; GUA+DMB, Table S1) | | |
| | $C_7H_8O_4$ (156.0423) (No. 28; GUA+VL, Table S2) | | | | $C_{14}H_{14}O_4$ (246.0892) (No. 1; GUA+DMB, Table S1) | | |
| 60 | $C_6H_4N_4$ (132.0436) | 7 |  | | $C_{20}H_{18}O_6$ (354.1103) (No. 21; GUA+DMB, Table S1) | | |
| 61 | $C_{12}H_{14}O_5$ (238.0841) | 6 |  | | $C_7H_{10}O_5$ (174.0528 (No. 18; GUA+DMB, Table S1) | | |
| 62 | $C_{13}H_{12}N_4O_5$ (304.0808) | 10 |  | 69 | $C_4H_3N_3O_3$ (141.0174) | 5 |  |
| | $C_{13}H_{12}O_6$ (264.0634) (No. 20; GUA+DMB, Table S1) | | | | $C_{13}H_{12}O_6$ (264.0634) (No. 20; GUA+DMB, Table S1) | | |
| | $C_{13}H_{12}O_4$ (232.0736) (No. 6; GUA+DMB, Table S1) | | | 70 | $C_{12}H_6N_4O_5$ (286.0338) | 12 |  |
| 63 | $C_8H_{10}N_4O$ (178.0855) | 6 |  | 71 | $C_{13}H_{12}O_5$ (248.0685) | 8 |  |
| 64 | $C_9H_{14}N_4O$ (194.1168) | 5 |  | 72 | $C_6H_6O_4$ (142.0266) | 4 |  |
| 65 | $C_8H_4N_4$ (156.0436) | 9 |  | | $C_{12}H_{10}O_3$ (202.0630) (No. 33; GUA+VL, Table S2) | | |
| 66 | $C_{15}H_{19}N_5O_2$ (301.1539) | 9 |  | 73 | $C_{12}H_{12}O_4$ (220.0736) | 7 |  |
| 67 | $C_7H_{10}N_4O_4$ (214.0702) | 5 |  | | $C_7H_6O_5$ (170.0215) (No. 15; GUA+DMB, Table S1) | | |
| | $C_7H_8O_5$ (172.0372) (No. 29; GUA+VL, Table S2) | | | | $C_7H_8O_4$ (156.0423) (No. 28; GUA+VL, Table S2) | | |

[Figure]

**Figure S1.** (a) The decay of GUA during (ammonium) nitrate-mediated photo-oxidation (GUA+AN) and photosensitized oxidation by $^3$VL* (GUA+VL) or $^3$DMB* (GUA+DMB). (b) The decay of DMB or VL during GUA photo-oxidation in GUA+DMB and GUA+VL, respectively. No statistically significant difference ($p > 0.05$) was noted between GUA+DMB and GUA+DMB+AN and between GUA+VL and GUA+VL+AN. Error bars represent 1 standard deviation; most error bars are smaller than the markers.

[Figure]

**Figure S2.** Signal-weighted distributions of aqSOA from GUA+DMB, GUA+VL,
GUA+DMB+AN, GUA+VL+AN, and GUA+AN. These product distributions were calculated
from UHPLC-HESI-Orbitrap-MS data obtained in the positive (POS) ion mode. The values
indicate the contribution of different product classifications to the total signals for each reaction
condition.

[Figure]

**Figure S3.** Signal-weighted distributions of aqSOA from GUA+DMB, GUA+VL,
GUA+DMB+AN, GUA+VL+AN, and GUA+AN. These product distributions were calculated
from UPLC-HESI-Orbitrap-MS data obtained in the negative (NEG) ion mode. The values
indicate the contribution of different product classifications to the total signals for each reaction
condition.

[Figure]

**Figure S4.** The concentration of formic, oxalic, and succinic acid for GUA+DMB, GUA+VL,
GUA+DMB+AN, and GUA+VL+AN aqSOA. Error bars represent one standard deviation of
triplicate experiments.

[Figure]

[Figure]

**Figure S5.** Van Krevelen diagrams of aqSOA from (a, b) GUA+DMB, (c, d) GUA+VL, (e, f) GUA+DMB+AN, and (g, h) GUA+VL+AN for positive (POS) and negative (NEG) ion modes. The blue circle markers indicate CHO classes, red triangle indicate CHON classes, and green diamond indicate CHN classes. The marker size reflects the relative abundance in the sample. The location of GUA, DMB, and VL in the plots are indicated only in panels a and c (red markers). The insets are expanded views of the crowded sections of the van Krevelen diagrams. Note the different scales on the axes.

[Figure]

**Figure S6**. Plots of the carbon oxidation state ($OS_C$) vs. the number of carbon atoms ($n_C$) of aqSOA from (a, b) GUA+DMB, (c, d) GUA+VL, (e, f) GUA+DMB+AN, and (g, h) GUA+VL+AN for positive (POS) and negative (NEG) ion modes, colored by the double bond equivalent (DBE) values. The circle, triangle, and diamond markers indicate CHO, CHON and CHN classes, respectively. The marker size reflects the relative abundance in the sample.

[Figure]

**Figure S7**. UV-Vis absorption spectra of GUA+DMB+AN, GUA+DMB, GUA+VL+AN, GUA+VL, and GUA+AN after 180 min of irradiation. The inset is the expanded view from 350 to 550 nm.

[Figure]

**Figure S8.** Plots of the double bond equivalent (DBE) values vs. the number of carbon atoms ($n_C$) (Lin et al., 2018) of aqSOA from (a, b) GUA+DMB and GUA+VL, (c, d) GUA+DMB+AN and GUA+VL+AN, and (e, f) GUA+AN for positive (POS) and negative (NEG) ion modes. For a and b, the blue markers indicate CHO classes for GUA+DMB and red indicate CHO classes for GUA+VL. For c and d, the blue markers indicate CHO classes, red indicate CHON classes, and green indicate CHN classes for GUA+DMB+AN; the pink markers indicate CHO classes, cyan indicate CHON classes, and purple indicate CHN classes for GUA+VL+AN. For e and f, the blue markers indicate CHO classes, red indicate CHON classes, and green indicate

CHN classes for GUA+AN. The marker size reflects the relative abundance in the sample. The
three lines indicate DBE reference values of fullerene-like hydrocarbons (top, black solid line;
Lobodin et al, 2012), cata-condensed polycyclic aromatic hydrocarbons (PAHs; Siegmann and
Sattler, 2000) (middle, orange solid line), and linear conjugated polyenes (general formula
$C_xH_{x+2}$) (bottom, brown solid line). Species within the shaded area are potential BrC
chromophores.

[Figure]

**Figure S9.** Van Krevelen diagrams of aqSOA from GUA+AN for (a) positive (POS) and (b)
negative (NEG) ion modes. The blue markers indicate CHO classes, red indicate CHON
classes, and green indicate CHN classes. The marker size reflects the relative abundance in the
sample. The location of GUA is indicated only in panel a (black marker).

[Figure]

**Figure S10**. Plots of the carbon oxidation state ($OS_C$) vs. the number of carbon atoms ($n_C$) of
aqSOA from GUA+AN for (a) positive (POS) and (b) negative (NEG) ion modes, colored by
the double bond equivalent (DBE) values. The circle, triangle, and diamond markers indicate
CHO, CHON and CHN classes, respectively. The marker size reflects the relative abundance
in the sample.